# Unleashing Scientific Reasoning for Bio-experimental Protocol Generation via Structured Component-based Reward Mechanism

**Haoran Sun**[1,2*], **Yankai Jiang**[1*✉], **Zhenyu Tang**[1,3], **Yaning Pan**[1,2], **Shuang Gu**[1,3],
**Zekai Lin**[2], **Lilong Wang**[1], **Wenjie Lou**[1], **Lei Liu**[2], **Lei Bai**[1], **Xiaosong Wang**[1✉]
[1]Shanghai Artificial Intelligence Laboratory
[2]Fudan University
[3]Shanghai Jiao Tong University
{jiangyankai,wangxiaosong}@pjlab.org.cn

## Abstract

The foundation of reproducible science lies in protocols that are precise, logically ordered, and executable. The autonomous generation of these protocols through natural language queries could greatly improve the efficiency of the reproduction process. However, current leading large language models (LLMs) often generate incomplete or inconsistent protocols, limiting their utility. To address this limitation, we first introduce SciRecipe, a large-scale dataset of over 12K structured protocols spanning 27 biological subfields and encompassing both comprehension and problem-solving tasks. To further improve protocol generation, we propose the "Sketch-and-Fill" paradigm, which separates analysis, structuring, and expression to ensure each step is explicit and verifiable. Complementing this, the structured component-based reward mechanism evaluates step granularity, action order, and semantic fidelity, aligning model optimization with experimental reliability. Building on these components, we develop Thoth, trained through a staged Knowledge-to-Action process that progresses from knowledge acquisition to operational reasoning and ultimately to robust, executable protocol generation. Across multiple benchmarks, Thoth consistently surpasses both proprietary and open-source LLMs, achieving significant improvements in step alignment, logical sequencing, and semantic accuracy. Our approach paves the way for reliable scientific assistants that bridge knowledge with experimental execution. Code is publicly available at https://github.com/InternScience/Thoth.

## 1 Introduction

The planning and execution of scientific experiments hinge on protocols that serve not merely as textual guidelines but as operational blueprints detailing procedures, materials, and logical dependencies (Freedman et al., 2015; Goodman et al., 2016). A well-structured protocol ensures that experiments are reproducible, safe, and scientifically valid (of Sciences et al., 2019), which is essential for cumulative progress in the life sciences. While recent advances in large language models (LLMs) have greatly expanded their role in biomedical research, ranging from literature-based discovery to domain-specific question answering (Devlin et al., 2019; Brown et al., 2020; Lee et al., 2020; Beltagy et al., 2019), their ability to produce reliable experimental protocols remains underdeveloped. Existing datasets and benchmarks are typically confined to comprehension tasks (Liu et al., 2025; Jiang et al., 2024), thereby neglecting the dimensions of planning and problem solving required to support reproducibility and practical execution. As a result, researchers often find that models offer fragmented recommendations on experimental procedures but fall short of producing concise, logically ordered protocols that can be directly implemented in laboratory workflows (O'Donoghue et al., 2023; Yi et al., 2024).

---

*Equal Contribution. ✉Corresponding authors.

Currently, proprietary models (Achiam et al., 2023; Comanici et al., 2025; Jaech et al., 2024; OpenAI, 2025) such as GPT-5 demonstrate strong capabilities in procedural reasoning, while domain-specific scientific systems complement LLMs with curated knowledge bases or tool-assisted pipelines (Huang et al., 2025; Jin et al., 2025). Despite these advances, the generated protocols often contain unordered steps, redundant operations, factual inconsistencies, or hallucinated actions (O'Donoghue et al., 2023; Yi et al., 2024; Jiang et al., 2024), undermining both reproducibility and scientific credibility. Moreover, evaluation remains a central bottleneck: metrics such as BLEU (Papineni et al., 2002), ROUGE (Lin, 2004), and BERTScore (Zhang et al., 2019) capture only superficial lexical overlap, failing to reflect whether the generated action sequence is logically consistent, semantically faithful, and practically executable. While "LLM-as-a-judge" frameworks align better with human preferences, they introduce prohibitive costs to reinforcement learning (RL) pipelines, limiting scalability (Liu et al., 2023; Zheng et al., 2023). Existing reward designs further neglect the structured and verifiable nature of protocols, producing outputs that are linguistically fluent but experimentally unreliable (Zeng et al., 2023). These limitations highlight the need for an approach that integrates structured and efficient evaluation.

In this work, we introduce a comprehensive framework that advances both data and modeling for protocol generation. At its core lies the "Sketch-and-Fill" paradigm, which formulates protocol generation as a structured reasoning process: each step is decomposed into essential components and expressed in natural language with explicit correspondence, ensuring logical coherence and experimental verifiability. To support this paradigm, we curate SciRecipe, a large-scale dataset of over 12K protocols spanning diverse domains and covering both Protocol-Comprehension and Problem-Solving tasks. Building on this foundation, we propose the **S**tructured **CO**mponent-based **RE**ward (SCORE) mechanism, the central innovation of our framework. SCORE provides a structured reward and evaluation scheme that captures three complementary dimensions: step granularity (controlling scale and avoiding redundancy), action ordering (ensuring logically consistent sequences for reproducibility), and semantic fidelity (verifying alignment between predicted and reference actions, objects, and parameters). By jointly modeling these dimensions, SCORE moves beyond conventional text-based metrics to directly assess whether protocols are executable, interpretable, and scientifically sound. It serves both as an effective RL training signal and as a reliable metric for evaluation. Based on these components, we develop Thoth, a protocol-generation model trained to combine structured reasoning with SCORE-guided evaluation. Extensive experiments demonstrate that Thoth outperforms SOTA models, particularly in step alignment, logical sequencing, and semantic accuracy. Just as importantly, the protocols it generates are concise and reproducible, which are qualities often absent from existing systems. Our main contributions are as follows:

- We curate **SciRecipe**, a large-scale, multi-task dataset covering over 27 biological subfields, designed to serve as a foundation for both training and evaluating on protocol generation.
- We introduce the **Sketch-and-Fill paradigm**, a reasoning framework that aligns with the logic of experimental design by converting open-ended queries into verifiable protocols.
- We propose the **SCORE mechanism**, a structured reward and evaluation framework that jointly measures step granularity, order consistency, and semantic fidelity, ensuring that protocols are not only linguistically fluent but also experimentally executable.
- We develop **Thoth**, a protocol-generation model with strong reasoning abilities, which achieves SOTA performance across protocol-specific and broader scientific benchmarks.

## 2 RELATED WORKS

**LLMs in the Life Science** Recent progress in life science LLMs has shifted from general-purpose systems to domain-specific models. BioBERT and SciBERT (Lee et al., 2020; Beltagy et al., 2019), pretrained on large-scale biomedical corpora, substantially improved tasks such as information extraction and question answering (QA). Autoregressive models like BioGPT (Luo et al., 2022) further advanced generative abilities, achieving strong and consistent results on PubMedQA (Jin et al., 2019). These works collectively demonstrate the effectiveness of domain-specific pretraining and task adaptation, yet remain limited to knowledge-based tasks (e.g., QA, summarization) and cannot produce executable protocols. Related to our goal, BioPlanner (O'Donoghue et al., 2023) proposes a pseudocode-based evaluation framework for assessing biological protocol planning. While valuable for benchmarking LLM reasoning, it focuses on pseudocode reconstruction rather than generating

natural-language, experimentally executable protocols, making it complementary to our work. In contrast, proprietary models (OpenAI, 2025; Comanici et al., 2025; Jaech et al., 2024) such as GPT-5 leverage massive corpora, parameter scales, and reasoning mechanisms to generate preliminary multi-step protocols. More recent efforts (Huang et al., 2025; Jin et al., 2025; Zhang et al., 2025c; Gao et al., 2025), including Biomni and STELLA, integrate external knowledge and tools to support task-oriented solutions. Nonetheless, current models still struggle with redundancy, misordered steps, and hallucinations, thereby limiting their practical usability. To address this gap, we propose a biologically oriented model trained on real protocols, designed to generate rigorous experimental procedures through a structured scientific reasoning paradigm.

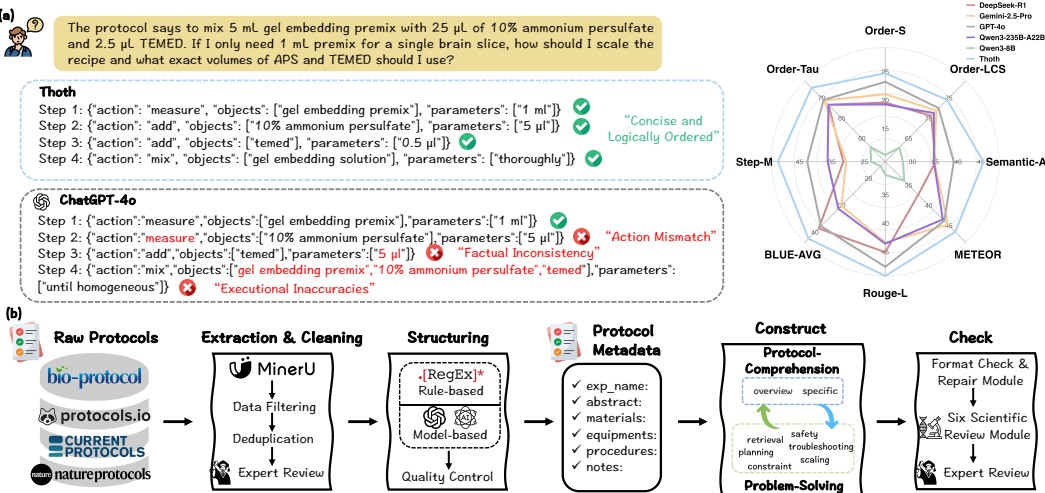

Figure 1: (a) Thoth's performance advantages over other models, shown through qualitative evaluation (left) and quantitative evaluation (right). (b) The construction pipeline of the SciRecipe dataset.

**Reward Design and Evaluation for Open-Ended Generation** Designing effective evaluation remains a key challenge in open-ended generation. Unlike short-text tasks with clear ground truth, long-text generation such as experimental protocols lacks definitive answers and must therefore be assessed along multiple dimensions, including coherence, factuality, and executability (Becker et al., 2024; Que et al., 2024). Metrics such as ROUGE, BLEU, and BERTScore (Lin, 2004; Papineni et al., 2002; Zhang et al., 2019) capture surface similarity but fail to fully reflect reasoning depth and often diverge from human judgment. The "LLM-as-a-judge" paradigm provides closer alignment with human preferences (Gu et al., 2024; Zheng et al., 2023), yet it usually introduces prohibitive computational costs in RL training (Shao et al., 2024; Schulman et al., 2017; Ahmadian et al., 2024; Hu, 2025). Alternative approaches, including smaller evaluator models or reward mechanisms such as PrefBERT, Direct Reasoning Optimization, and LoVeC (Li et al., 2025; Xu et al., 2025; Zhang et al., 2025a), can improve efficiency or reliability but still require repeated model calls. To address these issues, we propose a rule-based evaluation that extracts key protocol elements through structured reasoning, thereby offering direct and verifiable reward signals with efficiency and interpretability.

## 3 METHODS

This section outlines our approach to leveraging high-quality experimental protocol data and a component-wise reward mechanism to enhance the model's ability to generate experimental solutions. We first propose the "Sketch-and-Fill" reasoning paradigm, which aligns with experimental execution logic to guide the model in organizing experimental steps effectively. Building on this, we construct SciRecipe, a dataset encompassing diverse experimental tasks and scenarios. Subsequently, we develop the SCORE mechanism to evaluate protocol quality across multiple dimensions, serving as a reward signal to improve the model's generalization in protocol generation tasks. Ultimately, by leveraging SciRecipe and a multi-stage Knowledge-to-Action learning strategy, we train the Thoth model, which exhibits strong reasoning capabilities for protocol generation.

## 3.1 SCIRECIPE DATASET

In life science research, protocols serve as essential documents to ensure experimental reproducibility and reliability, providing detailed records of materials, equipment, procedures, and critical notes. Recently, platforms such as Nature Protocols, Bio-protocol, and Protocols.io (Wikipedia contributors, 2025; Bio-protocol, LLC, 2025; Teytelman et al., 2016) have compiled a vast number of standardized workflows, offering rich resources for the scientific community. From these sources, we collected over 23K protocols spanning 27 subfields including neuroscience, molecular biology, and cancer biology. After cleaning and structural processing, approximately 12K high-quality data were retained as the foundation of our dataset (the preprocessing pipeline is detailed in the Appendix A.1). Building on this foundation, we introduce the **SciRecipe** dataset, designed to improve and evaluate LLMs in experimental protocol understanding and generation. SciRecipe comprises eight task types, grouped into two categories: Protocol-Comprehension Tasks (overview and specific), targeting global summarization and fine-grained analysis, and Problem-Solving Tasks (retrieval, planning, troubleshooting, constraint, scaling, and safety), simulating typical challenges encountered throughout experimental workflows. Together, these tasks form a complementary "understanding–application" loop. The construction pipeline of SciRecipe is illustrated in Figure 1 and further elaborated in the Appendix A.4. Notably, benchmarks focused on scientific experimental protocol generation are currently scarce (Liu et al., 2025). To address this gap, we introduce **SciRecipe-Eval**, built with a similar pipeline, with dataset splits and difficulty levels detailed in the Appendix B.

## 3.2 "SKETCH-AND-FILL" REASONING PARADIGM

To transform protocol generation tasks into an executable and evaluable form, we propose the "Sketch-and-Fill" reasoning paradigm. This paradigm is centered on a structured three-stage output: `<think>` → `<key>` → `<orc>`, with an additional `<note>` section dedicated to laboratory safety precautions. The core idea is to organize outputs in the sequence of reasoning, structuring, and expression, thereby enabling the design of a structured reward mechanism for RL. An overview of this paradigm is provided in Figure 2. Specifically, `<think>`, `<key>`, and `<orc>` denote the reasoning process, the extraction of key information, and the final natural language output, respectively. In `<think>`, the model decomposes sub-goals, identifies sequential dependencies, and justifies the necessity of the proposed experimental steps, ensuring that the protocol is grounded in scientific reasoning. The parsed steps are then organized through a two-phase process of "Sketch" and "Fill." In the "Sketch" phase, represented by `<key>`, the strategies from `<think>` are transformed into a sequence of atomic, machine-readable steps. Each step is constrained to a single JSON dictionary representing one action unit: `{"action": verb, "objects": [...], "parameters": [...]}`. This abstraction reformulates natural language steps into predicate–object–adverbial triplets, bridging free-form instructions with structured sequences. The subsequent "Fill" phase, represented by `<orc>`, expands these steps into fluent natural language instructions, ensuring readability and executability. Formally, `<key>` is defined as $\mathbf{Y}$:

$$\mathbf{Y} = (y_1, \ldots, y_m), \quad y_i = (a_i, \mathcal{O}_i, \mathcal{P}_i), \tag{1}$$

where $a_i$ is the experimental operation, $\mathcal{O}_i$ denotes the objects being acted upon, and $\mathcal{P}_i$ specifies the parameters (e.g., temperature, concentration). Consistency constraints, such as enforcing "One-Action-Per-Step" and maintaining uniform parameter application across objects, are applied to standardize the data format. In the "Fill" phase, represented by `<orc>`, the elements of `<key>` are rendered into human-readable natural language. A strict one-to-one correspondence in step count and semantics is enforced, ensuring no information is added or omitted, with the focus solely on readability. Overall, the "Sketch-and-Fill" paradigm standardizes scientific protocol generation by grounding open-ended language output in an executable structural space. This not only supports stable RL training but also provides a consistent foundation for automatic evaluation.

## 3.3 SCORE MECHANISM

### 3.3.1 OPTIMIZATION OBJECTIVE

In scientific research, experimental protocols are not merely narrative texts but highly structured action guidelines refined through long-term practice. A qualified protocol must describe steps completely while conveying experimental logic: what to do (action), on what objects (objects), and under what conditions (parameters). Traditional text generation metrics (e.g., ROUGE, BLEU,

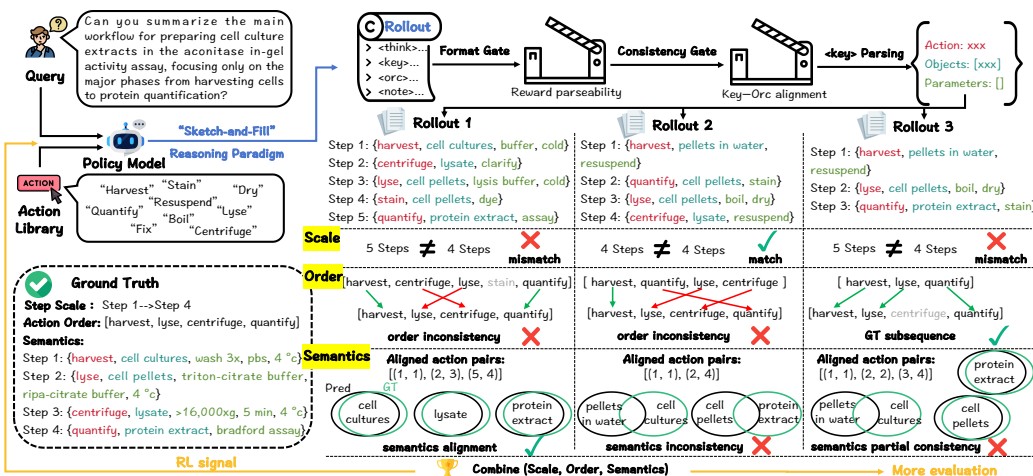

Figure 2: Illustration of the SCORE mechanism and the "Sketch-and-Fill" reasoning paradigm. Three representative rollouts are subjected to a concise analysis across step scale, order consistency, and semantic consistency, while detailed computations are provided in Section 3.3.2.

BERTScore) often overlook such structure, rewarding lexical overlap even when action sequences are disordered. To address this, we propose the SCORE mechanism. Its optimization objective emphasizes both local alignment of protocol content and global logical consistency, defined as:

$$\max_{\theta} \ \mathbb{E}_{(x,\mathcal{A},y^*)\sim\mathcal{D}} \ \mathbb{E}_{y\sim\pi_\theta(\cdot|x,\mathcal{A})} \left[ \sum_{i=1}^{n} \sum_{j=1}^{m} R_{\text{local}}(y_i, \ y_j^*) \otimes R_{\text{global}}(y, \ y^*) \right], \quad (2)$$

which defines $\pi_\theta$ as the policy model with parameters $\theta$, $x$ as the context, $\mathcal{A}$ as the action library, and $y^*$ as the ground-truth protocol. $R_{\text{local}}$ and $R_{\text{global}}$ denote reward functions at different granularities, combined by $\otimes$ (details in Section 3.3.2). The core idea is to shift evaluation from superficial similarity to multidimensional alignment, including logical order and execution precision. Unlike metrics focusing only on text overlap or embeddings, SCORE directly measures the operability of generated protocols, providing scientific, interpretable, experiment-aligned optimization signals.

### 3.3.2 REWARD DESIGN

The SCORE mechanism adopts a progressive design (see Figure 2). First, a gating stage ensures basic structural and consistency requirements. Then, fine-grained rewards for step scale and step semantics are combined into the final reward signal. This approach encourages the model to follow a rational reasoning paradigm and enhances both coherence and executability of the generated protocols. **Format Gate**: The output must contain the four sections `<think>`, `<key>`, `<orc>`, and `<note>` in the correct order. Each step in `<key>` must follow the format `Step x:{json}`, explicitly specifying `action`, `objects`, and `parameters`, ensuring parseability for subsequent reward calculation. **Consistency Gate**: This gate verifies step-by-step correspondence between `<key>` and `<orc>`. Every action, object, and parameter in `<key>` must appear in `<orc>` with at least 95% coverage (see Appendix C.1 for details), ensuring the protocol is a practical guide rather than a hollow framework. Only protocols passing both gates are used for reward computation.

**Step Scale**: The quality of a protocol depends on proper granularity, as too few steps cause omissions while too many lead to redundancy. We design a step scale reward that measures the gap between generated and gold step counts, with one atomic action per step guaranteed by the "Sketch-and-Fill" paradigm. Matching counts yield a score of 1, and deviations are penalized by cosine decay that drops to zero beyond a threshold. A length penalty is also applied, and when the average word length per step exceeds a limit, the score is proportionally reduced to discourage verbosity. Formally:

$$f(d) = \left\{ \begin{array}{ll} \cos\left(\frac{\pi d}{2M}\right), & d < M \\ 0, & d \geq M \end{array} \right\}, \quad g(\bar{L}) = \left\{ \begin{array}{ll} 1, & \bar{L} \leq L \\ \frac{L}{\bar{L}}, & \bar{L} > L \end{array} \right\}, \quad r_{\text{scale}} = \frac{f(|N_{\text{pred}} - N_{\text{gold}}|)}{g(\bar{L})}, \quad (3)$$

where $N_{\text{pred}}$ and $N_{\text{gold}}$ are predicted and gold step counts, and $M = \max(1, \lfloor 0.6 N_{\text{gold}} \rfloor)$ is the deviation threshold. $\bar{L}$ denotes the average text length per step. $f(d)$ penalizes step mismatch via cosine decay, while $g(\bar{L})$ penalizes verbosity. Together, they yield the step scale reward $r_{\text{scale}}$.

**Step Semantics** The step semantics reward is central to SCORE, as it reflects the core execution logic of protocols. It consists of two components: Order Consistency and Semantic Consistency.

**a) Order Consistency**, denoted as $Order(\cdot)$, assesses whether the generated sequence of experimental actions matches the ground truth. We adopt a "Strict" mode (see details in Appendix C.2), rewarding only if the predicted and ground-truth sequences are identical or mutually subsequences, and zero otherwise. This design mirrors laboratory reality since some steps may be repeated or omitted, but a disordered sequence renders the protocol invalid regardless of textual similarity.

**b) Semantic Consistency** is then calculated on the basis of action alignment, since actions function as the anchor of experimental steps (see Appendix C.3). In practice, researchers first decide what operation to perform (e.g., "incubate", "add"), and only then specify the relevant objects and conditions. Without the action anchor, textual similarity cannot guarantee executability. Thus, we align predicted and gold actions step by step. For each aligned pair $(i, j)$, the overlap of object sets is measured using intersection-over-union, with subword-based similarity used as compensation when objects differ but are semantically related. Parameters are compared only if object overlap $\geq 0.5$, reflecting the principle that condition descriptions are meaningless for incorrect objects. Formally:

$$\text{Obj}(i,j) = \frac{|\hat{\mathcal{O}}_i \cap \mathcal{O}_j^*|}{|\hat{\mathcal{O}}_i \cup \mathcal{O}_j^*|}, \quad \text{Par}(i,j) = \begin{cases} 1, & \hat{\mathcal{P}}_i = \varnothing, \mathcal{P}_j^* = \varnothing \\ 0, & \hat{\mathcal{P}}_i = \varnothing \vee \mathcal{P}_j^* = \varnothing \\ \frac{|\mathcal{K}(\hat{\mathcal{P}}_i) \cap \mathcal{K}(\mathcal{P}_j^*)|}{|\mathcal{K}(\hat{\mathcal{P}}_i) \cup \mathcal{K}(\mathcal{P}_j^*)|}, & \text{otherwise} \end{cases}, \quad (4)$$

where $\hat{\mathcal{O}}_i$ and $\hat{\mathcal{P}}_i$ are the predicted object and parameter sets at step $i$, $\mathcal{O}_j^*$ and $\mathcal{P}_j^*$ are the corresponding ground truth sets at step $j$, and $\mathcal{K}(\cdot)$ extracts subword sets for reliable comparison. To better incorporate positional fidelity, we apply a decay factor $m_{ij} = \max\{0, 1 - (|i - j|/D)^\lambda\}$, where $D$ is the number of ground truth steps. This reduces the score when predicted actions, though correct in type, appear far from their reference positions. The final score is:

$$r_{\text{semantics}} = Order(\hat{a}, a^*) + \frac{1}{|\mathcal{W}|} \sum_{(i,j) \in \mathcal{W}} m_{ij} \left( \text{Obj}(i,j) + \tfrac{1}{2} \text{Par}(i,j) \right), \quad (5)$$

where $\mathcal{W}$ is the set of aligned action pairs. Unlike the step scale (equation 6), we adopt an additive combination to avoid over-penalization. This allows partial credit when objects or parameters are imperfectly matched, as long as the action sequence remains reasonable. Such tolerance improves training stability and aligns with how real researchers judge protocols, where minor detail errors may be acceptable if the experimental logic is preserved. Finally, the overall SCORE is defined as:

$$\text{SCORE}(y, y^*) = \mathbb{I}_{\text{format}}(y) \cdot \mathbb{I}_{\text{cons}}(y) \cdot r_{\text{scale}}(y, y^*) \cdot r_{\text{semantics}}(y, y^*), \quad (6)$$

where $\mathbb{I}_{\text{format}}$ and $\mathbb{I}_{\text{cons}}$ are gating functions. This design ensures that only protocols satisfying structural, step scale, and semantic requirements obtain high rewards (Wang et al., 2025), thereby mitigating reward hacking and more faithfully simulating how protocols are evaluated in practice. SCORE also serves as a multidimensional evaluation framework, with details in "Evaluation Metrics" of Section 4.1.

### 3.4 KNOWLEDGE-TO-ACTION LEARNING STRATEGY

Inspired by curriculum learning (Bengio et al., 2009; Wang et al., 2021), we propose a three-stage Knowledge-to-Action Learning framework that progressively enables the transition from textual knowledge to protocol generation. The framework parallels human learning, progressing from knowledge accumulation to standardized operations and finally to exploratory optimization. In the first stage, pre-training, the model learns the semantic structure and operational logic of experimental language from large-scale protocol texts (see Appendix J.2 for further experiments). The second stage, supervised instruction tuning (SFT), is conducted on data following the "Sketch-and-Fill" paradigm, incorporating subtasks such as parameter filling, step ordering, and error correction (Liu et al., 2025). SFT both injects domain knowledge and provides a cold start for RL, aligning outputs with the designated paradigm. The third stage applies RL with the GRPO algorithm (details in Appendix D & J.1.1) (Shao et al., 2024). By removing entropy loss and reducing the KL penalty, we

enhance exploration and avoid premature convergence. Combined with SCORE rewards, this stage improves generalizability and robustness, ensuring more reliable and executable protocol generation.

# 4 EXPERIMENTS AND RESULTS

## 4.1 EXPERIMENT SETTINGS

**Benchmarks & Baselines** To evaluate our model, we adopt two types of benchmarks. The first focuses on real-world protocol generation, represented by SciRecipe-Eval, which emphasizes linguistic quality and practical executability. The second covers broader scientific reasoning and question answering across domains, including Humanity's Last Exam (HLE), LAB-Bench, and PubMedQA (Phan et al., 2025; Laurent et al., 2024; Jin et al., 2019). (Additionally, BioProBench sub-tasks (Liu et al., 2025) are used as supplementary experiments for protocol comprehension.) We further compare against proprietary, open-source, reasoning, and scientific LLMs (see Appendix E for details).

**Implementation Details** We used GPT-5 Chat (OpenAI, 2025) (temperature 0.6) to construct SciRecipe and Gemini 2.5 Flash (Comanici et al., 2025) (temperature 0.2) for validation. In the SCORE mechanism, hyperparameters $L = 30$ and $\lambda = 1.5$ were determined through iterative experiments. For training Thoth, we adopted Qwen3-8B as the base model (Yang et al., 2025). Pretraining and SFT used LoRA fine-tuning via LLaMA-Factory (Zheng et al., 2024), while RL full-parameter tuning was conducted with the VeRL framework (Sheng et al., 2025). All experiments ran on eight Nvidia H100 GPUs. Further hyperparameters and data ratios are given in Appendix F.

## 4.2 EVALUATION METRICS

We designed metrics tailored to different task types. For natural language generation, BLEU, ROUGE, METEOR, and keyword matching (Liu et al., 2025) were used to assess surface-level semantic similarity (Papineni et al., 2002; Lin, 2004; Banerjee & Lavie, 2005; Grootendorst, 2020). Based on the SCORE mechanism, we introduced five executability metrics, namely Step-MATCH (Step-M), Order-LSC/S/Tau, and Semantic-Alignment (Semantic-A), to evaluate step scale, order, and semantic fidelity For multiple-choice, classification, and ranking tasks, we employed Accuracy, F1, Exact Match, and Kendall's Tau (Abdi, 2007).

Although SCORE was originally introduced as a reward mechanism for reinforcement learning, its structured design naturally yields a set of evaluation metrics for protocol generation. Because each step in a protocol is decomposed into an action, an object, and a set of parameters, the resulting metrics allow the assessment of executability and structural fidelity beyond surface-level text similarity.

Formally, let the predicted key-step sequence be $\hat{\mathbf{s}} = (\hat{s}_1, \ldots, \hat{s}_n)$ and the reference sequence be $\mathbf{s}^* = (s_1^*, \ldots, s_m^*)$, with corresponding action sequences $\hat{\boldsymbol{a}}$ and $\boldsymbol{a}^*$. Let $\mathcal{W} = \{(i_k, j_k)\}_{k=1}^K$ denote the aligned action anchors (See Appendix C.3). Based on this structure, we introduce five metrics.

**Step-M** measures step-level completeness through strict length matching:

$$\text{Step-M}(\hat{\mathbf{s}}, \mathbf{s}^*) = \mathbb{I}[\,|\hat{\mathbf{s}}| = |\mathbf{s}^*|\,]. \tag{7}$$

**Order-S** enforces exact sequential agreement of actions:

$$\text{Order-S}(\hat{\boldsymbol{a}}, \boldsymbol{a}^*) = \mathbb{I}[\,\hat{\boldsymbol{a}} = \boldsymbol{a}^*\,]. \tag{8}$$

**Order-LCS** provides a more tolerant measure based on the normalized longest common subsequence:

$$\text{Order-LCS}(\hat{\boldsymbol{a}}, \boldsymbol{a}^*) = \frac{2\,L(\hat{\boldsymbol{a}}, \boldsymbol{a}^*)}{n + m}. \tag{9}$$

**Order-Tau** evaluates the relative order of aligned anchors using a Kendall-style correlation:

$$\text{Order-Tau}(\mathcal{W}) = \frac{C - D'}{C + D'}, \tag{10}$$

where $C$ and $D'$ denote concordant and discordant anchor pairs, defaulting to 0 when $C + D' = 0$.

Finally, **Semantic-A** corresponds to the semantic consistency score described in Section 3.3.2. It evaluates object correctness, parameter fidelity, and positional consistency over the aligned anchors, providing a structured measure of semantic accuracy and executability.

## 4.3 MAIN RESULTS

Table 1: Main results on SciRecipe-Eval. Metrics left of the dashed line evaluate executability, those on the right measure lexical similarity. **Bold** denotes the best score (see Appendix I.2 for details).

| Methods | Semantic-A | Order-LCS | Order-S | Order-Tau | Step-M | BLUE-AVG | ROUGE-L | METEOR | KW-F1 | AVG |
|---|---|---|---|---|---|---|---|---|---|---|
| *Close-Source SOTA* | | | | | | | | | | |
| ChatGPT-4o | 40.04 | 73.27 | 24.00 | 70.33 | 44.00 | 38.95 | 48.42 | 44.66 | 52.05 | 48.41 |
| GPT-5 | 27.79 | 58.12 | 11.35 | 53.55 | 18.79 | 21.31 | 32.96 | 32.55 | 39.17 | 32.84 |
| GPT-5 Chat | 36.30 | 73.21 | 21.17 | 65.67 | 25.00 | 29.57 | 42.04 | 41.95 | 47.87 | 42.53 |
| Claude Sonnet 4 | 39.35 | 71.97 | 20.83 | 70.00 | 35.83 | 34.24 | 44.27 | 40.97 | 49.40 | 45.21 |
| Claude Opus 4.1 | 41.32 | 71.70 | 21.80 | 71.93 | 34.59 | 34.69 | 44.42 | 40.36 | 50.00 | 45.65 |
| Gemini 2.5 Flash | 36.35 | 70.61 | 20.00 | 70.33 | 32.33 | 33.19 | 42.91 | 39.26 | 48.07 | 43.67 |
| Gemini 2.5 Pro | 35.80 | 72.68 | 21.83 | 70.17 | 32.00 | 31.37 | 44.16 | 45.59 | 48.58 | 44.69 |
| Doubao-1.5-pro | 33.33 | 73.29 | 23.67 | 70.00 | 47.50 | 38.16 | 46.88 | 38.71 | 48.74 | 46.70 |
| Qwen2.5-Max | 40.34 | 72.88 | 21.83 | 71.33 | 47.50 | 30.81 | 48.02 | 43.82 | 51.98 | 47.61 |
| *Open-Source SOTA* | | | | | | | | | | |
| Qwen2.5-72B-Instruct | 36.40 | 70.82 | 21.00 | 69.17 | 42.17 | 29.55 | 46.06 | 43.44 | 49.73 | 45.37 |
| Qwen3-235B-A22B-Instruct | 35.68 | 72.07 | 20.03 | 69.12 | 37.73 | 32.48 | 44.37 | 44.30 | 47.89 | 44.85 |
| DeepSeek-V3 | 41.72 | 73.97 | 21.44 | 70.54 | 41.71 | 38.18 | 48.49 | 45.08 | 52.33 | 48.16 |
| GPT-OSS-120B | 32.86 | 69.97 | 17.67 | 64.17 | 27.83 | 30.72 | 43.44 | 42.90 | 49.92 | 42.16 |
| Llama-3.1-405B-Instruct | 35.92 | 69.46 | 18.03 | 67.78 | 39.23 | 36.42 | 44.86 | 42.17 | 48.09 | 44.66 |
| Kimi-K2-Instruction | 36.99 | 71.83 | 20.83 | 69.83 | 40.00 | 33.81 | 44.49 | 42.99 | 49.00 | 45.53 |
| *Reasoning Models* | | | | | | | | | | |
| DeepSeek-R1 | 36.07 | 71.38 | 20.37 | 69.12 | 32.89 | 39.80 | 45.86 | 38.19 | 49.83 | 44.83 |
| Grok 3 | 37.40 | 73.27 | 21.92 | 69.73 | 39.25 | 34.72 | 46.04 | 46.21 | 48.59 | 46.35 |
| Grok 4 | 36.73 | 72.08 | 20.25 | 65.34 | 34.66 | 37.21 | 46.17 | 40.18 | 51.81 | 44.94 |
| OpenAI-o1 | 34.74 | 73.40 | 18.53 | 67.45 | 35.39 | 35.68 | 46.29 | 43.82 | 50.13 | 45.05 |
| OpenAI-o3 | 35.40 | 70.38 | 15.38 | 65.05 | 24.08 | 28.62 | 43.08 | 44.33 | 50.26 | 41.84 |
| Qwen3-4B | 24.37 | 53.55 | 13.67 | 50.50 | 28.83 | 14.52 | 24.74 | 23.95 | 27.69 | 29.09 |
| **Thoth-mini** | 44.28 | 74.68 | 25.33 | 70.83 | 52.67 | 43.32 | 49.23 | 46.41 | 53.13 | 51.10 |
| Qwen3-8B | 28.89 | 63.51 | 11.17 | 58.67 | 24.33 | 16.66 | 32.31 | 34.72 | 38.63 | 34.32 |
| **Thoth** | 46.60 | 75.34 | 25.50 | 73.33 | 53.00 | 43.62 | 50.02 | 47.39 | 54.13 | 52.10 |

**Protocol Generation** We evaluated protocol generation across two dimensions: executability, which emphasizes consistency in objects and step order, and lexical similarity, which reflects surface-level matching. As shown in Table 1, Thoth achieves SOTA results across all metrics. Compared to baselines, Thoth and Thoth-mini improve average performance by 17.78% and 22.01%, underscoring the effectiveness of our approach. Thoth also surpasses much larger proprietary models, outperforming ChatGPT-4o by 3.69% on average. Against the strongest open-source model, DeepSeek-V3, Thoth achieves gains of 4.88%, 4.06%, and 11.29% on Semantic-Alignment, Order-S, and Step-MATCH, respectively, showing clear advantages in step alignment, logical order, and action fidelity. General-purpose SOTA

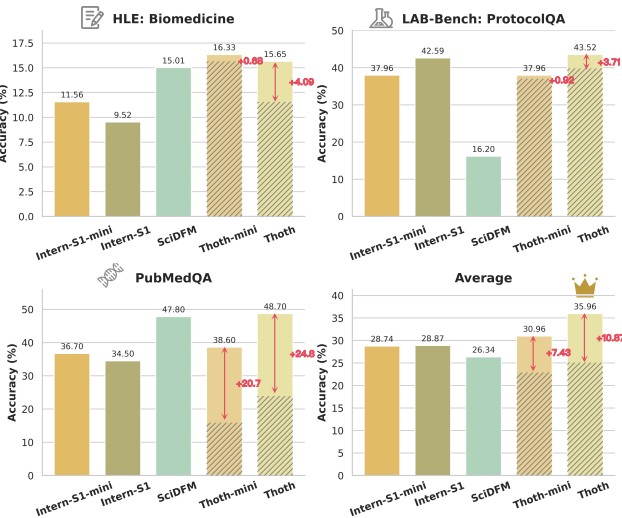

Figure 3: Results on other scientific benchmarks. Slash shading denotes baseline of Thoth models.

models show notable capability, likely due to large-scale pre-training, but mainly achieve superficial similarity and lack real-world executability. Reasoning-oriented models also perform poorly, often producing overly complex outputs unsuited for laboratory workflows (see Appendix I.1 for qualitative cases). Overall, Thoth demonstrates consistent advantages across metrics, bridging knowledge-based text generation with executable protocols.

**Fundamental Scientific Tasks** Beyond protocol generation, we evaluated Thoth on three out-of-domain biomedical benchmarks against recent scientific LLMs (Intern-S1 and SciDFM), with results shown in Figure 3. Thoth consistently outperforms models fine-tuned on biomedical corpora, exceeding Intern-S1 and Intern-S1-mini by 7.09% and 2.22% on average, and notably both Thoth and Intern-S1-mini share Qwen3-8B as their base model. As illustrated in Figure 3, Thoth also

shows clear improvements over its baseline models (slash shading), achieving an average improvement of 10.87%, while Thoth-mini improves by 7.43%. The performance gap between Thoth and Thoth-mini further confirms the benefits of our approach. These results demonstrate that knowledge and reasoning skills acquired from scientific protocols generalize effectively to broader biomedical tasks. Additional experiments on protocol comprehension are provided in Appendix I.3.

## 4.4 ANALYSES

**SciRecipe** To assess the impact of different data components, we conducted ablation studies on SciRecipe (Table 2). Models trained only on QA or generation QA showed weak results, with BLEU-AVG below 40%. Adding Protocol-Comprehension tasks markedly improved executability, with Order-LCS reaching 74.50%. Joint training on both task types further enhanced alignment, with Step-M rising to 52.33%. The best overall performance was obtained when combining QA with scientific review, achieving 46.60% on Semantic-A and 49.45% on AVG. These findings highlight that diverse task coverage and systematic quality checks enhance the reliability of protocol generation.

Table 2: Ablation studies on training set. ♣: trained only on SciRecipe Protocol-Comprehension tasks; ♠: trained on both task types. QA and SciCheck denote inclusion of basic protocol QA and scientific review of SciRecipe, respectively. ⋆: trained only on protocol generation QA.

| SciRecipe | QA | SciCheck | Semantic-A | Order-LCS | Order-S | Step-M | BLUE-AVG | ROUGE-L | METEOR | KW-F1 | AVG |
|---|---|---|---|---|---|---|---|---|---|---|---|
| | ✓ | — | — | — | — | — | 23.88 | 31.50 | 29.53 | 34.50 | 29.85 |
| | ⋆ | — | — | — | — | — | 23.71 | 32.08 | 27.92 | 37.78 | 30.37 |
| ♣ | | ✓ | 42.34 | 74.50 | 24.33 | 50.50 | 41.33 | 47.34 | 44.81 | 51.72 | 47.11 |
| ♣ | ✓ | ✓ | 44.19 | 74.85 | 23.83 | 43.67 | 42.42 | 47.59 | 45.44 | 52.29 | 46.79 |
| ♠ | | ✓ | 44.51 | 74.67 | 25.00 | 52.33 | 42.74 | 48.53 | 46.84 | 52.30 | 48.37 |
| ♠ | ✓ | | 44.54 | 75.14 | 23.50 | 47.00 | 41.79 | 48.09 | 47.01 | 51.89 | 47.37 |
| ♠ | ✓ | ✓ | 46.60 | 75.34 | 25.50 | 53.00 | 43.62 | 50.02 | 47.36 | 54.13 | 49.45 |

**Reasoning Paradigm** To align model outputs with experimental logic and generate evaluable protocols, we propose the "Sketch-and-Fill" reasoning paradigm. Its effect was systematically tested through ablation experiments on four representative models: DeepSeek-V3, GPT-5 Chat, Thoth, and its base model Qwen3-8B. Since outputs are in natural language, executability metrics were not applicable, and Figure 4 reports results based on lexical similarity. The results show that "Sketch-and-Fill" improves performance for all models except the base model, with notable gains on BLEU-AVG. DeepSeek-V3, GPT-5 Chat, and Thoth achieve average improvements of 3.89%, 2.92%, and 3.79% over counterparts without the paradigm. For the base model, the lack of improvement is attributed to reasoning failure due to missing knowledge injection from scientific protocols.

Table 3: Ablation studies on the SCORE mechanism. Evaluation covers step scale, semantic alignment, and overall reward design. $KL(\cdot)$ denotes the KL-divergence penalty in the loss function, and "Vanilla" refers to using standard semantic similarity metrics as the reward signal.

| SCORE | | Semantic-A | Order-LCS | Order-S | Order-Tau | Step-M | BLUE-AVG | ROUGE-L | METEOR | KW-F1 | AVG |
|---|---|---|---|---|---|---|---|---|---|---|---|
| **Step Scale** | w/o $f(d)$ | 43.67 | 55.97 | 6.83 | 51.83 | 10.00 | 43.09 | 37.28 | 26.08 | 43.31 | 35.34 |
| | w/o $g(\bar{L})$ | 38.89 | 74.37 | 23.67 | 71.00 | 51.00 | 43.18 | 46.14 | 44.92 | 52.80 | 49.55 |
| | w/o $f(d)$+$g(\bar{L})$ | 45.35 | 49.50 | 3.00 | 34.17 | 4.17 | 42.49 | 33.44 | 21.12 | 40.41 | 30.41 |
| **Semantic Alignment** | w/o $m_{ij}$ | 38.68 | 73.70 | 23.17 | 67.67 | 50.17 | 42.84 | 48.91 | 45.58 | 52.85 | 49.29 |
| | w/o $Order(\cdot)$ | 40.93 | 61.27 | 12.83 | 48.00 | 33.33 | 37.26 | 45.35 | 43.27 | 49.46 | 41.30 |
| | w/o $Order(\cdot)$ +$f(d)$+$g(\bar{L})$ | 44.15 | 58.22 | 7.67 | 42.83 | 43.33 | 38.45 | 45.78 | 43.52 | 48.74 | 41.41 |
| **Reward** | w/o $KL(\cdot)$ | 39.96 | 74.43 | 22.67 | 69.33 | 46.67 | 42.73 | 46.03 | 44.82 | 53.36 | 48.89 |
| | Vanilla | 38.74 | 63.41 | 21.50 | 52.36 | 44.50 | 45.52 | 50.12 | 43.90 | 50.54 | 45.62 |
| **Thoth** | | 46.60 | 75.34 | 25.50 | 73.33 | 53.00 | 43.62 | 50.02 | 47.39 | 54.13 | 52.10 |

**SCORE Mechanism** In Section 3.3.2, we introduced the SCORE reward mechanism, consisting of step scale, order consistency, and semantic consistency. To evaluate its role in RL training, we conducted ablation studies, including variants that removed individual components, dropped the KL divergence penalty, or replaced SCORE with a vanilla reward based only on BLEU-AVG, ROUGE-L, and BERTScore. Results are summarized in Table 3. Omitting the step scale reward leads to sharp declines in Order-S and Step-M (6.83%/10% and 3%/4.17%, rows 1 and 3), indicating failures

to generate executable protocols and producing either verbose or incomplete outputs. Excluding the positional decay factor (row 4) weakens penalties for misaligned steps and reduces semantic consistency, under which Thoth still achieves a 7.92% gain in Semantic-A. The order consistency reward is also critical because without enforcing execution order, steps become misarranged and semantic coherence breaks down. Thoth surpasses the variants in rows 5 and 6 by 6.36% and 5.17% on BLEU-AVG, respectively. Removing the KL penalty further degrades performance, with Table 3 showing a 3.21% average drop compared to Thoth. Although the vanilla reward improves BLEU-AVG and ROUGE-L, the generated protocols exhibit poor executability with a 10.65% average reduction. These results demonstrate the effectiveness of SCORE in guiding the model to generate coherent and executable protocols. For further analyses of reward, see Appendix J.1.2 & J.1.3.

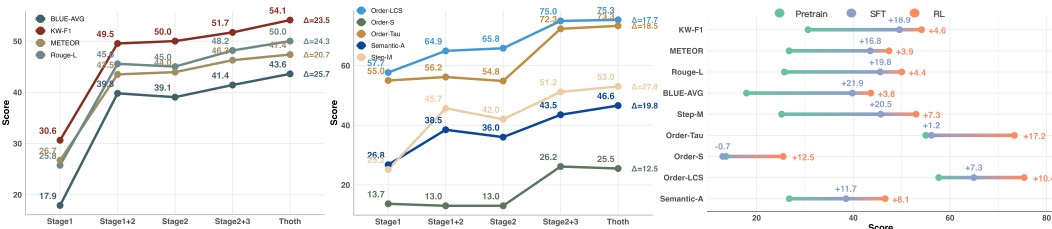

Figure 5: Ablation studies on the training strategy. Left and middle: rightmost $\Delta$ marks extreme values across strategies. Right: Thoth's training trajectory across stages.

**Training Strategy** To progressively equip the model for solving complex scientific problems, we propose the three-stage Knowledge-to-Action Learning strategy. We compared models trained at different stages with Thoth to validate its effectiveness. As shown in Figure 5 (left and middle), Thoth achieves SOTA performance after staged training and consistently outperforms models trained with only Stage 1+2 (pre-training and SFT) or without pre-training (Stage 2+3 or Stage 2 alone). Specifically, Thoth improves by an average of 11.08% on executability and 4.2% on lexical similarity over Stage 1+2. It further achieves mean gains of 1.44% and 8.79% over Stage 2+3 and Stage 2 across both dimensions. Figure 5 (right) further demonstrates that the three-stage process progressively enhances performance, with SFT improving lexical similarity and RL strengthening rationality and executability.

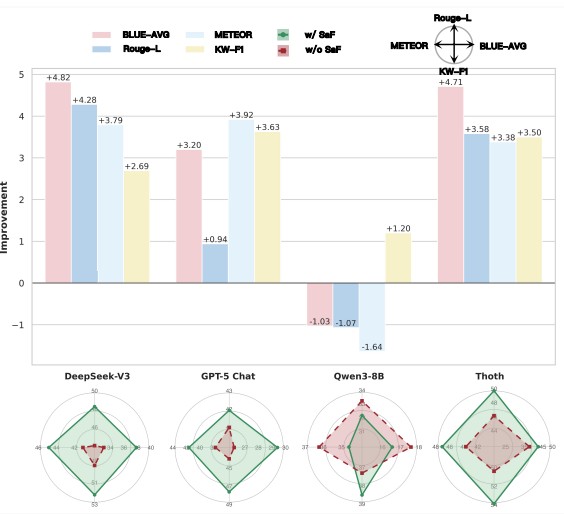

Figure 4: Ablation studies on reasoning paradigm. Performance differences with/without Sketch-and-Fill (SaF) are shown in the top panel, and a visual comparison is provided in the bottom panel.

## 5 CONCLUSION

In this paper, we introduced Thoth, a protocol-generation model grounded in the SciRecipe dataset, the "Sketch-and-Fill" reasoning paradigm, and the SCORE mechanism. By integrating structured reasoning with verifiable rewards, Thoth achieves SOTA performance on both protocol-specific and broader scientific benchmarks. Equally important, the protocols it generates are concise, logically coherent, and experimentally executable. This work provides a blueprint for building reliable scientific assistants and demonstrates that structured reasoning frameworks combined with targeted rewards are critical for advancing protocol generation and improving reproducibility.

ETHICS STATEMENT

This work does not involve human subjects, animal experiments, or personally identifiable data. All datasets used in this paper will be publicly released upon publication to ensure transparency and reproducibility. We have carefully considered potential ethical issues related to data privacy, fairness, and misuse. We believe our findings pose no foreseeable risks of harm, and all contributions were made in compliance with the ICLR Code of Ethics.

REPRODUCIBILITY STATEMENT

We have made extensive efforts to ensure the reproducibility of our results. Details on training data distributions, hyperparameter settings for pre-training, SFT, and RL, as well as implementation frameworks, are provided in Appendix F. We will also release all code, models, and processed datasets to facilitate faithful reproduction of our findings.

ACKNOWLEDGEMENT

This work was supported by Shanghai Artificial Intelligence Laboratory and Intern Discovery.

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

APPENDIX CONTENTS

# A  SciRecipe

## A.1  Preprocessing of Original Scientific Protocols

In the preprocessing phase, we first employed MinerU (Wang et al., 2024) to identify and extract protocol texts, as the majority of online experimental protocols are distributed in non-editable PDF format. This step was essential because protocols published in PDF form often contain heterogeneous content such as figures, tables, references, and lengthy narratives, which substantially increase the difficulty of accurate parsing and subsequent question–answer pair construction. To mitigate this issue, we analyzed both the page count and the number of figures/tables in each document, and applied empirically determined thresholds to filter out overly long or structurally complex protocols. Another important consideration was the removal of redundancy, since many protocols share highly similar structures or represent slightly modified versions of the same experiment. Without careful filtering, these near-duplicate entries could cause information leakage across training and evaluation splits. To address this, we applied similarity detection and clustering. Specifically, we used Qwen3-Embedding-8B (Zhang et al., 2025b) to vectorize protocol titles, built an efficient similarity search index using FAISS (Douze et al., 2024), and conducted Top-K nearest-neighbor searches. Similar pairs were identified by thresholding cosine similarity scores, after which a union–find algorithm was applied to partition protocols into connected components. This process produced both pairwise and grouped similarity sets, effectively eliminating redundant samples while preserving data diversity and representativeness. In addition, some protocols were accompanied by electronic supplementary materials, which frequently contained fragmented or ambiguous instructions. Because these supplementary files risked introducing inconsistencies and annotation difficulties, they were uniformly removed from the dataset. To further guarantee the reliability of the corpus, we recruited three doctoral students with extensive experimental experience to manually review a subset of the data. Their verification ensured not only correctness but also balanced coverage across experimental domains, ultimately enhancing both the diversity and representativeness of the retained corpus.

## A.2  Structured Integration of Scientific Protocols

Because experimental protocols vary widely in writing style and formatting, we extracted common elements to form a unified structured representation. The core fields included experiment name, research background, reagents and materials, experimental equipment, and experimental procedures. This integration was carried out in two complementary steps. First, we applied a rule-based method that relied on regular expressions to match predefined keywords and extract the corresponding fields. During this stage, Markdown tables were removed to avoid parsing errors, while figure and table captions were preserved as valuable contextual information. The rule-based approach achieved reliable performance for protocols written in a highly standardized format, but its applicability was limited for documents with more complex or irregular structures. To address these limitations, we further adopted a model-based strategy. Specifically, we leveraged Grok-4 (xAI, 2025) to summarize and restructure the original protocols, ensuring accurate extraction of key fields without introducing fabricated content. The model also adjusted the ordering of experimental steps to maintain logical and procedural consistency. This hybrid two-step pipeline allowed us to combine the precision of rule-based parsing with the flexibility of model-based restructuring. Through this process, we obtained over 12K structured protocols. Quality control measures were subsequently applied to filter out entries with missing fields, excessively lengthy procedures, or dependencies on external online resources. The resulting corpus provides a clean, logically consistent, and representative collection of protocols suitable for downstream training and evaluation.

## A.3  Supplementary Details on SciRecipe

When designing the eight task types in SciRecipe, we defined explicit objectives and representative scenarios for each. Together, these tasks cover the entire lifecycle of experimental research, ensuring that the dataset trains and evaluates both general reasoning and practical execution.

- **Overview**: Requires the model to generate a hierarchically organized summary of the protocol, enabling rapid comprehension of its global structure.
- **Specific**: Focuses on step-by-step decomposition of procedures, ensuring the model captures the micro-level operational logic.

- **Retrieval**: Targets precise extraction of experimental parameters (e.g., temperature, pH, concentration), emphasizing accuracy in reporting.

- **Planning**: Involves transforming high-level objectives into coherent, logically ordered experimental steps.

- **Troubleshooting**: Simulates diagnosing experimental failures, identifying error sources, and proposing corrective strategies.

- **Constraint**: Tests adaptability under resource limitations, such as restricted equipment or reagent availability.

- **Scaling**: Requires numerical adjustments and unit conversions to adapt experiments across different scales.

- **Safety**: Focuses on compliance with safety standards and identification of potential laboratory risks.

By encompassing comprehension, planning, error handling, adaptability, scalability, and safety, SciRecipe ensures a comprehensive and targeted task design. See Figure 21 for detailed prompts.

## A.4  SUPPLEMENTARY DETAILS ON SCIRECIPE CONSTRUCTION PIPELINE

During the construction of SciRecipe, we first generated QA pairs for multiple task types using the structured protocol corpus as the foundation. Particularly, for Protocol-Comprehension tasks, we relied on hierarchical tree extraction (e.g., "Step 1: xxx; Step 1.1: xxx; Step 1.1.1: xxx"), which preserved both the hierarchical structure and the stepwise dependencies of experimental procedures. This design ensured that the model could capture not only surface-level descriptions but also the nested logic inherent in complex experiments. Then, we introduced the "Sketch-and-Fill" reasoning paradigm. This approach first summarizes core elements in an outline-like format and then fills in the specific operations, thereby enhancing both interpretability and training effectiveness. The paradigm was particularly effective in decomposing high-level objectives into concrete experimental actions while maintaining logical consistency. Because LLMs may produce outputs with randomness and formatting inconsistencies, directly discarding such samples would risk losing high-quality data. To address this, we designed a format validation and repair module. This module automatically detected non-compliant outputs and prompted the model to regenerate them under low-temperature settings, ensuring structural consistency while retaining valid content.

After passing the format checks, each QA pair underwent content review across six scientific dimensions:

1. **Scientific Accuracy**: correctness of operations, parameters, and outcomes.

2. **Safety & Compliance**: adherence to laboratory safety standards and ethical guidelines.

3. **Logical Coherence & Actionability**: stepwise consistency and rational flow of procedures.

4. **Clarity & Ambiguity**: precision and unambiguity of textual descriptions.

5. **Generality & Specificity**: balanced coverage of broad applicability and detailed context.

6. **Efficiency & Resource Optimization**: optimization of time, materials, and experimental resources.

For this step, we employed Gemini 2.5 Flash (Comanici et al., 2025) as the validation model, which performed dimension-wise evaluations and produced review reports. QA pairs that failed to meet requirements were systematically discarded. Finally, to guarantee the reliability of the constructed dataset, we performed manual verification. Doctoral students from interdisciplinary backgrounds conducted thorough inspections, ensuring that the resulting data were not only structurally correct but also scientifically feasible and practically reasonable. Through this multi-stage pipeline, which combines automated validation, targeted repair, model-assisted review, and expert-level manual checks, SciRecipe achieved both rigor and diversity, providing a robust foundation for protocol-oriented training and evaluation.

Table 4: Disciplinary distribution of the SciRecipe dataset.

| Subdomain | Count | Percentage |
|---|---|---|
| Cell Biology | 2366 | 19.82% |
| Biochemistry | 1935 | 16.21% |
| Molecular Biology | 1788 | 14.98% |
| Microbiology | 1163 | 9.74% |
| Plant Science | 936 | 7.84% |
| Immunology | 717 | 6.01% |
| Neuroscience | 688 | 5.77% |
| General Laboratory Procedure | 521 | 4.37% |
| Bioinformatics | 319 | 2.67% |
| Bioimaging Technologies | 284 | 2.38% |
| Cancer Biology | 275 | 2.30% |
| Model Organism-Specific Techniques | 246 | 2.06% |
| Others | 127 | 1.06% |
| Genomics Technologies | 111 | 0.93% |
| Structural Biology Techniques | 79 | 0.66% |
| Biophysics | 77 | 0.65% |
| Pharmacology & Drug Development | 67 | 0.56% |
| Developmental Biology | 64 | 0.54% |
| Genetics | 55 | 0.46% |
| Histology Techniques | 31 | 0.26% |
| Synthetic Biology & Bioengineering | 31 | 0.26% |
| Stem Cells | 18 | 0.15% |
| Bioengineering | 11 | 0.09% |
| Toxicology & Safety Testing | 11 | 0.09% |
| Systems Biology | 8 | 0.07% |
| Medicine | 4 | 0.03% |
| Drug Discovery | 3 | 0.03% |

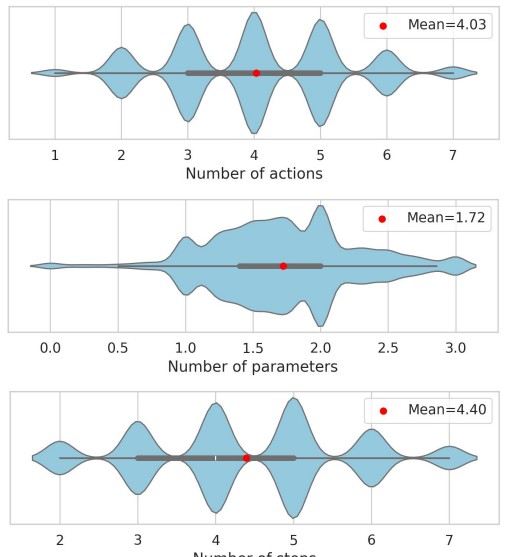

Figure 6: Data distribution of experimental actions, parameters (each step), and step counts in SciRecipe after quality control.

## B  SCIRECIPE-EVAL

Although many benchmarks address scientific question answering, resources that directly evaluate experimental protocol generation are still very limited. Existing evaluations often require models to restate or paraphrase the original protocol, which does not reflect the ability to generate new and executable procedural content in realistic laboratory contexts. To address this gap, we construct SciRecipe-Eval as a benchmark designed to assess protocol generation under practical usage scenarios. Following the design principles of SciRecipe, we independently curate and process a set of 400 protocols. The benchmark is organized into two main categories, and each category contains approximately 300 samples with a balanced distribution across subtasks.

While the names of these categories may resemble comprehension-oriented tasks, all subtasks require the model to generate structured protocol content based on the user query. The Protocol Comprehension tasks evaluate whether the model can understand the user's goal and produce procedural steps that align with standard experimental workflows. The Problem Solving tasks focus on situations in which experimenters encounter practical issues that are not explicitly addressed in the original protocol, and the model must generate corrective or supplementary steps. In this sense, **SciRecipe-Eval serves as a protocol generation benchmark** rather than a traditional question answering benchmark. The understanding and reasoning involved in these tasks are integral components of the protocol generation process.

- **Protocol-Comprehension**:
  - *Overview*: generating hierarchical summaries of experimental procedures.
  - *Specific*: decomposing steps to capture micro-level logic.
- **Problem-Solving**:
  - *Retrieval*: extracting precise parameters such as temperature, pH, or concentration.
  - *Planning*: transforming objectives into coherent and logically ordered plans.
  - *Troubleshooting*: diagnosing potential causes of experimental failure.
  - *Constraint*: adapting protocols under resource limitations.
  - *Scaling*: performing numerical adjustments and conversions for different scales.
  - *Safety*: ensuring compliance and risk control.

Because the QAs were designed under specific reasoning paradigms, experimental actions (predicates) were treated as the core elements of the ground truth. The number of actions defines the size of the action space and thereby provides a direct measure of experimental complexity. Based on this criterion, tasks containing fewer than four actions were categorized as **Level 1**, whereas those with four or more actions were categorized as **Level 2**. For Level 2 tasks, we further inserted one to four randomly sampled distractor actions, enabling more comprehensive evaluation of a model's ability to discriminate between correct and incorrect operations. In line with the data quality control standards established for SciRecipe, we recruited three doctoral students in biology-related fields to perform manual sampling and review of the dataset. Their verification ensured that the final benchmark was not only structurally consistent but also scientifically accurate and representative of real laboratory practices.

## C  SCORE MECHANISM

### C.1  CONSISTENCY GATE

In protocol evaluation, **consistency** refers to the step-by-step correspondence between the structured outline (`<key>`) and its natural language expansion (`<orc>`). If an action, object, or parameter declared in `<key>` is missing from the corresponding `<orc>` description, the generated protocol is semantically incomplete and cannot serve as a reliable experimental guide. Therefore, the consistency gate is adopted as the second checkpoint in SCORE, and only protocols passing this constraint are eligible for reward computation.

Concretely, we first require that the number of steps in `<key>` matches exactly with `<orc>`, with consecutive numbering covering 1 through $N$. For each step $i$, we construct a token set $T_i$ by

concatenating the declared action $a_i$, object set $\mathcal{O}_i$, and parameter set $\mathcal{P}_i$. After normalization (e.g., case folding, unit standardization, removal of subscript/superscript variants), we compute the coverage rate of these tokens in the natural language step $\texttt{<orc>}_i$ as:

$$\mathrm{cov}_i = \frac{\left|\{t \in T_i : t \subseteq \mathrm{norm}(\texttt{<orc>}_i)\}\right|}{|T_i|} \tag{11}$$

If all steps satisfy $\mathrm{cov}_i \geq \tau$ (we set $\tau = 0.95$ in this work) and the step indices are strictly aligned, the consistency gate is considered passed:

$$\mathbb{I}_{\mathrm{cons}}(y) = \begin{cases} 1, & \hat{N} = N \ \wedge \ \{\hat{n}_1, \ldots, \hat{n}_{\hat{N}}\} = \{1, \ldots, N\} \ \wedge \ \min_i \mathrm{cov}_i \geq \tau \\ 0, & \text{otherwise} \end{cases} \tag{12}$$

where $\hat{N}$ and $N$ denote the step counts of the generated and reference protocols, respectively, and $\hat{n}_i$ denotes the predicted step indices. If the gate fails, all subsequent rewards are set to zero. This hard gating design reflects a fundamental experimental principle: when the structured outline and its natural language realization are inconsistent, even partially correct fragments cannot make the protocol executable.

## C.2 ORDER CONSISTENCY

Let $\hat{\boldsymbol{a}} = (\hat{a}_1, \ldots, \hat{a}_n)$ denote the predicted action sequence extracted from $\texttt{<key>}$, and $\boldsymbol{a}^* = (a_1^*, \ldots, a_m^*)$ the reference sequence. The goal of order consistency is to evaluate whether the predicted execution order preserves the logical progression of the reference protocol, since deviations in ordering can undermine reproducibility even when individual steps appear correct. Two complementary scoring modes are provided.

The first mode, Strict Subsequence, enforces a conservative executability constraint. The score equals 1 if the predicted sequence is identical to the reference, or if one is a subsequence of the other; otherwise it is 0:

$$\mathrm{Order}_{\mathrm{strict}}(\hat{\boldsymbol{a}}, \boldsymbol{a}^*) = \begin{cases} 1, & \hat{\boldsymbol{a}} = \boldsymbol{a}^* \ \vee \ \mathrm{subseq}(\hat{\boldsymbol{a}}, \boldsymbol{a}^*) \ \vee \ \mathrm{subseq}(\boldsymbol{a}^*, \hat{\boldsymbol{a}}) \\ 0, & \text{otherwise} \end{cases} \tag{13}$$

where $\mathrm{subseq}(\mathbf{x}, \mathbf{y})$ holds when $\mathbf{x}$ can be embedded in $\mathbf{y}$ with order preserved. This criterion accepts insertions or omissions as long as the preserved actions remain in order, but strictly rejects any reordering. It is also computationally efficient, requiring only $O(n + m)$ time.

The second mode, Longest Common Subsequence (LCS), provides a graded similarity that reflects how much of the reference order is retained. The score is defined as:

$$\mathrm{Order}_{\mathrm{LCS}}(\hat{\boldsymbol{a}}, \boldsymbol{a}^*) = \frac{\mathrm{LCS}(\hat{\boldsymbol{a}}, \boldsymbol{a}^*)}{m} \tag{14}$$

where $m = |\boldsymbol{a}^*|$ is the length of the reference sequence and $\mathrm{LCS}(\cdot, \cdot)$ is the length of the longest common subsequence between the predicted and reference sequences. This normalization yields a value in $[0, 1]$ that represents the fraction of the reference order preserved. Unlike the strict mode, LCS assigns partial credit when the prediction maintains some correct subsequences even if other steps are misplaced. For example, given the ground truth [harvest, lyse, centrifuge, quantify], a prediction [harvest, lyse, quantify] achieves a strict score of 1 and an LCS score of $3/4 = 0.75$ (omission but order preserved), whereas [harvest, centrifuge, lyse, quantify] scores 0 under the strict rule but $3/4 = 0.75$ under LCS, since most of the order is still preserved. By contrast, [lyse, harvest, quantify, centrifuge] is more severely misordered, yielding an LCS score of $2/4 = 0.5$.

For completeness, the LCS length is computed via dynamic programming. Let $L(i, j)$ denote the LCS length between the prefix subsequences $(\hat{a}_1, \ldots, \hat{a}_i)$ of the predicted sequence $\hat{\boldsymbol{a}}$ and $(a_1^*, \ldots, a_j^*)$ of the reference sequence $\boldsymbol{a}^*$. Then

$$L(i, j) = \begin{cases} 0, & i = 0 \ \text{or} \ j = 0 \\ L(i-1, j-1) + 1, & \hat{a}_i = a_j^* \\ \max\{L(i-1, j), L(i, j-1)\}, & \hat{a}_i \neq a_j^* \end{cases} \tag{15}$$

with $\mathrm{LCS}(\hat{\boldsymbol{a}}, \boldsymbol{a}^*) = L(n, m)$. The algorithm runs in $O(nm)$ time and requires $O(nm)$ space.

Together, these two modes serve distinct but complementary purposes, with the strict subsequence mode being the primary criterion. Strict alignment best reflects the hard requirements of laboratory executability, where even minor order violations typically render a protocol unusable. The LCS mode is introduced only as a supplementary measure, offering smoother and more informative feedback during training by granting partial credit when some procedural intent is preserved. In practice, strict mode is preferred for evaluation of executable protocols, while LCS can be selectively applied to provide graded optimization signals when softer supervision is required. All formal experimental results in this work are reported under the strict mode. See Appendix J.1.2 for further experiments.

### C.3 ACTION ANCHOR

In semantic consistency evaluation, actions (i.e., step verbs) are treated as anchors to align predicted steps with reference steps before any further comparison. The alignment is performed in a left-to-right, order-preserving manner. Formally, let the predicted action sequence be $\hat{\boldsymbol{a}} = (\hat{a}_1, \ldots, \hat{a}_n)$ and the reference action sequence be $\boldsymbol{a}^* = (a_1^*, \ldots, a_m^*)$. For each predicted action $\hat{a}_i$, we select the earliest reference index $j$ *after* the previous match such that $a_j^* = \hat{a}_i$. If such a $j$ exists, we record the pair $(i, j)$, otherwise $\hat{a}_i$ remains unmatched and contributes no anchor. This procedure yields an ordered set of index pairs $\mathcal{W} = \{(i_1, j_1), \ldots, (i_K, j_K)\}$ with $1 \leq i_1 < \cdots < i_K \leq n$, $1 \leq j_1 < \cdots < j_K \leq m$, and $\hat{a}_{i_k} = a_{j_k}^*$ for all $(i_k, j_k) \in \mathcal{W}$. The construction is greedy and monotone (first-come–first-match), which guarantees that (i) sequence order is preserved, (ii) repeated actions are matched consistently, with each predicted occurrence aligning only to the next unused reference occurrence, and (iii) the overall alignment runs in linear time $O(n + m)$.

**Example** For the ground truth sequence [*harvest*, *lyse*, *centrifuge*, *quantify*] and the prediction [*harvest*, *centrifuge*, *lyse*, *stain*, *quantify*], the alignment proceeds as

$$\begin{aligned}
\hat{a}_1 &= \textit{harvest} && \mapsto a_1^* = \textit{harvest}, \\
\hat{a}_2 &= \textit{centrifuge} && \mapsto a_3^* = \textit{centrifuge}, \\
\hat{a}_3 &= \textit{lyse} && \mapsto a_2^* = \textit{lyse} \ \ (\text{rejected: order violation}), \\
\hat{a}_5 &= \textit{quantify} && \mapsto a_4^* = \textit{quantify}.
\end{aligned}$$

Hence the retained alignment is $\mathcal{W} = \{(1, 1), (2, 3), (5, 4)\}$, while the unmatched action *stain* is discarded. This anchor-based procedure ensures that semantic comparisons focus only on correctly aligned procedural units, avoiding misleading matches across unrelated operations.

## D RL ALGORITHM

During the RL stage, we adopt Group Relative Policy Optimization (GRPO) (Shao et al., 2024), a variant of PPO (Schulman et al., 2017) designed to reduce variance and stabilize updates by exploiting multiple responses generated from the same query. Unlike standard PPO, which normalizes the reward across a batch, GRPO performs group-wise normalization within each query group. This strategy ensures that the relative quality of responses to the same query is emphasized, thereby mitigating reward scale differences across queries and improving training stability. Formally, let $x$ denote a query, and $\{y_i\}_{i=1}^G$ be $G$ responses sampled from the current policy $\pi_\theta(\cdot|x)$. For each response $y_i$, the normalized advantage is defined as

$$\hat{A}_i = \frac{r(x, y_i) - \mathrm{mean}\big(\{r(x, y_j)\}_{j=1}^G\big)}{\mathrm{std}\big(\{r(x, y_j)\}_{j=1}^G\big)}, \tag{16}$$

where $r(x, y_i)$ denotes the scalar reward assigned to $y_i$. By centering and scaling rewards within the group, the advantage highlights relative performance among candidate responses.

The GRPO objective combines this normalized advantage with the clipped surrogate function and a KL-regularization term that constrains the updated policy toward the reference model:

$$\mathcal{L}_{\mathrm{GRPO}}(\theta) = \mathbb{E}_{x \sim \mathcal{D}, \{y_i\}_{i=1}^G \sim \pi_\theta(\cdot|x)} \left[ \frac{1}{G} \sum_{i=1}^G \min\Big(s_i(\theta)\hat{A}_i, \ \mathrm{clip}\big(s_i(\theta), 1 - \epsilon, 1 + \epsilon\big)\hat{A}_i\Big) \right] \tag{17}$$

$$- \beta \, \mathbb{E}_{x \sim \mathcal{D}} \left[ \mathrm{KL}\big(\pi_\theta(\cdot|x) \, \| \, \pi_{\theta_{\mathrm{ref}}}(\cdot|x)\big) \right],$$

where $\epsilon$ is a clipping threshold controlling the size of policy updates and $\beta$ determines the strength of KL regularization, with $\pi_\theta$ denoting the current policy being optimized and $\pi_{\theta_{\mathrm{ref}}}$ the fixed reference policy. The importance sampling ratio $s_i(\theta)$ is computed at the token level to account for the sequential nature of text generation. Specifically, it is defined as the geometric mean of the per-token probability ratios:

$$s_i(\theta) = \exp\left( \frac{1}{|y_i|} \sum_{t=1}^{|y_i|} \log \frac{\pi_\theta(y_{i,t} \mid x, y_{i,<t})}{\pi_{\theta_{\mathrm{old}}}(y_{i,t} \mid x, y_{i,<t})} \right), \tag{18}$$

where $\pi_{\theta_{\mathrm{old}}}$ denote the old policy models, respectively, and $y_{i,t}$ is the $t$-th token of response $y_i$. This per-token formulation maintains sensitivity to fine-grained action probabilities and avoids domination by sequence length. Overall, GRPO stabilizes training by normalizing advantages within query groups, constrains policy updates through clipping and KL regularization, and ensures robustness at the token level via per-token importance ratios.

## E    BASELINE MODELS

Given the lack of models tailored for scientific protocol generation, we compare our approach against a broad set of large-scale language models. The closed-source baselines include the GPT-5 series (OpenAI, 2025) (GPT-5, GPT-5 Chat), the Claude series (Anthropic, 2025) (Claude Sonnet 4, Claude Opus 4.1), the Gemini 2.5 series (Comanici et al., 2025) (Flash and Pro), as well as ChatGPT-4o (Achiam et al., 2023), Doubao-1.5-Pro (ByteDance, 2025), and Qwen2.5-Max (Qwen et al., 2025). For open-source baselines, we evaluate strong instruction-tuned models such as Qwen2.5-72B (Qwen et al., 2025), Qwen3-235B (Yang et al., 2025), DeepSeek-V3 (Liu et al., 2024), GPT-OSS-120B (Agarwal et al., 2025), Llama-3.1-405B (Grattafiori et al., 2024), and Kimi-K2 (Team et al., 2025). In addition, we consider reasoning-oriented models including DeepSeek-R1 (Guo et al., 2025), Grok 3/4 (xAI, 2025), and OpenAI's o1 and o3 (Jaech et al., 2024). Finally, to assess scientific capabilities, we further evaluate domain-specific models including Intern-S1-mini, Intern-S1 (Bai et al., 2025), and SciDFM (Sun et al., 2024). Together, these baselines provide a comprehensive landscape of frontier, open-source, reasoning, and domain-oriented models for comparison.

## F    REPRODUCIBILITY STATEMENT

We provide essential information to ensure the reproducibility of our experiments. Table 5 reports the training data distribution across stages, covering both SciRecipe tasks and BioProBench subtasks. Key hyperparameters for pre-training, SFT, and RL are listed in Tables 6, 7, and 8. All experiments were conducted with the LLaMA-Factory framework for pre-training and SFT, and VeRL for RL training. We release detailed settings, including LoRA configurations, learning rates, epochs, scheduling, and hardware, so that independent researchers can replicate our pipeline and validate the reported results. All data, code, and models will be released publicly.

Table 5: Training data distribution across different stages (one epoch). "PC" denotes Protocol-Comprehension tasks and "PS" denotes Problem-Solving tasks. Four task types from BioProBench (Liu et al., 2025) are also incorporated to enhance the model's ability to interpret protocols.

| Dataset | SFT stage | RL stage |
|---|---|---|
| SciRecipe_PC | 34461 | 6956 |
| SciRecipe_PS | 16164 | 3263 |
| BioProBench_PQA | 52640 | — |
| BioProBench_ERR | 103983 | — |
| BioProBench_ORD | 31039 | — |
| BioProBench_GEN | 59477 | — |

Table 6: Key hyperparameters used in LLaMA-Factory for pre-training.

| Parameter | Value |
|---|---|
| finetuning_type | lora |
| lora_rank | 8 |
| lora_target | all |
| preprocessing_num_workers | 16 |
| dataloader_num_workers | 4 |
| per_device_train_batch_size | 1 |
| gradient_accumulation_steps | 8 |
| learning_rate | 1e-4 |
| num_train_epochs | 6 |
| lr_scheduler_type | cosine |
| warmup_ratio | 0.1 |
| bf16 | true |

Table 7: Key hyperparameters used in LLaMA-Factory for SFT training.

| Parameter | Value |
|---|---|
| finetuning_type | lora |
| lora_rank | 32 |
| lora_target | all |
| preprocessing_num_workers | 8 |
| dataloader_num_workers | 4 |
| per_device_train_batch_size | 1 |
| gradient_accumulation_steps | 8 |
| learning_rate | 3e-4 |
| num_train_epochs | 5 |
| lr_scheduler_type | cosine |
| warmup_ratio | 0.05 |
| bf16 | true |

## G  LIMITATIONS

While our work presents a structured framework for scientific protocol generation and shows consistent improvements across benchmarks, some limitations remain. First, although SciRecipe spans 27 biological subfields, it is still skewed toward widely used laboratory techniques, with rare or highly specialized protocols underrepresented. Second, the SCORE metrics mainly emphasize step completeness and order fidelity, but do not yet capture finer aspects such as stylistic variation, cross-step dependencies, or long-range experimental context. Third, our experiments were conducted in relatively controlled settings, which may not fully reflect the complexity and variability of real laboratory environments. We believe these issues can be progressively addressed in future work by expanding the dataset to cover rarer domains, enriching evaluation dimensions beyond structural executability, and validating the framework under more diverse practical scenarios.

## H  USAGE OF LLMS ON PAPER WRITING

We used LLMs (e.g., GPT-5) to aid in polishing the writing of this paper. Specifically, LLMs were employed to refine grammar, improve fluency, and adjust the tone of academic writing (e.g., smoothing transitions between sections, shortening overly long sentences, and rephrasing repetitive expressions). The models were not used for generating technical content, conducting experiments, or formulating research ideas. All substantive contributions remain the work of the authors.

Table 8: Key hyperparameters used in VeRL for RL training.

| Parameter | Value |
| --- | --- |
| algorithm.adv_estimator | grpo |
| data.train_batch_size | 1024 |
| data.max_prompt_length | 1024 |
| data.max_response_length | 1024 |
| actor.optim.lr | 7e-6 |
| actor.ppo_mini_batch_size | 256 |
| actor.ppo_micro_batch_size_per_gpu | 32 |
| actor.use_kl_loss | true |
| actor.kl_loss_coef | 0.001 |
| actor.kl_loss_type | low_var_kl |
| actor.entropy_coeff | 0 |
| rollout.tensor_model_parallel_size | 2 |
| rollout.gpu_memory_utilization | 0.6 |
| rollout.n | 5 |
| trainer.total_epochs | 15 |
| trainer.n_gpus_per_node | 8 |
| trainer.nnodes | 1 |

# I  ADDITIONAL EXPERIMENTAL RESULTS

## I.1  QUALITATIVE STUDY OF REASONING MODELS

As discussed in the main text, general-purpose reasoning models often "overthink" protocol generation, producing overly long procedures with redundant actions and no clear stopping criteria. In the permeability assay, the gold-standard protocol requires only four steps (wash–remove–incubate–wash) to ensure structural preservation and experimental efficiency. Thoth closely followed this structure, preserving both the correct sequence and action fidelity. In contrast, Grok 4 expanded the task into a 13-step immunostaining workflow, adding permeabilization, blocking, antibody staining, and repeated washes far beyond the intended fixation step. Although logically coherent within its reasoning chain, this output diverges from laboratory practice, where clarity and parsimony are essential. This case underscores a broader limitation of general-purpose reasoning models: their tendency to prioritize exhaustive reasoning over practical executability, reinforcing our quantitative findings that domain-adapted models like Thoth achieve more concise, faithful, and actionable protocols.

**Grok 4**

**Query:**

In the "Generation of a Human Conditionally Immortalized Cell-based Multicellular Spheroidal Blood-Brain Barrier Model for Permeability Evaluation of Macromolecules" protocol, what are the exact sequential steps required for fixing and staining the spheroids during the permeability assay to ensure proper preservation and preparation for microscopy?

**Ground Truth:**

Step 1: Wash the spheroids twice with 500 µl of ice-cold D-PBS(+).
Step 2: Remove the supernatant with a pipette.
Step 3: Incubate the spheroids with 500 µl of 4% PFA at room temperature for 10 min.
Step 4: Wash the spheroids twice with 500 µl of ice-cold D-PBS(+).

---

**Model Output:**

<think>
The goal is to outline the sequential steps for fixing and staining spheroids in the specified blood-brain barrier model protocol during the permeability assay, ensuring preservation of structure and preparation for microscopy. The hypothesis is that a standard fixation and immunostaining procedure will enable visualization of macromolecule permeability without compromising spheroid integrity. The sequence starts with removing medium to prepare for fixation, followed by washing, fixing, permeabilizing, blocking, antibody staining, and final washes; this order is justified by established immunohistochemical techniques that sequentially preserve, permeabilize, and label cellular components while minimizing background and artifacts.
</think>

<key>
Step 1: {"action":"remove","objects":["medium"],"parameters":[]}
Step 2: {"action":"wash","objects":["spheroids"],"parameters":["with 1x pbs"]}
Step 3: {"action":"incubate","objects":["spheroids"],"parameters":["in 4% pfa","30 min","room temperature"]}
Step 4: {"action":"wash","objects":["spheroids"],"parameters":["with 1x pbs","3 times"]}
Step 5: {"action":"incubate","objects":["spheroids"],"parameters":["in 0.1% triton x-100","15 min","room temperature"]}
Step 6: {"action":"wash","objects":["spheroids"],"parameters":["with 1x pbs","3 times"]}
Step 7: {"action":"incubate","objects":["spheroids"],"parameters":["in blocking buffer","1 h","room temperature"]}
Step 8: {"action":"incubate","objects":["spheroids"],"parameters":["in primary antibody","overnight","4 °c"]}
Step 9: {"action":"wash","objects":["spheroids"],"parameters":["with 1x pbs","3 times"]}
Step 10: {"action":"incubate","objects":["spheroids"],"parameters":["in secondary antibody","1 h","room temperature"]}
Step 11: {"action":"wash","objects":["spheroids"],"parameters":["with 1x pbs","3 times"]}
Step 12: {"action":"incubate","objects":["spheroids"],"parameters":["in dapi","5 min","room temperature"]}
Step 13: {"action":"wash","objects":["spheroids"],"parameters":["with 1x pbs","3 times"]}
</key>

<orc>
Step 1: Remove the medium.
Step 2: Wash the spheroids with 1x pbs.
Step 3: Incubate the spheroids in 4% pfa for 30 min at room temperature.
Step 4: Wash the spheroids with 1x pbs 3 times.
Step 5: Incubate the spheroids in 0.1% triton x-100 for 15 min at room temperature.
Step 6: Wash the spheroids with 1x pbs 3 times.
Step 7: Incubate the spheroids in blocking buffer for 1 h at room temperature.
Step 8: Incubate the spheroids in primary antibody overnight at 4 °c.
Step 9: Wash the spheroids with 1x pbs 3 times.
Step 10: Incubate the spheroids in secondary antibody for 1 h at room temperature.
Step 11: Wash the spheroids with 1x pbs 3 times.
Step 12: Incubate the spheroids in dapi for 5 min at room temperature.
Step 13: Wash the spheroids with 1x pbs 3 times.
</orc>

<note>
Handle paraformaldehyde and triton x-100 with care in a fume hood using gloves and eye protection, as they are hazardous chemicals that can cause skin irritation, respiratory issues, or toxicity; ensure proper disposal of waste to avoid environmental contamination.
</note>

Figure 7: Case study of the reasoning model Grok 4 on Protocol-Comprehension task.

**OpenAI o1**

**Query:**

In the "Generation of a Human Conditionally Immortalized Cell-based Multicellular Spheroidal Blood-Brain Barrier Model for Permeability Evaluation of Macromolecules" protocol, what are the exact sequential steps required for fixing and staining the spheroids during the permeability assay to ensure proper preservation and preparation for microscopy?

**Ground Truth:**

Step 1: Wash the spheroids twice with 500 μl of ice-cold D-PBS(+).
Step 2: Remove the supernatant with a pipette.
Step 3: Incubate the spheroids with 500 μl of 4% PFA at room temperature for 10 min.
Step 4: Wash the spheroids twice with 500 μl of ice-cold D-PBS(+).

**Model Output:**

<think>
We will sequentially remove old medium, wash, fix with paraformaldehyde, remove fixative, wash, permeabilize, wash, then incubate with primary and secondary antibodies. We will end with a final wash and removal before imaging. Each step uses only "incubate", "wash", or "remove" to maintain consistency with the allowed actions.
</think>

<key>
Step 1: {"action":"remove","objects":["culture medium"],"parameters":["from wells"]}
Step 2: {"action":"wash","objects":["spheroids"],"parameters":["1x pbs","once","rt"]}
Step 3: {"action":"remove","objects":["wash buffer"],"parameters":[]}
Step 4: {"action":"incubate","objects":["spheroids"],"parameters":["4% pfa","15 min","rt"]}
Step 5: {"action":"remove","objects":["fixative"],"parameters":[]}
Step 6: {"action":"wash","objects":["spheroids"],"parameters":["1x pbs","twice","rt"]}
Step 7: {"action":"remove","objects":["wash buffer"],"parameters":[]}
Step 8: {"action":"incubate","objects":["spheroids"],"parameters":["0.1% triton x-100","15 min","rt"]}
Step 9: {"action":"remove","objects":["permeabilization solution"],"parameters":[]}
Step 10: {"action":"wash","objects":["spheroids"],"parameters":["1x pbs","once","rt"]}
Step 11: {"action":"remove","objects":["wash buffer"],"parameters":[]}
Step 12: {"action":"incubate","objects":["spheroids"],"parameters":["primary antibody","1 h","rt"]}
Step 13: {"action":"remove","objects":["antibody solution"],"parameters":[]}
Step 14: {"action":"wash","objects":["spheroids"],"parameters":["1x pbs","once","rt"]}
Step 15: {"action":"remove","objects":["wash buffer"],"parameters":[]}
Step 16: {"action":"incubate","objects":["spheroids"],"parameters":["secondary antibody","1 h","rt"]}
Step 17: {"action":"remove","objects":["antibody solution"],"parameters":[]}
Step 18: {"action":"wash","objects":["spheroids"],"parameters":["1x pbs","once","rt"]}
Step 19: {"action":"remove","objects":["wash buffer"],"parameters":[]}
</key>

<orc>
Step 1: Remove culture medium from wells.
Step 2: Wash spheroids with 1x pbs once at rt.
Step 3: Remove wash buffer.
Step 4: Incubate spheroids with 4% pfa for 15 min at rt.
Step 5: Remove fixative.
Step 6: Wash spheroids with 1x pbs twice at rt.
Step 7: Remove wash buffer.
Step 8: Incubate spheroids with 0.1% triton x-100 for 15 min at rt.
Step 9: Remove permeabilization solution.
Step 10: Wash spheroids with 1x pbs once at rt.
Step 11: Remove wash buffer.
Step 12: Incubate spheroids with primary antibody for 1 h at rt.
Step 13: Remove antibody solution.
Step 14: Wash spheroids with 1x pbs once at rt.
Step 15: Remove wash buffer.
Step 16: Incubate spheroids with secondary antibody for 1 h at rt.
Step 17: Remove antibody solution.
Step 18: Wash spheroids with 1x pbs once at rt.
Step 19: Remove wash buffer.
</orc>

<note>
Wear personal protective equipment, handle paraformaldehyde in a ventilated area, and dispose of hazardous solutions according to institutional safety guidelines.
</note>

Figure 8: Case study of the reasoning model o1 on Protocol-Comprehension tasks.

**Grok 4**

**Query:**

If the berlin (1800 ml vessel) normally used to hold distilled water for immersing the basket is unavailable, how can the QuantiSlakeTest be adapted using smaller vessels while still ensuring accurate immersion and balance measurements?

**Ground Truth:**

Step 1: Measure 1800 ml of distilled water.
Step 2: Transfer the distilled water into an alternative container.
Step 3: Suspend the empty basket in the distilled water without touching the sides.
Step 4: Tare the balance with the basket immersed.

**Model Output:**

<think>
The goal is to adapt the QuantiSlakeTest to use smaller vessels instead of the unavailable 1800 ml berlin vessel for holding distilled water and immersing the basket, while ensuring accurate immersion and balance measurements. Hypothesis/plan of attack: the test can be adapted by conducting it in a smaller vessel with appropriate water volume, ensuring full basket immersion through suspension and restriction, and achieving accurate balance measurements by taring, measuring at key points, recapturing slaked material, and scrubbing to recover all residue for complete quantification. Justification: smaller vessels may limit water volume and affect immersion stability or residue recovery, so the sequence starts with cleaning the vessel, transferring water, suspending and restricting the basket for secure immersion, taring and measuring initially, then post-slaking recapture, scrubbing for residue, transferring residue, and final measuring to ensure all material is accounted for, minimizing errors in smaller-scale setup.
</think>

<key>
Step 1: {"action":"scrub","objects":["smaller vessel"],"parameters":[]}
Step 2: {"action":"transfer","objects":["distilled water"],"parameters":["to smaller vessel"]}
Step 3: {"action":"suspend","objects":["basket"],"parameters":["in smaller vessel"]}
Step 4: {"action":"restrict","objects":["basket"],"parameters":["to ensure immersion"]}
Step 5: {"action":"tare","objects":["balance"],"parameters":[]}
Step 6: {"action":"measure","objects":["immersed basket"],"parameters":["using balance"]}
Step 7: {"action":"recapture","objects":["slaked material"],"parameters":["from smaller vessel"]}
Step 8: {"action":"scrub","objects":["smaller vessel"],"parameters":["to recover residue"]}
Step 9: {"action":"transfer","objects":["residue"],"parameters":["to slaked material"]}
Step 10: {"action":"measure","objects":["slaked material"],"parameters":["using balance"]}
</key>

<orc>
Step 1: Scrub the smaller vessel.
Step 2: Transfer distilled water to the smaller vessel.
Step 3: Suspend the basket in the smaller vessel.
Step 4: Restrict the basket to ensure immersion.
Step 5: Tare the balance.
Step 6: Measure the immersed basket using the balance.
Step 7: Recapture the slaked material from the smaller vessel.
Step 8: Scrub the smaller vessel to recover residue.
Step 9: Transfer the residue to the slaked material.
Step 10: Measure the slaked material using the balance.
</orc>

<note>
Ensure the balance is stable and level before taring and measuring to prevent inaccurate readings or tip-overs, and wear protective gloves during scrubbing, transferring, and recapturing to avoid skin contact with potential residues or contaminants.
</note>

Figure 9: Case study of the reasoning model Grok 4 on Problem-Solving task.

**OpenAI o1**

**Query:**

If the berlin (1800 ml vessel) normally used to hold distilled water for immersing the basket is unavailable, how can the QuantiSlakeTest be adapted using smaller vessels while still ensuring accurate immersion and balance measurements?

**Ground Truth:**

Step 1: Measure 1800 ml of distilled water.
Step 2: Transfer the distilled water into an alternative container.
Step 3: Suspend the empty basket in the distilled water without touching the sides.
Step 4: Tare the balance with the basket immersed.

---

**Model Output:**

<think>
We will employ smaller vessels and ensure precise immersion by scrubbing them clean, taring the scale with each empty vessel, transferring distilled water, verifying the fill and adjusting if necessary, suspending the basket for measurement, and finally recapturing water for reuse.
</think>

<key>
Step 1: {"action":"scrub","objects":["smaller vessels"],"parameters":["to remove residue"]}
Step 2: {"action":"tare","objects":["scale"],"parameters":["with empty vessel"]}
Step 3: {"action":"transfer","objects":["distilled water"],"parameters":["into vessel"]}
Step 4: {"action":"measure","objects":["filled vessel"],"parameters":["to confirm total mass"]}
Step 5: {"action":"restrict","objects":["water level"],"parameters":["to required volume"]}
Step 6: {"action":"suspend","objects":["basket"],"parameters":["in vessel","for immersion"]}
Step 7: {"action":"measure","objects":["basket weight"],"parameters":["for accurate reading"]}
Step 8: {"action":"recapture","objects":["distilled water"],"parameters":["for reuse"]}
</key>

<orc>
Step 1: Scrub the smaller vessels to remove residue.
Step 2: Tare the scale with empty vessel.
Step 3: Transfer distilled water into vessel.
Step 4: Measure the filled vessel to confirm total mass.
Step 5: Restrict the water level to required volume.
Step 6: Suspend the basket in vessel for immersion.
Step 7: Measure the basket weight for accurate reading.
Step 8: Recapture the distilled water for reuse.
</orc>

<note>
Always verify the vessel's stability before suspending objects, and ensure all surfaces remain dry to prevent accidental spills or scale errors.
</note>

Figure 10: Case study of the reasoning model o1 on Problem-Solving task.

## I.2 Fine-grained Results on SciRecipe-Eval

Table 9: Main results on SciRecipe-Eval at level 1. Metrics left of the dashed line evaluate executability, those on the right measure semantic similarity. **Bold** denotes the best score.

| Methods | Semantic-A | Order-LCS | Order-S | Order-Tau | Step-M | BLUE-AVG | ROUGE-L | METEOR | KW-F1 | AVG |
|---|---|---|---|---|---|---|---|---|---|---|
| *Close-Source SOTA* | | | | | | | | | | |
| ChatGPT-4o | 44.44 | 78.95 | 39.49 | 80.04 | 56.49 | 40.98 | 52.11 | 48.02 | 55.37 | 55.10 |
| GPT-5 | 32.82 | 63.40 | 19.27 | 63.63 | 27.62 | 22.34 | 35.92 | 35.41 | 42.61 | 38.11 |
| GPT-5 Chat | 41.50 | 78.91 | 35.75 | 73.50 | 35.06 | 31.74 | 45.77 | 45.80 | 50.60 | 48.74 |
| Claude Sonnet 4 | 44.35 | 76.44 | 33.70 | 76.90 | 47.65 | 35.91 | 47.54 | 43.61 | 52.96 | 51.01 |
| Claude Opus 4.1 | 45.43 | 75.42 | 34.82 | 76.88 | 46.89 | 36.29 | 46.90 | 42.14 | 52.36 | 50.79 |
| Gemini 2.5 Flash | 39.66 | 75.32 | 33.70 | 75.66 | 43.91 | 34.60 | 46.21 | 42.55 | 50.68 | 49.14 |
| Gemini 2.5 Pro | 38.78 | 77.51 | 36.08 | 76.56 | 40.17 | 32.71 | 46.84 | 47.84 | 50.64 | 49.68 |
| Doubao-1.5-pro | 37.14 | 78.42 | 38.47 | 76.00 | 60.23 | 39.78 | 49.57 | 41.77 | 51.50 | 52.54 |
| Qwen2.5-Max | 45.77 | 78.43 | 37.78 | 79.28 | 61.93 | 31.96 | 51.43 | 47.16 | 55.29 | 54.34 |
| *Open-Source SOTA* | | | | | | | | | | |
| Qwen2.5-72B-Instruct | 40.13 | 77.15 | 37.10 | 77.93 | 59.21 | 31.90 | 49.87 | 46.58 | 52.54 | 52.49 |
| Qwen3-235B-A22B-Instruct-2507 | 39.68 | 78.19 | 35.08 | 78.28 | 54.82 | 35.68 | 48.10 | 47.11 | 51.20 | 52.02 |
| DeepSeek-V3 | 45.10 | 78.59 | 33.84 | 75.64 | 52.27 | 39.66 | 51.46 | 48.04 | 54.55 | 53.24 |
| GPT-OSS-120B | 36.56 | 76.12 | 31.66 | 74.18 | 44.25 | 34.12 | 47.31 | 46.23 | 53.08 | 49.28 |
| Llama-3.1-405B-Instruct | 39.80 | 74.42 | 28.34 | 73.06 | 49.50 | 37.28 | 47.38 | 45.11 | 50.29 | 49.46 |
| Kimi-K2-Instruction | 42.58 | 76.96 | 34.72 | 78.60 | 52.07 | 35.40 | 47.65 | 45.78 | 51.61 | 51.71 |
| *Reasoning Models* | | | | | | | | | | |
| DeepSeek-R1 | 37.86 | 77.08 | 35.43 | 76.35 | 45.97 | 40.81 | 48.96 | 42.45 | 52.56 | 50.83 |
| Grok 3 | 41.53 | 78.84 | 37.68 | 77.96 | 54.19 | 37.12 | 49.82 | **49.98** | 52.28 | 53.27 |
| Grok 4 | 42.52 | 77.27 | 34.92 | 75.66 | 49.52 | 38.91 | 49.75 | 43.72 | 54.34 | 51.85 |
| OpenAI o1 | 39.78 | 77.70 | 28.00 | 74.43 | 43.02 | 37.25 | 49.23 | 46.19 | 53.12 | 49.86 |
| OpenAI o3 | 42.15 | 74.94 | 26.02 | 72.95 | 36.29 | 31.10 | 46.42 | 46.78 | 53.59 | 47.80 |
| Qwen3-4B | 27.21 | 59.83 | 24.19 | 59.89 | 40.17 | 16.29 | 28.17 | 27.22 | 31.41 | 34.93 |
| Thoth-mini | 49.49 | **80.41** | **42.88** | 80.30 | **64.66** | **46.50** | 51.91 | 48.24 | 55.77 | 57.80 |
| Qwen3-8B | 29.98 | 68.08 | 21.12 | 65.00 | 38.80 | 17.98 | 34.59 | 36.67 | 40.15 | 39.15 |
| Thoth | **52.21** | 80.12 | 41.19 | **83.20** | 62.27 | 46.41 | **52.69** | 49.21 | **56.88** | **58.24** |

Table 10: Main results on SciRecipe-Eval at level 2. Metrics left of the dashed line evaluate executability, those on the right measure semantic similarity. **Bold** denotes the best score.

| Methods | Semantic-A | Order-LCS | Order-S | Order-Tau | Step-M | BLUE-AVG | ROUGE-L | METEOR | KW-F1 | AVG |
|---|---|---|---|---|---|---|---|---|---|---|
| *Close-Source SOTA* | | | | | | | | | | |
| ChatGPT-4o | 35.82 | 67.82 | 9.12 | 61.01 | 32.00 | 37.00 | 44.88 | 41.44 | 48.87 | 42.00 |
| GPT-5 | 22.96 | 53.05 | 3.75 | 43.87 | 10.31 | 20.33 | 30.12 | 29.81 | 35.87 | 27.79 |
| GPT-5 Chat | 31.31 | 67.74 | 7.17 | 58.15 | 15.34 | 27.49 | 38.46 | 38.26 | 45.25 | 36.57 |
| Claude Sonnet 4 | 34.55 | 67.68 | 8.47 | 63.38 | 24.48 | 32.64 | 41.13 | 38.44 | 45.98 | 39.64 |
| Claude Opus 4.1 | 37.38 | 68.13 | 9.30 | **67.18** | 22.78 | 33.16 | 42.04 | 38.65 | 47.74 | 40.71 |
| Gemini 2.5 Flash | 33.17 | 66.09 | 6.84 | 65.21 | 21.21 | 31.84 | 39.74 | 36.10 | 45.57 | 38.42 |
| Gemini 2.5 Pro | 32.94 | 68.04 | 8.14 | 64.04 | 24.16 | 30.09 | 41.59 | 43.43 | 46.61 | 39.89 |
| Doubao-1.5-pro | 29.67 | 68.37 | 9.46 | 64.24 | 35.27 | 36.61 | 44.30 | 35.77 | 46.09 | 41.09 |
| Qwen2.5-Max | 35.13 | 67.55 | 6.51 | 63.70 | 33.64 | 29.71 | 44.75 | 40.62 | 48.80 | 41.16 |
| *Open-Source SOTA* | | | | | | | | | | |
| Qwen2.5-72B-Instruct | 32.82 | 64.74 | 5.54 | 60.76 | 25.80 | 27.30 | 42.40 | 40.43 | 47.04 | 38.54 |
| Qwen3-235B-A22B-Instruct-2507 | 31.84 | 66.19 | 5.58 | 60.32 | 21.32 | 29.41 | 40.79 | 41.61 | 44.71 | 37.97 |
| DeepSeek-V3 | 38.48 | 69.54 | 9.53 | 65.64 | 31.57 | 36.76 | 45.64 | 42.24 | 50.20 | 43.29 |
| GPT-OSS-120B | 29.31 | 64.07 | 4.23 | 54.56 | 12.06 | 27.46 | 39.73 | 39.71 | 46.89 | 35.34 |
| Llama-3.1-405B-Instruct | 32.20 | 64.70 | 8.13 | 62.71 | 29.37 | 35.60 | 42.44 | 39.35 | 45.98 | 40.05 |
| Kimi-K2-Instruction | 31.62 | 66.91 | 7.49 | 61.41 | 28.41 | 32.29 | 41.46 | 40.31 | 46.50 | 39.60 |
| *Reasoning Models* | | | | | | | | | | |
| DeepSeek-R1 | 34.36 | 65.91 | 5.91 | 62.18 | 20.33 | 38.83 | 42.89 | 34.10 | 47.21 | 39.08 |
| Grok 3 | 33.44 | 67.92 | 6.78 | 61.83 | 24.90 | 32.42 | 42.41 | 42.59 | 45.05 | 39.70 |
| Grok 4 | 31.17 | 67.10 | 6.16 | 55.43 | 20.39 | 35.58 | 42.74 | 36.78 | 49.38 | 38.30 |
| OpenAI o1 | 29.90 | 69.27 | 9.44 | 60.75 | 28.06 | 34.18 | 43.47 | 41.55 | 47.26 | 40.43 |
| OpenAI o3 | 28.92 | 66.00 | 5.16 | 57.46 | 12.35 | 26.24 | 39.88 | 41.98 | 47.07 | 36.12 |
| Qwen3-4B | 21.65 | 47.52 | 3.57 | 41.48 | 17.94 | 12.82 | 21.45 | 20.81 | 24.12 | 23.48 |
| Thoth-mini | 39.28 | 69.18 | 8.47 | 61.74 | 41.16 | 40.27 | 46.66 | 44.66 | 50.60 | 44.67 |
| Qwen3-8B | 27.85 | 59.12 | 1.62 | 52.59 | 10.43 | 15.40 | 30.12 | 32.85 | 37.17 | 29.68 |
| Thoth | **41.21** | **70.75** | **10.43** | 63.85 | **44.10** | **40.94** | **47.46** | **45.65** | **51.49** | **46.21** |

Table 11: Main results on SciRecipe-Eval (Overview). Metrics left of the dashed line evaluate executability, those on the right measure semantic similarity. **Bold** denotes the best score.

| Methods | Semantic-A | Order-LCS | Order-S | Order-Tau | Step-M | BLUE-AVG | ROUGE-L | METEOR | KW-F1 | AVG |
|---|---|---|---|---|---|---|---|---|---|---|
| *Close-Source SOTA* | | | | | | | | | | |
| ChatGPT-4o | 32.36 | 72.47 | 13.33 | 72.02 | 30.67 | 34.24 | 42.36 | 37.24 | 44.03 | 42.08 |
| GPT-5 | 27.85 | 65.51 | 9.59 | 64.64 | 16.29 | 23.26 | 33.20 | 30.29 | 35.30 | 33.99 |
| GPT-5 Chat | 30.99 | 70.02 | 10.01 | 68.01 | 11.33 | 21.35 | 32.70 | 32.98 | 38.10 | 35.05 |
| Claude Sonnet 4 | 30.08 | 72.64 | **16.00** | 71.33 | 30.67 | 31.42 | 38.94 | 35.62 | 41.20 | 40.88 |
| Claude Opus 4.1 | 30.72 | 71.52 | 14.52 | 68.38 | 24.69 | 31.65 | 39.70 | 36.02 | 42.32 | 39.95 |
| Gemini 2.5 Flash | 28.87 | 70.62 | 12.67 | 73.67 | 26.00 | 31.85 | 39.04 | 33.00 | 41.57 | 39.70 |
| Gemini 2.5 Pro | 28.96 | 71.83 | 12.00 | 75.35 | 26.67 | 28.04 | 38.99 | 38.83 | 41.38 | 40.23 |
| Doubao-1.5-pro | 26.85 | 71.67 | 12.01 | 76.34 | 34.67 | 37.87 | 42.05 | 30.53 | 41.36 | 41.48 |
| Qwen2.5-Max | 34.73 | 71.56 | 10.00 | 76.67 | 36.00 | 28.11 | 42.12 | 36.27 | 44.08 | 42.17 |
| *Open-Source SOTA* | | | | | | | | | | |
| Qwen2.5-72B-Instruct | 30.22 | 69.32 | 10.00 | 75.35 | 27.35 | 24.89 | 39.97 | 37.00 | 42.17 | 39.59 |
| Qwen3-235B-A22B-Instruct-2507 | 28.70 | 68.69 | 8.67 | 70.70 | 21.33 | 26.67 | 37.29 | 37.33 | 39.26 | 37.63 |
| DeepSeek-V3 | 35.24 | **73.23** | 15.54 | 71.01 | 37.16 | 35.33 | **43.43** | 39.58 | 43.86 | **43.82** |
| GPT-OSS-120B | 27.34 | 67.29 | 10.01 | 64.68 | 20.00 | 24.75 | 36.10 | 36.71 | 39.61 | 36.28 |
| Llama-3.1-405B-Instruct | 31.99 | 71.19 | 14.67 | **79.36** | 32.00 | 36.59 | 40.78 | 35.59 | 40.62 | 42.53 |
| Kimi-K2-Instruction | 30.92 | 72.21 | 12.00 | 74.67 | 36.67 | 30.90 | 39.20 | 37.88 | 41.18 | 41.74 |
| *Reasoning Models* | | | | | | | | | | |
| DeepSeek-R1 | 31.65 | 70.90 | 12.00 | 69.67 | 27.34 | **40.45** | 42.47 | 32.09 | 43.34 | 41.10 |
| Grok 3 | 30.56 | 70.29 | 12.42 | 64.01 | 26.86 | 30.53 | 39.60 | **40.02** | 40.21 | 39.39 |
| Grok 4 | 32.19 | 72.43 | 12.52 | 67.45 | 29.30 | 36.06 | 42.33 | 36.12 | **45.54** | 41.55 |
| OpenAI o1 | 27.78 | 72.80 | 12.67 | 70.70 | 31.33 | 31.80 | 40.20 | 37.84 | 40.75 | 40.65 |
| OpenAI o3 | 28.48 | 71.01 | 8.61 | 72.00 | 20.64 | 25.24 | 37.21 | 38.26 | 41.18 | 38.07 |
| Qwen3-4B | 28.05 | 62.52 | 8.02 | 62.00 | 24.00 | 13.46 | 24.50 | 25.40 | 27.38 | 30.59 |
| Thoth-mini | 37.49 | 71.74 | 11.34 | 71.67 | **38.34** | 38.02 | 42.61 | 38.55 | 43.79 | 43.73 |
| Qwen3-8B | 30.46 | 65.41 | 2.68 | 62.01 | 10.67 | 11.60 | 26.80 | 31.97 | 32.14 | 30.42 |
| Thoth | **40.15** | 72.10 | 9.35 | 71.01 | 36.35 | 37.75 | 43.27 | 39.09 | 44.79 | 43.76 |

Table 12: Main results on SciRecipe-Eval (Specific). Metrics left of the dashed line evaluate executability, those on the right measure semantic similarity. **Bold** denotes the best score.

| Methods | Semantic-A | Order-LCS | Order-S | Order-Tau | Step-M | BLUE-AVG | ROUGE-L | METEOR | KW-F1 | AVG |
|---|---|---|---|---|---|---|---|---|---|---|
| *Close-Source SOTA* | | | | | | | | | | |
| ChatGPT-4o | 33.63 | 70.95 | 24.00 | 67.32 | 48.67 | 36.26 | 45.07 | 41.29 | 46.81 | 46.00 |
| GPT-5 | 18.99 | 48.00 | 12.59 | 43.55 | 24.26 | 19.21 | 27.31 | 24.67 | 28.01 | 27.40 |
| GPT-5 Chat | 27.99 | 71.05 | 20.67 | 64.67 | 18.67 | 23.93 | 36.23 | 36.46 | 39.56 | 37.69 |
| Claude Sonnet 4 | 36.39 | 66.46 | 20.67 | 66.67 | 37.33 | 33.11 | 40.55 | 35.55 | 42.88 | 42.18 |
| Claude Opus 4.1 | 37.78 | 67.09 | 20.18 | 67.81 | 35.95 | 35.76 | 40.69 | 33.60 | 44.92 | 42.64 |
| Gemini 2.5 Flash | 36.67 | 68.25 | 20.00 | 70.33 | 30.67 | 31.72 | 40.34 | 36.03 | 43.34 | 41.93 |
| Gemini 2.5 Pro | 35.48 | 71.68 | 23.33 | 69.33 | 30.00 | 27.89 | 40.09 | 41.84 | 42.08 | 42.41 |
| Doubao-1.5-pro | 30.96 | 71.96 | 26.00 | 66.33 | 52.00 | 37.91 | 44.14 | 35.84 | 43.69 | 45.43 |
| Qwen2.5-Max | 36.91 | 71.18 | 23.33 | 64.67 | 46.67 | 27.73 | 45.37 | 40.97 | 46.94 | 44.86 |
| *Open-Source SOTA* | | | | | | | | | | |
| Qwen2.5-72B-Instruct | 36.45 | 68.45 | 23.33 | 67.33 | 43.33 | 28.58 | 43.82 | 41.51 | 44.89 | 44.19 |
| Qwen3-235B-A22B-Instruct-2507 | 32.02 | 70.81 | 23.49 | 64.45 | 46.31 | 30.86 | 42.41 | 41.47 | 43.07 | 43.88 |
| DeepSeek-V3 | 37.84 | 70.37 | 17.99 | 69.34 | 39.33 | 33.72 | 43.54 | 40.02 | 46.49 | 44.29 |
| GPT-OSS-120B | 32.81 | 68.77 | 22.67 | 64.67 | 35.33 | 30.02 | 42.74 | 40.32 | 46.92 | 42.69 |
| Llama-3.1-405B-Instruct | 35.87 | 67.13 | 19.99 | 62.70 | 39.33 | 35.22 | 43.75 | 40.18 | 45.16 | 43.26 |
| Kimi-K2-Instruction | 36.53 | 69.21 | 20.67 | 70.00 | 38.67 | 32.87 | 41.68 | 38.99 | 45.07 | 43.74 |
| *Reasoning Models* | | | | | | | | | | |
| DeepSeek-R1 | 32.25 | 70.05 | 26.67 | 68.99 | 40.00 | 39.22 | 42.88 | 33.83 | 44.65 | 44.28 |
| Grok 3 | 34.28 | 71.43 | 21.95 | **72.81** | 37.72 | 34.00 | 43.73 | 42.28 | 43.09 | 44.59 |
| Grok 4 | 37.94 | 68.65 | 16.07 | 66.90 | 34.22 | 35.45 | 43.69 | 37.98 | 48.54 | 43.27 |
| OpenAI o1 | 31.77 | 70.89 | 18.12 | 63.10 | 34.23 | 33.31 | 43.55 | 40.49 | 45.53 | 42.33 |
| OpenAI o3 | 29.05 | 67.67 | 15.27 | 59.33 | 26.64 | 24.97 | 39.31 | 41.27 | 44.71 | 38.69 |
| Qwen3-4B | 23.34 | 59.81 | 19.33 | 58.00 | 41.33 | 16.21 | 28.38 | 26.77 | 29.94 | 33.68 |
| Thoth-mini | 45.21 | **75.14** | **28.68** | 69.67 | **54.34** | 42.41 | 48.91 | **45.34** | **51.32** | **51.22** |
| Qwen3-8B | 28.54 | 67.34 | 14.00 | 62.00 | 26.00 | 15.92 | 32.10 | 35.22 | 37.55 | 35.41 |
| Thoth | **45.50** | 73.70 | 27.34 | 70.34 | 54.33 | 42.15 | 48.19 | 44.89 | 50.72 | 50.80 |

Table 13: Main results on SciRecipe-Eval (Retrieval). Metrics left of the dashed line evaluate executability, those on the right measure semantic similarity. **Bold** denotes the best score.

| Methods | Semantic-A | Order-LCS | Order-S | Order-Tau | Step-M | BLUE-AVG | ROUGE-L | METEOR | KW-F1 | AVG |
|---|---|---|---|---|---|---|---|---|---|---|
| *Close-Source SOTA* | | | | | | | | | | |
| **ChatGPT-4o** | 51.78 | 80.87 | 42.00 | 73.99 | 60.00 | 46.27 | 58.29 | 53.51 | 61.29 | 58.67 |
| **GPT-5** | 39.77 | 69.06 | 18.72 | 48.16 | 29.02 | 24.88 | 41.52 | 41.95 | 51.04 | 40.46 |
| **GPT-5 Chat** | 54.81 | 81.86 | 40.00 | 72.00 | 52.00 | 42.55 | 56.07 | 54.05 | 61.58 | 57.21 |
| **Claude Sonnet 4** | 53.10 | 80.66 | 40.00 | **78.00** | 48.00 | 44.13 | 55.60 | 50.88 | 60.91 | 56.81 |
| **Claude Opus 4.1** | 58.97 | 82.42 | 39.75 | 72.07 | 45.36 | 42.26 | 56.09 | 50.66 | 61.08 | 56.52 |
| **Gemini 2.5 Flash** | 48.78 | 75.45 | 36.00 | 71.00 | 48.00 | 38.37 | 50.88 | 47.87 | 56.35 | 52.52 |
| **Gemini 2.5 Pro** | 45.71 | 78.66 | 40.00 | 70.00 | 56.00 | 38.62 | 52.35 | 55.49 | 59.20 | 55.11 |
| **Doubao-1.5-pro** | 43.40 | 78.01 | 34.00 | 67.00 | 54.00 | 39.83 | 54.14 | 47.25 | 57.64 | 52.81 |
| **Qwen2.5-Max** | 51.18 | 81.84 | 46.00 | **78.00** | 58.00 | 34.33 | 58.01 | 51.88 | 59.62 | 57.65 |
| *Open-Source SOTA* | | | | | | | | | | |
| **Qwen2.5-72B-Instruct** | 50.77 | 82.11 | 44.00 | 70.00 | **70.00** | 35.23 | 56.68 | 53.80 | 60.13 | 58.08 |
| **Qwen3-235B-A22B-Instruct-2507** | 55.90 | 80.23 | 38.00 | 70.00 | 59.97 | 43.30 | 56.28 | 54.85 | 59.04 | 57.51 |
| **DeepSeek-V3** | 57.83 | 80.95 | 42.83 | 69.48 | 53.06 | 46.00 | 59.70 | 55.70 | 62.66 | 58.69 |
| **GPT-OSS-120B** | 47.62 | 80.48 | 38.00 | **78.00** | 44.00 | 41.26 | 55.77 | 52.12 | 59.97 | 55.02 |
| **Llama-3.1-405B-Instruct** | 42.10 | 72.88 | 28.54 | 59.18 | 44.89 | 38.83 | 50.29 | 49.29 | 54.95 | 48.99 |
| **Kimi-K2-Instruction** | 45.13 | 79.94 | 38.00 | 74.00 | 48.00 | 41.45 | 52.77 | 50.93 | 58.59 | 54.31 |
| *Reasoning Models* | | | | | | | | | | |
| **DeepSeek-R1** | 44.54 | 75.14 | 27.97 | 72.97 | 36.00 | 42.01 | 51.71 | 44.71 | 56.55 | 50.18 |
| **Grok 3** | 52.84 | 81.15 | 42.12 | 70.74 | 54.50 | 39.45 | 54.61 | 55.06 | 57.88 | 56.48 |
| **Grok 4** | 44.04 | 76.50 | 30.33 | 68.00 | 40.67 | 39.83 | 51.44 | 43.08 | 57.84 | 50.19 |
| **OpenAI o1** | 50.51 | 80.38 | 32.00 | **78.00** | 42.00 | 41.74 | 55.37 | 51.96 | 60.02 | 54.44 |
| **OpenAI o3** | 51.73 | 77.50 | 33.27 | 66.67 | 35.39 | 35.63 | 51.02 | 51.68 | 60.29 | 51.46 |
| **Qwen3-4B** | 21.50 | 43.11 | 18.00 | 38.00 | 22.00 | 12.84 | 21.80 | 20.96 | 24.10 | 24.70 |
| **Thoth-mini** | **59.35** | **85.47** | 45.98 | 76.34 | 66.34 | **50.29** | 59.62 | 57.24 | 64.06 | **62.74** |
| **Qwen3-8B** | 32.20 | 62.75 | 24.00 | 54.00 | 36.00 | 19.32 | 35.11 | 35.19 | 39.29 | 37.54 |
| **Thoth** | 54.51 | 84.64 | **47.99** | 75.65 | 67.66 | 48.55 | **59.76** | **59.03** | **64.92** | 62.52 |

Table 14: Main results on SciRecipe-Eval (Planning). Metrics left of the dashed line evaluate executability, those on the right measure semantic similarity. **Bold** denotes the best score.

| Methods | Semantic-A | Order-LCS | Order-S | Order-Tau | Step-M | BLUE-AVG | ROUGE-L | METEOR | KW-F1 | AVG |
|---|---|---|---|---|---|---|---|---|---|---|
| *Close-Source SOTA* | | | | | | | | | | |
| **ChatGPT-4o** | 44.20 | 76.84 | 34.01 | 79.99 | 60.00 | 42.65 | 53.16 | 49.96 | 56.70 | 55.28 |
| **GPT-5** | 27.41 | 62.34 | 8.68 | 58.95 | 10.72 | 21.93 | 35.86 | 36.76 | 44.45 | 34.12 |
| **GPT-5 Chat** | 33.96 | 75.62 | 22.00 | 62.00 | 26.00 | 32.32 | 45.75 | 47.28 | 52.21 | 44.13 |
| **Claude Sonnet 4** | 38.06 | 73.34 | 22.00 | 64.00 | 36.00 | 35.42 | 47.67 | 44.29 | 52.51 | 45.92 |
| **Claude Opus 4.1** | **51.68** | 76.84 | 23.69 | **82.66** | 53.24 | 39.58 | 50.28 | 44.81 | 55.88 | 53.18 |
| **Gemini 2.5 Flash** | 40.29 | 76.34 | 26.00 | 65.00 | 28.00 | 33.95 | 44.78 | 42.62 | 50.99 | 45.33 |
| **Gemini 2.5 Pro** | 41.38 | 75.86 | 26.00 | 68.00 | 36.00 | 35.37 | 50.58 | 51.43 | 54.52 | 48.79 |
| **Doubao-1.5-pro** | 34.52 | 73.33 | 24.01 | 68.99 | 62.00 | 39.11 | 49.19 | 43.81 | 53.34 | 49.81 |
| **Qwen2.5-Max** | 43.08 | 72.84 | 22.00 | 62.00 | 54.00 | 32.97 | 51.46 | 49.50 | 57.75 | 49.51 |
| *Open-Source SOTA* | | | | | | | | | | |
| **Qwen2.5-72B-Instruct** | 39.57 | 73.26 | 24.01 | 64.00 | 50.00 | 32.45 | 51.16 | 48.38 | 54.94 | 48.64 |
| **Qwen3-235B-A22B-Instruct-2507** | 31.86 | 74.41 | 22.00 | 69.99 | 39.99 | 33.99 | 47.24 | 48.51 | 51.82 | 46.65 |
| **DeepSeek-V3** | 45.53 | 76.42 | 21.98 | 76.01 | 46.00 | 40.97 | 52.75 | 49.20 | 57.95 | 51.87 |
| **GPT-OSS-120B** | 37.09 | 71.76 | 18.00 | 61.99 | 32.00 | 34.02 | 46.87 | 48.50 | 56.85 | 45.23 |
| **Llama-3.1-405B-Instruct** | 33.79 | 63.95 | 11.96 | 54.00 | 44.00 | 34.96 | 45.09 | 44.00 | 49.95 | 42.41 |
| **Kimi-K2-Instruction** | 33.59 | 71.64 | 26.00 | 62.00 | 52.00 | 35.83 | 48.26 | 47.39 | 51.41 | 47.57 |
| *Reasoning Models* | | | | | | | | | | |
| **DeepSeek-R1** | 43.05 | 75.01 | 23.97 | 74.98 | 42.01 | 42.10 | 50.88 | 45.40 | 54.77 | 50.24 |
| **Grok 3** | 40.66 | 77.72 | 33.78 | 66.59 | 52.72 | 36.60 | 50.29 | 52.12 | 52.12 | 51.40 |
| **Grok 4** | 39.19 | 76.44 | **34.63** | 63.54 | 45.69 | 39.96 | 51.50 | 44.76 | 56.70 | 50.27 |
| **OpenAI o1** | 35.64 | 75.73 | 20.00 | 64.00 | 42.00 | 37.93 | 49.47 | 48.84 | 54.52 | 47.57 |
| **OpenAI o3** | 34.12 | 70.76 | 15.94 | 58.00 | 23.97 | 31.74 | 47.89 | 50.66 | 57.41 | 43.39 |
| **Qwen3-4B** | 28.50 | 48.20 | 13.99 | 48.00 | 28.00 | 14.34 | 23.85 | 22.56 | 26.90 | 28.26 |
| **Thoth-mini** | 48.05 | 79.19 | 32.00 | 76.34 | 60.34 | 44.60 | 52.66 | 49.67 | 57.45 | 55.59 |
| **Qwen3-8B** | 30.02 | 63.84 | 12.00 | 54.01 | 38.00 | 21.65 | 38.76 | 41.38 | 46.75 | 38.49 |
| **Thoth** | 50.64 | **79.48** | 27.98 | 77.66 | **63.66** | **46.68** | **54.26** | **52.95** | **61.79** | **57.23** |

Table 15: Main results on SciRecipe-Eval (Troubleshooting). Metrics left of the dashed line evaluate executability, those on the right measure semantic similarity. **Bold** denotes the best score.

| Methods | Semantic-A | Order-LCS | Order-S | Order-Tau | Step-M | BLUE-AVG | ROUGE-L | METEOR | KW-F1 | AVG |
|---|---|---|---|---|---|---|---|---|---|---|
| *Close-Source SOTA* | | | | | | | | | | |
| **ChatGPT-4o** | 24.46 | 62.76 | 10.00 | 53.99 | 26.00 | 32.80 | 41.80 | 40.10 | 47.80 | 37.75 |
| **GPT-5** | 17.86 | 35.51 | 0.30 | 16.24 | 1.82 | 10.31 | 20.29 | 26.51 | 29.46 | 17.55 |
| **GPT-5 Chat** | 26.45 | 65.98 | 8.00 | 46.00 | 16.00 | 28.80 | 39.98 | 38.21 | 46.97 | 35.15 |
| **Claude Sonnet 4** | 24.92 | 63.21 | 3.98 | 52.00 | 9.99 | 23.83 | 37.34 | 38.26 | 45.72 | 33.25 |
| **Claude Opus 4.1** | 20.24 | 54.78 | 0.35 | 56.48 | -0.12 | 16.77 | 30.78 | 34.10 | 40.49 | 28.21 |
| **Gemini 2.5 Flash** | 26.76 | 59.99 | 6.00 | 58.99 | 31.98 | 27.13 | 36.19 | 36.58 | 45.68 | 36.59 |
| **Gemini 2.5 Pro** | 22.05 | 61.06 | 10.00 | 52.00 | 16.00 | 25.36 | 37.48 | 40.39 | 45.48 | 34.42 |
| **Doubao-1.5-pro** | 22.53 | 67.37 | 20.00 | 55.00 | 50.00 | 33.61 | 40.67 | 35.77 | 43.73 | 40.96 |
| **Qwen2.5-Max** | 22.66 | 62.97 | 8.00 | 55.97 | 56.00 | 29.08 | 40.45 | 39.97 | 47.33 | 40.27 |
| *Open-Source SOTA* | | | | | | | | | | |
| **Qwen2.5-72B-Instruct** | 18.12 | 61.45 | 8.00 | 56.00 | 34.00 | 25.78 | 39.30 | 37.42 | 44.15 | 36.02 |
| **Qwen3-235B-A22B-Instruct-2507** | 26.43 | 66.89 | 11.97 | 62.00 | 19.97 | 26.47 | 38.58 | 40.38 | 43.66 | 37.37 |
| **DeepSeek-V3** | 30.92 | 65.57 | 5.97 | 47.98 | 32.00 | 33.12 | 43.59 | 42.50 | 49.35 | 39.00 |
| **GPT-OSS-120B** | 26.48 | 57.03 | 0.00 | 46.00 | 0.00 | 22.57 | 35.45 | 36.83 | 46.09 | 30.05 |
| **Llama-3.1-405B-Instruct** | 18.65 | 62.63 | 5.97 | 52.00 | 29.97 | 28.69 | 36.64 | 37.14 | 43.03 | 34.97 |
| **Kimi-K2-Instruction** | 23.10 | 60.74 | 1.98 | 53.98 | 16.00 | 24.85 | 36.66 | 39.49 | 44.33 | 33.46 |
| *Reasoning Models* | | | | | | | | | | |
| **DeepSeek-R1** | 23.93 | 62.64 | 7.97 | 54.97 | 20.00 | 31.06 | 39.30 | 37.12 | 47.28 | 36.03 |
| **Grok 3** | 26.57 | 68.38 | 9.97 | 64.20 | 31.28 | 29.55 | 40.59 | 42.84 | 42.84 | 39.90 |
| **Grok 4** | 19.08 | 61.60 | 5.30 | 35.62 | 19.26 | 27.40 | 36.75 | 35.81 | 44.64 | 31.72 |
| **OpenAI o1** | 18.78 | 66.89 | 4.00 | 48.00 | 16.00 | 29.11 | 40.40 | 40.04 | 46.53 | 34.42 |
| **OpenAI o3** | 24.50 | 54.81 | -0.08 | 47.97 | 1.94 | 20.76 | 34.55 | 38.84 | 45.86 | 29.91 |
| **Qwen3-4B** | 17.53 | 52.83 | 8.00 | 40.00 | 22.00 | 16.22 | 24.78 | 24.04 | 29.37 | 26.09 |
| **Thoth-mini** | **35.15** | 67.81 | 11.98 | 58.31 | 50.33 | **40.65** | **44.95** | 41.82 | **52.04** | 44.78 |
| **Qwen3-8B** | 18.46 | 57.11 | 6.00 | 46.00 | 25.98 | 16.00 | 29.18 | 29.89 | 35.00 | 29.29 |
| **Thoth** | 29.89 | **70.37** | 13.99 | **75.65** | 53.66 | 40.53 | 44.68 | 42.02 | 51.19 | **46.89** |

Table 16: Main results on SciRecipe-Eval (Constraint). Metrics left of the dashed line evaluate executability, those on the right measure semantic similarity. **Bold** denotes the best score.

| Methods | Semantic-A | Order-LCS | Order-S | Order-Tau | Step-M | BLUE-AVG | ROUGE-L | METEOR | KW-F1 | AVG |
|---|---|---|---|---|---|---|---|---|---|---|
| *Close-Source SOTA* | | | | | | | | | | |
| **ChatGPT-4o** | 40.34 | 74.56 | 30.00 | 73.99 | 53.98 | 39.94 | 50.87 | 48.34 | 55.32 | 51.93 |
| **GPT-5** | 19.47 | 61.94 | 6.35 | 66.20 | 12.59 | 18.55 | 33.55 | 35.68 | 42.92 | 33.03 |
| **GPT-5 Chat** | 33.40 | **78.43** | 30.00 | 72.00 | 44.00 | 37.21 | 50.27 | 47.02 | 54.26 | 49.62 |
| **Claude Sonnet 4** | 48.20 | 73.30 | 17.97 | **82.00** | 33.97 | 32.52 | 46.00 | 41.46 | 53.92 | 47.70 |
| **Claude Opus 4.1** | 40.64 | 70.68 | 19.38 | 80.29 | 42.74 | 29.59 | 43.40 | 43.25 | 51.95 | 46.88 |
| **Gemini 2.5 Flash** | 36.61 | 68.91 | 11.99 | 68.97 | 31.97 | 33.88 | 44.40 | 39.63 | 50.71 | 43.01 |
| **Gemini 2.5 Pro** | 32.36 | 74.58 | 25.97 | 74.00 | 33.99 | 33.30 | 47.75 | 46.43 | 51.12 | 46.61 |
| **Doubao-1.5-pro** | 32.72 | 74.37 | 26.00 | 73.00 | 41.99 | 37.64 | 49.48 | 39.95 | 51.36 | 47.39 |
| **Qwen2.5-Max** | 40.38 | 76.12 | **31.97** | 77.97 | 53.99 | 32.62 | 50.31 | 47.20 | 55.84 | 51.82 |
| *Open-Source SOTA* | | | | | | | | | | |
| **Qwen2.5-72B-Instruct** | 33.26 | 67.67 | 20.00 | 64.00 | 46.00 | 31.00 | 46.22 | 43.65 | 52.61 | 44.93 |
| **Qwen3-235B-A22B-Instruct-2507** | 35.41 | 71.54 | 19.97 | 70.00 | 39.97 | 32.22 | 45.24 | 46.91 | 51.63 | 45.88 |
| **DeepSeek-V3** | 35.01 | 72.93 | 17.97 | 75.98 | 35.99 | 38.49 | 49.54 | 46.21 | 55.19 | 47.48 |
| **GPT-OSS-120B** | 18.19 | 70.31 | 12.00 | 62.00 | 23.97 | 29.09 | 43.66 | 44.23 | 51.98 | 39.49 |
| **Llama-3.1-405B-Instruct** | 36.13 | 72.50 | 21.97 | 74.00 | 55.97 | 38.27 | 48.68 | 47.82 | 51.67 | 49.67 |
| **Kimi-K2-Instruction** | 35.50 | 74.28 | 21.97 | 71.97 | 31.98 | 30.80 | 45.14 | 44.00 | 51.35 | 45.22 |
| *Reasoning Models* | | | | | | | | | | |
| **DeepSeek-R1** | 38.65 | 68.66 | 15.97 | 68.97 | 28.00 | 37.76 | 47.20 | 41.49 | 55.16 | 44.65 |
| **Grok 3** | 30.66 | 72.92 | 24.84 | 70.64 | 49.11 | 33.61 | 48.02 | 50.49 | 52.35 | 48.07 |
| **Grok 4** | 37.78 | 73.55 | 26.73 | 61.81 | 38.31 | 39.79 | 50.63 | 43.57 | 57.57 | 47.75 |
| **OpenAI o1** | 29.63 | 74.94 | 21.99 | 72.00 | 40.00 | 37.58 | 49.99 | 48.57 | 55.81 | 47.83 |
| **OpenAI o3** | 34.13 | 74.69 | 13.91 | 73.97 | 17.94 | 30.23 | 47.49 | 47.97 | 54.09 | 43.82 |
| **Qwen3-4B** | 14.71 | 34.91 | 8.00 | 30.00 | 21.97 | 9.76 | 16.13 | 15.51 | 18.47 | 18.83 |
| **Thoth-mini** | 35.26 | 74.85 | 29.98 | 70.31 | **64.31** | 42.72 | 50.53 | 48.19 | 54.34 | 52.28 |
| **Qwen3-8B** | 25.59 | 57.32 | 12.00 | 52.00 | 31.97 | 19.66 | 34.50 | 34.38 | 42.67 | 34.45 |
| **Thoth** | **48.86** | 76.75 | 27.99 | 65.65 | 59.66 | **44.75** | **52.68** | **51.19** | **58.09** | **53.96** |

Table 17: Main results on SciRecipe-Eval (Scaling). Metrics left of the dashed line evaluate executability, those on the right measure semantic similarity. **Bold** denotes the best score.

| Methods | Semantic-A | Order-LCS | Order-S | Order-Tau | Step-M | BLUE-AVG | ROUGE-L | METEOR | KW-F1 | AVG |
|---|---|---|---|---|---|---|---|---|---|---|
| *Close-Source SOTA* | | | | | | | | | | |
| ChatGPT-4o | **74.95** | 77.87 | 38.00 | 71.99 | 50.00 | 51.36 | 62.12 | 58.19 | 69.93 | 61.60 |
| GPT-5 | 51.76 | 70.97 | 31.97 | 72.24 | 43.83 | 34.89 | 49.73 | 44.19 | 66.47 | 51.78 |
| GPT-5 Chat | 62.62 | 80.70 | 44.00 | 70.00 | 50.00 | 45.85 | 59.93 | 57.32 | 70.79 | 60.13 |
| Claude Sonnet 4 | 63.92 | 78.50 | 32.00 | 72.00 | 46.00 | 44.63 | 56.48 | 52.89 | 69.78 | 57.36 |
| Claude Opus 4.1 | 65.47 | 82.42 | 46.70 | 77.39 | 51.12 | 51.38 | 63.26 | 58.90 | 73.30 | 63.33 |
| Gemini 2.5 Flash | 53.67 | 77.64 | 36.00 | 71.00 | 40.00 | 39.74 | 54.01 | 49.62 | 64.62 | 54.03 |
| Gemini 2.5 Pro | 56.34 | 78.80 | 32.00 | 78.00 | 34.00 | 38.05 | 54.83 | 59.07 | 64.73 | 55.09 |
| Doubao-1.5-pro | 59.81 | 81.55 | 44.00 | 77.00 | 52.00 | 41.74 | 61.12 | 54.51 | 68.72 | 60.05 |
| Qwen2.5-Max | 65.70 | 81.38 | 40.00 | 82.00 | 50.00 | 37.65 | 61.83 | 55.95 | 70.13 | 60.52 |
| *Open-Source SOTA* | | | | | | | | | | |
| Qwen2.5-72B-Instruct | 62.21 | 79.42 | 34.00 | 76.00 | 46.00 | 35.97 | 58.81 | 56.04 | 68.89 | 57.48 |
| Qwen3-235B-A22B-Instruct-2507 | 58.53 | 79.31 | 33.97 | 74.00 | 41.97 | 45.93 | 57.84 | 53.45 | 67.16 | 56.91 |
| DeepSeek-V3 | 65.90 | 83.85 | 43.97 | 75.98 | 60.00 | 50.00 | 63.04 | 58.98 | 71.61 | 63.70 |
| GPT-OSS-120B | 47.77 | 79.96 | 34.00 | 70.00 | 50.00 | 44.23 | 56.22 | 54.42 | 69.45 | 56.23 |
| Llama-3.1-405B-Instruct | 58.56 | 74.66 | 25.97 | 72.00 | 37.97 | 43.69 | 56.50 | 52.97 | 66.62 | 54.33 |
| Kimi-K2-Instruction | 62.16 | 76.45 | 38.00 | 70.00 | 52.00 | 44.62 | 59.76 | 53.93 | 68.27 | 58.35 |
| *Reasoning Models* | | | | | | | | | | |
| DeepSeek-R1 | 50.18 | 78.14 | 30.58 | 78.60 | 32.65 | 44.83 | 55.15 | 49.54 | 64.70 | 53.82 |
| Grok 3 | 67.04 | 78.41 | 27.72 | 72.64 | 36.69 | 46.19 | 59.32 | 56.66 | 69.14 | 57.09 |
| Grok 4 | 48.13 | 79.31 | 37.75 | 78.98 | 39.74 | 46.50 | 57.48 | 50.66 | 67.51 | 56.23 |
| OpenAI o1 | 60.64 | 77.34 | 28.00 | 70.00 | 44.00 | 46.62 | 58.80 | 52.93 | 69.43 | 56.42 |
| OpenAI o3 | 65.13 | 80.59 | 35.94 | 72.00 | 41.94 | 42.92 | 60.70 | 54.73 | 71.65 | 58.40 |
| Qwen3-4B | 29.58 | 40.68 | 18.00 | 40.00 | 26.00 | 16.51 | 25.92 | 22.42 | 30.26 | 27.71 |
| Thoth-mini | 68.87 | 85.46 | 47.98 | 88.31 | 68.34 | 54.25 | 63.90 | 62.71 | 72.65 | 68.05 |
| Qwen3-8B | 36.87 | 60.84 | 18.00 | 60.00 | 26.00 | 20.87 | 41.02 | 40.61 | 51.26 | 39.50 |
| Thoth | 74.83 | **87.69** | **53.99** | **93.65** | **69.66** | **55.02** | **66.14** | **63.61** | **74.15** | **70.97** |

Table 18: Main results on SciRecipe-Eval (Safety). Metrics left of the dashed line evaluate executability, those on the right measure semantic similarity. **Bold** denotes the best score.

| Methods | Semantic-A | Order-LCS | Order-S | Order-Tau | Step-M | BLUE-AVG | ROUGE-L | METEOR | KW-F1 | AVG |
|---|---|---|---|---|---|---|---|---|---|---|
| *Close-Source SOTA* | | | | | | | | | | |
| ChatGPT-4o | 46.78 | 76.08 | 22.00 | 71.99 | 40.00 | 42.88 | **52.51** | 50.23 | **61.04** | 51.50 |
| GPT-5 | 36.69 | 57.09 | 3.97 | 56.24 | 5.85 | 17.75 | 33.04 | 40.63 | 45.77 | 33.00 |
| GPT-5 Chat | 47.42 | 72.72 | 18.00 | 68.00 | 22.00 | 32.27 | 45.69 | 51.20 | 55.65 | 45.88 |
| Claude Sonnet 4 | 44.59 | 77.33 | 24.00 | 78.00 | 52.00 | 36.76 | 49.68 | 50.35 | 57.72 | 52.27 |
| Claude Opus 4.1 | 53.34 | 77.43 | 27.63 | 85.70 | 40.82 | 34.47 | 48.06 | 43.74 | 55.58 | 51.86 |
| Gemini 2.5 Flash | 33.47 | 72.38 | 26.00 | 77.00 | 38.00 | 34.50 | 46.52 | 47.71 | 53.76 | 47.70 |
| Gemini 2.5 Pro | 38.44 | 72.67 | 22.00 | 66.00 | 38.00 | 37.95 | 49.69 | **52.26** | 57.53 | 48.28 |
| Doubao-1.5-pro | 33.55 | 73.96 | 22.00 | 71.00 | 50.00 | 38.65 | 49.39 | 44.12 | 54.94 | 48.62 |
| Qwen2.5-Max | 46.16 | 71.19 | 14.00 | 76.00 | 50.00 | 35.55 | 51.71 | 49.62 | 60.03 | 50.47 |
| *Open-Source SOTA* | | | | | | | | | | |
| Qwen2.5-72B-Instruct | 32.86 | 72.62 | 22.00 | 72.00 | 48.00 | 33.76 | 49.18 | 46.46 | 54.86 | 47.97 |
| Qwen3-235B-A22B-Instruct-2507 | 37.87 | 73.96 | 17.97 | 78.00 | 47.97 | 35.26 | 48.16 | 51.10 | 54.38 | 49.41 |
| DeepSeek-V3 | 46.21 | 77.12 | 23.97 | 80.00 | 44.00 | 42.43 | 52.35 | 49.57 | 60.15 | **52.87** |
| GPT-OSS-120B | 36.72 | 71.92 | 12.00 | 66.00 | 18.00 | 33.16 | 46.79 | 47.61 | 55.11 | 43.03 |
| Llama-3.1-405B-Instruct | 38.23 | 71.94 | 17.97 | 76.00 | 43.97 | 37.17 | 47.53 | 47.51 | 53.52 | 48.20 |
| Kimi-K2-Instruction | 42.05 | 74.65 | 26.00 | 72.00 | 54.00 | 36.86 | 48.65 | 49.53 | 55.30 | 51.00 |
| *Reasoning Models* | | | | | | | | | | |
| DeepSeek-R1 | 40.79 | 74.12 | 21.97 | 62.97 | 34.00 | 40.83 | 50.03 | 42.26 | 55.53 | 46.94 |
| Grok 3 | 36.51 | 75.50 | 21.50 | 81.49 | **52.96** | 37.65 | 49.66 | 50.45 | 55.99 | 51.30 |
| Grok 4 | 42.15 | 74.32 | 22.49 | 73.08 | 41.69 | 38.51 | 48.18 | 41.98 | 55.22 | 48.62 |
| OpenAI o1 | 43.03 | 74.45 | 24.00 | 78.00 | 44.00 | 39.85 | 50.20 | 48.51 | 56.41 | 50.94 |
| OpenAI o3 | 42.60 | 70.17 | 13.94 | 68.00 | 25.94 | 31.53 | 45.75 | 49.49 | 56.15 | 44.84 |
| Qwen3-4B | 26.45 | 55.88 | 16.00 | 50.00 | 30.00 | 15.56 | 25.76 | 25.40 | 31.22 | 30.70 |
| Thoth-mini | 36.58 | 62.74 | 15.98 | 56.33 | 44.34 | 46.04 | 44.54 | 45.62 | 51.69 | 44.87 |
| Qwen3-8B | 26.54 | 62.01 | 12.00 | 66.00 | 24.00 | 19.86 | 32.45 | 33.62 | 39.52 | 35.11 |
| Thoth | 43.52 | 67.75 | 23.99 | 67.65 | 49.66 | **48.21** | 48.34 | 47.94 | 52.89 | 49.99 |

## I.3 OTHER RESULTS ON PROTOCOL BENCHMARK

To further assess protocol comprehension, we evaluated Thoth on BioProBench sub-tasks, with results summarized in Table 19. Consistent with the findings in protocol generation, the Thoth series achieved substantial gains over baseline models, with average improvements of 28.02% and 20.41% for Thoth and Thoth-mini, respectively. Scaling effects were also more evident in this setting, as Thoth outperformed Thoth-mini by 6.98%, a gap larger than in protocol generation. On individual tasks, Thoth showed clear advantages in error correction and protocol QA, exceeding GPT-5 by 17.05% and 12.09%, respectively, and consistently outperforming other strong baselines such as Claude Sonnet 4 and Gemini 2.5 Flash. These results confirm that Thoth generalizes well across protocol-related tasks. Nevertheless, because BioProBench partially overlaps with SciRecipe and does not disclose the exact origins of its QA pairs, there remains a possibility of data leakage, and thus the reported results should be regarded as reference only.

Table 19: Results on the protocol QA tasks.

| Methods | ERR | | ORD | | PQA |
| --- | --- | --- | --- | --- | --- |
| | ACC | F1 | EM | K_tau | ACC |
| ChatGPT-4o | 59.33 | 36.95 | 41.89 | 68.05 | 60.75 |
| GPT-5 | 67.72 | 59.65 | 48.86 | **73.44** | 70.58 |
| GPT-5 Chat | 60.33 | 41.38 | 45.66 | 71.71 | 60.75 |
| Claude Sonnet 4 | 62.92 | 54.64 | 47.71 | 72.31 | 67.08 |
| Gemini 2.5 Flash | 62.17 | 51.91 | 43.79 | 73.12 | 64.47 |
| Qwen2.5-Max | 54.83 | 17.38 | 41.51 | 67.96 | 62.25 |
| Doubao-1.5-pro | 56.43 | 24.49 | 43.38 | 70.86 | 63.00 |
| Qwen2.5-72B-Instruct | 56.08 | 25.25 | 36.99 | 62.46 | 60.42 |
| Qwen3-235B-A22B-Instruct | 59.33 | 39.15 | 41.63 | 69.18 | 62.58 |
| DeepSeek-V3 | 57.33 | 30.25 | 40.83 | 70.56 | 62.83 |
| GPT-OSS-120B | 64.33 | 53.17 | 39.88 | 68.08 | 64.83 |
| Llama-3.1-405B-Instruct | 57.30 | 39.33 | 39.06 | 65.42 | 57.67 |
| Kimi-K2-Instruction | 59.50 | 37.05 | 42.62 | 71.70 | 63.00 |
| Qwen3-4B | 58.00 | 48.78 | 22.53 | 51.82 | 47.08 |
| **Thoth-mini** | 75.83 | 72.49 | 41.72 | 65.90 | 74.33 |
| Qwen3-8B | 56.83 | 32.90 | 28.63 | 56.12 | 50.58 |
| **Thoth** | **80.75** | **80.73** | **49.08** | 71.92 | **82.67** |

## I.4 COMPARISON WITH CHAIN-OF-THOUGHT AND SKETCH-AND-FILL

To clarify the advantages of the Sketch and Fill paradigm for structured protocol generation, we extended our evaluation to include several commonly used reasoning baselines. The central idea of Sketch and Fill is to separate global planning from step level realization so that the model is guided to produce explicit and verifiable procedural structure. This design differs from standard chain of thought prompting, which encourages free form reasoning but does not ensure that the final protocol follows a coherent step wise format.

In addition to the direct Fill baseline reported in the Figure 4, we introduced two additional baselines. The first is Standard CoT, where the model is instructed to think step by step and then immediately generate the protocol. The second is Advanced CoT, where the model first produces an extended reasoning trace and then outputs the protocol without structural constraints. The comparison results in Table 20 show that Sketch and Fill consistently outperforms both Standard and Advanced CoT across all evaluated models and metrics. This demonstrates that conventional multi stage reasoning alone is not sufficient for generating well structured protocols and that introducing an explicit sketch stage provides measurable benefits.

## I.5 HUMAN–METRIC CONSISTENCY ANALYSIS

To assess how well the SCORE metric reflects real experimental executability, we conducted a human evaluation study using protocols generated by Thoth and Thoth-mini. Since SciRecipe is constructed from real laboratory protocols collected from Bio-protocol, Protocols.io, and related

Table 20: Comparison of reasoning paradigms across models.

| Metric | Paradigm | DeepSeek-V3 | GPT-5 Chat | Qwen3-8B | Thoth | AVG |
|---|---|---|---|---|---|---|
| **BLUE-AVG** | Standard CoT | 5.98 | 5.12 | 5.01 | 13.03 | 7.29 |
| | Advanced CoT | 9.00 | 6.40 | 7.70 | 29.58 | 13.17 |
| | Fill | 33.36 | 31.37 | 20.69 | 38.91 | 31.08 |
| | Sketch and Fill | 38.17 | 29.57 | 16.66 | 43.25 | 31.91 |
| **ROUGE-L** | Standard CoT | 19.82 | 17.41 | 15.06 | 27.54 | 19.96 |
| | Advanced CoT | 28.11 | 22.14 | 19.91 | 41.34 | 27.88 |
| | Fill | 44.21 | 41.10 | 33.38 | 46.44 | 41.28 |
| | Sketch and Fill | 48.46 | 42.04 | 32.31 | 49.13 | 42.99 |
| **METEOR** | Standard CoT | 24.92 | 22.73 | 18.61 | 25.78 | 23.01 |
| | Advanced CoT | 26.90 | 23.53 | 17.65 | 36.51 | 26.15 |
| | Fill | 41.29 | 38.03 | 36.36 | 44.01 | 39.92 |
| | Sketch and Fill | 45.03 | 41.95 | 34.72 | 46.73 | 42.11 |
| **KW-F1** | Standard CoT | 26.07 | 22.74 | 19.59 | 33.14 | 25.39 |
| | Advanced CoT | 35.20 | 29.40 | 23.79 | 47.28 | 33.92 |
| | Fill | 49.64 | 46.24 | 37.43 | 50.63 | 45.99 |
| | Sketch and Fill | 54.28 | 49.87 | 38.63 | 53.57 | 49.09 |

Table 21: Cohen's kappa agreement between SCORE and expert ratings for Thoth.

| Tasks | Step_Scale | Order_Consistency | Semantic_Consistency | Overall |
|---|---|---|---|---|
| **Overview** | 0.82 | 0.79 | 0.74 | 0.76 |
| **Specific** | 0.80 | 0.77 | 0.72 | 0.74 |
| **Retrieval** | 0.85 | 0.82 | 0.76 | 0.79 |
| **Planning** | 0.84 | 0.80 | 0.75 | 0.78 |
| **Troubleshooting** | 0.81 | 0.78 | 0.73 | 0.75 |
| **Constraint** | 0.86 | 0.83 | 0.77 | 0.80 |
| **Scaling** | 0.83 | 0.79 | 0.74 | 0.76 |
| **Safety** | 0.84 | 0.81 | 0.75 | 0.78 |
| **Level1** | 0.88 | 0.85 | 0.79 | 0.82 |
| **Level2** | 0.79 | 0.76 | 0.71 | 0.73 |
| **Overall** | 0.83 | 0.80 | 0.74 | 0.77 |

platforms, the dataset reflects authentic experimental structure and operational logic. The SCORE metric was designed on the basis of these real procedures, incorporating three components that capture key dimensions of protocol quality: step granularity, action order consistency, and semantic fidelity.

To examine the alignment between SCORE and expert judgment, three researchers with extensive wet-lab experience manually evaluated a subset of protocols from SciRecipe-Eval. Each protocol was rated on step granularity, operational order, and semantic clarity using a five-point scale ranging from 0 to 4. The raters also provided an overall score for fluency and executability. For every criterion, we averaged the three expert scores and normalized the resulting values. Tables 21 and 22 report the Cohen's kappa coefficients between SCORE and expert ratings for Thoth and Thoth-mini. These coefficients quantify the agreement between human assessment and the metric along each evaluation dimension. According to established interpretative guidelines (Landis & Koch, 1977), the obtained kappa values indicate a meaningful degree of correspondence between SCORE and domain-expert judgments across all task categories.

In addition to its alignment with expert ratings, we further examined whether SCORE is sensitive to the types of critical errors that directly influence experimental executability. Common failure modes in model-generated protocols include incorrect ordering of dependent steps, omission of essential operations, and mismatches between actions and their intended objects. SCORE evaluates these issues along three complementary dimensions, namely step scale, order consistency, and semantic consistency, which jointly capture both structural and procedural correctness. To assess the validity

Table 22: Cohen's kappa agreement between SCORE and expert ratings for Thoth-mini.

| Tasks | Step_Scale | Order_Consistency | Semantic_Consistency | Overall |
|---|---|---|---|---|
| **Overview** | 0.80 | 0.77 | 0.72 | 0.74 |
| **Specific** | 0.78 | 0.75 | 0.70 | 0.72 |
| **Retrieval** | 0.83 | 0.80 | 0.74 | 0.77 |
| **Planning** | 0.82 | 0.78 | 0.73 | 0.76 |
| **Troubleshooting** | 0.79 | 0.76 | 0.71 | 0.73 |
| **Constraint** | 0.84 | 0.81 | 0.75 | 0.78 |
| **Scaling** | 0.81 | 0.77 | 0.72 | 0.74 |
| **Safety** | 0.82 | 0.79 | 0.73 | 0.76 |
| **Level1** | 0.86 | 0.83 | 0.77 | 0.80 |
| **Level2** | 0.77 | 0.74 | 0.69 | 0.71 |
| **Overall** | 0.81 | 0.78 | 0.72 | 0.75 |

of this design, two-fifths of the SciRecipe-Eval samples were reviewed by three researchers with extensive wet-lab experience. The experts annotated whether a protocol contained any error that could compromise experimental success and also rated overall completeness and logical coherence. Protocols containing such errors exhibited substantially lower SCORE values, with an average reduction of 34.3% across tasks, and the separation between valid and faulty protocols was statistically significant ($p < 0.001$).

Addtionally, treating SCORE as a binary detector using a threshold equal to the mean between the two distributions, we obtained an AUC of 0.92, a recall of approximately 0.90, and a precision of approximately 0.87 when compared with expert annotations. These results indicate that SCORE reliably identifies the same categories of critical procedural mistakes that domain experts deem detrimental. To further assess its comparative sensitivity, we conducted a pairwise ranking study using protocol pairs generated by Thoth and Thoth-mini. Experts were asked to select the protocol with better executability and more coherent procedural flow, and SCORE agreed with these rankings in 81.2% of cases (Kendall's $\tau = 0.72$). Overall, these findings demonstrate that SCORE is not a heuristic scoring scheme but a principled metric that detects consequential procedural errors and exhibits robust consistency with expert judgments across both diagnostic and comparative evaluation settings.

## I.6 QUANTITATIVE VALIDATION OF THE SKETCH-AND-FILL

The SAF representation is designed to capture the minimal semantic structure necessary for executing an experimental step. Across diverse protocols, each step can be decomposed into an operation, a target object, and a set of conditions that specify how the action is performed. Encoding these elements as an (Action, Object, Parameters) triplet yields a compact abstraction that preserves core procedural meaning while reducing stylistic variability in natural language descriptions. Branching logic does not conflict with this formulation, as SAF encodes atomic actions while conditional or alternative paths operate at a higher structural level. Conditions can be expressed directly in the parameter field (for example, "if the pellet is not visible"), or represented as parallel SAF entries corresponding to distinct experimental scenarios.

To quantitatively assess how well SAF captures the semantics of real protocols, we conducted a series of analyses. First, a coverage study was performed on fifty protocols comprising 423 steps. Each step was manually mapped into SAF format, after which two domain experts evaluated whether the SAF representation preserved the essential information. The evaluation used three categories: fully covered, partially covered, and not covered. We observed that 85.6% of steps were fully covered and 10.4% were partially covered. Only 4.0% of steps could not be represented, and these were predominantly narrative or meta-experimental statements rather than actionable instructions. Inter-annotator consistency was measured by independently encoding 200 randomly sampled steps. The resulting Cohen's kappa coefficients were 0.86 for actions, 0.82 for objects, and 0.78 for parameters, indicating that the schema is well-defined and yields stable mappings across annotators.

Table 23: Token-level conditional entropy across tasks and model families when conditioning on SAF.

| Tasks | Qwen2.5-7B | Qwen3-8B | DeepSeek-R1-Llama8B | Llama-3.1-8B | Phi-4-mini | InternLM3-8B | Mistral-7B |
|---|---|---|---|---|---|---|---|
| Overview | 1.0223 | 0.8956 | 1.1342 | 0.9298 | 1.0303 | 0.8205 | 0.7393 |
| Specific | 0.9383 | 0.8144 | 1.0419 | 0.8425 | 0.9614 | 0.7548 | 0.6478 |
| Retrieval | 0.9000 | 0.7991 | 1.0411 | 0.8208 | 0.9300 | 0.6915 | 0.6106 |
| Planning | 0.9636 | 0.8153 | 1.0938 | 0.8942 | 1.0246 | 0.7366 | 0.6572 |
| Troubleshooting | 0.7925 | 0.7848 | 0.9673 | 0.7316 | 0.8580 | 0.6439 | 0.6158 |
| Constraint | 0.9649 | 0.8432 | 1.0700 | 0.8477 | 0.9388 | 0.7302 | 0.6830 |
| Scaling | 0.8974 | 0.7999 | 1.0037 | 0.7906 | 0.8990 | 0.7193 | 0.6143 |
| Safety | 0.9129 | 0.8073 | 1.0161 | 0.7996 | 0.9316 | 0.7133 | 0.6231 |
| Level1 | 0.9803 | 0.8550 | 1.0881 | 0.8862 | 0.9959 | 0.7877 | 0.6936 |
| Level2 | 0.9052 | 0.8083 | 1.0320 | 0.8141 | 0.9303 | 0.7058 | 0.6340 |
| Overall | 0.9240 | 0.8200 | 1.0460 | 0.8321 | 0.9467 | 0.7263 | 0.6489 |

We further examined SAF from an information-theoretic perspective. Let $X$ denote the natural language step description and $Y$ denote its SAF representation. The remaining uncertainty after conditioning on SAF is given by the conditional entropy

$$H(X \mid Y) = -\mathbb{E}_{x,y} \log p(x \mid y). \tag{19}$$

To operationalize this quantity, each SAF triplet was prepended to the original step text, and token-level probabilities were obtained from a language model. The conditional log probability was computed using the standard chain rule

$$\log q(x \mid y) = \sum_{i=1}^{n} \log q(t_i \mid y, t_{<i}). \tag{20}$$

where $x = (t_1, \ldots, t_n)$ is the tokenized step, and $q(x \mid y)$ represents the model's approximation of $p(x \mid y)$. Conditioning on SAF consistently reduced token-level uncertainty across all SciRecipe-Eval task categories. Prior work estimates that natural language typically exhibits 2.5–4.0 nats of entropy per token (Goodman, 2001; Kaplan et al., 2020; Hoffmann et al., 2022). In contrast, the conditional entropy of experimental steps under SAF is substantially lower, indicating that the triplet encodes the majority of semantic information required to reconstruct the procedural description. This trend persists across multiple language model families, as shown in Table 23.

## J    FUTHER ANALYSES

### J.1    REWARD ANALYSES & RESULTS

#### J.1.1    RL ALGORITHMS

We conducted ablation experiments to assess the robustness of the SCORE mechanism under different RL algorithms, replacing the third-stage optimization algorithm while keeping earlier training stages consistent. As shown in Table 24, SCORE remained stable across a variety of methods, confirming its reliability as a reward mechanism. Executability-oriented metrics revealed that GRPO offered the best overall balance, benefiting from group-wise normalization that stabilized training and strengthened action fidelity. This supports our choice of GRPO as the primary optimization method in Thoth.

#### J.1.2    REWARD COMPUTATIONS

We further examined the impact of different reward computation strategies within the SCORE mechanism, considering the choice of order consistency logic (**L**CS or **S**trict), the method of combining order and semantic rewards (**P**roduct or **S**um), and the integration of step scale with step semantics (**P**roduct or **S**um). As summarized in Table 25, strict order scoring consistently outperformed LCS, product-based integration proved more effective than summation, and multiplicative incorporation of step scale with semantics yielded stronger executability. These trends collectively support the adoption of the S-S-P configuration in Thoth, which achieved the best balance between surface similarity and procedural fidelity.

Table 24: Further analysis results on RL algorithms.

(a)

| Algorithm | BERTScore-F1 | BERTScore-P | BERTScore-R | BLEU-1 | BLEU-2 |
|---|---|---|---|---|---|
| GPG | 97.81 | 98.01 | 97.62 | 68.56 | 47.30 |
| GRPO_Dr | 97.81 | 97.99 | 97.63 | 68.10 | 47.31 |
| GSPO | 97.81 | 97.98 | 97.65 | 68.65 | 47.36 |
| REINFORCE ++-baseline | 97.82 | 98.08 | 97.56 | 68.83 | 48.18 |
| ReMax | 97.81 | 98.05 | 97.57 | 67.93 | 47.87 |
| RLOO | 97.81 | 98.04 | 97.59 | 67.35 | 47.64 |
| Thoth (GRPO) | 97.82 | 97.99 | 97.65 | 68.56 | 48.28 |

(b)

| Algorithm | BLEU-3 | BLEU-4 | BLEU-AVG | ROUGE-1 | ROUGE-2 |
|---|---|---|---|---|---|
| GPG | 34.44 | 21.80 | 43.03 | 59.98 | 36.58 |
| GRPO_Dr | 34.44 | 21.85 | 42.93 | 60.31 | 36.89 |
| GSPO | 33.50 | 22.08 | 42.90 | 60.40 | 37.11 |
| REINFORCE ++-baseline | 34.71 | 22.69 | 43.60 | 59.92 | 37.08 |
| ReMax | 33.59 | 22.79 | 43.05 | 59.96 | 36.59 |
| RLOO | 33.60 | 22.61 | 42.80 | 60.41 | 37.15 |
| Thoth (GRPO) | 34.58 | 23.09 | 43.62 | 60.30 | 37.14 |

(c)

| Algorithm | ROUGE-L | Meteor | KW-F1 | KW-P | KW-R |
|---|---|---|---|---|---|
| GPG | 49.01 | 45.81 | 53.47 | 58.52 | 50.84 |
| GRPO_Dr | 49.12 | 47.14 | 53.98 | 59.29 | 51.06 |
| GSPO | 49.32 | 46.91 | 54.20 | 58.59 | 51.98 |
| REINFORCE ++-baseline | 49.19 | 46.19 | 53.59 | 60.40 | 49.71 |
| ReMax | 49.52 | 45.95 | 53.47 | 60.27 | 49.56 |
| RLOO | 50.00 | 46.66 | 53.65 | 60.29 | 49.94 |
| Thoth (GRPO) | 50.02 | 47.39 | 54.13 | 58.82 | 51.70 |

(d)

| Algorithm | Order-LCS | Order-S | Order-Tau | Semantic-A | Step-M |
|---|---|---|---|---|---|
| GPG | 74.02 | 22.50 | 70.33 | 45.07 | 46.67 |
| GRPO_Dr | 75.34 | 26.67 | 71.50 | 44.11 | 50.50 |
| GSPO | 74.93 | 23.67 | 68.67 | 43.93 | 51.17 |
| REINFORCE ++-baseline | 75.69 | 25.50 | 70.67 | 44.14 | 50.67 |
| ReMax | 73.58 | 23.00 | 69.33 | 44.89 | 51.11 |
| RLOO | 74.42 | 25.00 | 70.17 | 43.87 | 51.50 |
| Thoth (GRPO) | 75.34 | 25.50 | 73.33 | 46.60 | 53.00 |

### J.1.3 REWARD MAGNITUDES

To better understand the sensitivity of SCORE to the absolute scale of feedback, we examined different strategies for adjusting reward magnitude. Four configurations were compared: constant scaling, scaling up, symmetric shifting with equal positive and negative ranges, and the default Thoth setting. As shown in Table 26, model performance remained highly stable across all strategies, with only minimal variations observed on both text-level and executability-oriented metrics. This confirms that the effectiveness of SCORE arises mainly from its structured reward design rather

Table 25: Further analysis results on Reward Computations.

(a)

| Combine | BERTScore-F1 | BERTScore-P | BERTScore-R | BLEU-1 | BLEU-2 |
|---|---|---|---|---|---|
| L-P-P | 97.76 | 97.85 | 97.67 | 63.17 | 44.25 |
| L-P-S | 97.73 | 97.80 | 97.66 | 62.56 | 43.74 |
| L-S-P | 97.78 | 97.87 | 97.69 | 63.23 | 44.37 |
| L-S-S | 97.59 | 97.66 | 97.51 | 61.91 | 43.03 |
| S-P-P | 97.80 | 98.00 | 97.61 | 70.04 | 49.18 |
| S-P-S | 97.80 | 97.91 | 97.70 | 64.18 | 45.14 |
| S-S-S | 97.80 | 97.92 | 97.68 | 67.28 | 47.15 |
| Thoth (S-S-P) | 97.82 | 97.99 | 97.65 | 68.56 | 48.28 |

(b)

| Combine | BLEU-3 | BLEU-4 | BLEU-AVG | ROUGE-1 | ROUGE-2 |
|---|---|---|---|---|---|
| L-P-P | 31.31 | 20.74 | 39.87 | 58.81 | 35.40 |
| L-P-S | 30.95 | 20.30 | 39.39 | 58.86 | 35.31 |
| L-S-P | 31.32 | 20.54 | 39.86 | 59.24 | 35.82 |
| L-S-S | 30.36 | 19.79 | 38.77 | 58.49 | 34.62 |
| S-P-P | 35.18 | 23.44 | 44.46 | 59.52 | 36.48 |
| S-P-S | 32.20 | 21.34 | 40.71 | 59.33 | 36.05 |
| S-S-S | 33.73 | 22.49 | 42.66 | 60.08 | 36.56 |
| Thoth (S-S-P) | 34.58 | 23.09 | 43.62 | 60.30 | 37.14 |

(c)

| Combine | ROUGE-L | Meteor | KW-F1 | KW-P | KW-R |
|---|---|---|---|---|---|
| L-P-P | 48.17 | 45.93 | 53.21 | 55.78 | 52.72 |
| L-P-S | 47.23 | 46.11 | 52.67 | 54.75 | 52.47 |
| L-S-P | 48.09 | 46.08 | 53.50 | 56.30 | 52.78 |
| L-S-S | 46.85 | 45.63 | 52.83 | 54.64 | 53.05 |
| S-P-P | 49.40 | 45.57 | 53.35 | 59.23 | 50.02 |
| S-P-S | 48.47 | 46.37 | 53.62 | 56.23 | 53.10 |
| S-S-S | 48.47 | 46.54 | 53.56 | 57.36 | 51.65 |
| Thoth (S-S-P) | 50.02 | 47.39 | 54.13 | 58.82 | 51.70 |

(d)

| Combine | Order-LCS | Order-S | Order-Tau | Semantic-A | Step-M |
|---|---|---|---|---|---|
| L-P-P | 71.30 | 19.50 | 65.50 | 45.02 | 32.33 |
| L-P-S | 69.70 | 16.17 | 64.00 | 45.65 | 34.50 |
| L-S-P | 71.82 | 20.83 | 66.67 | 45.18 | 35.33 |
| L-S-S | 71.52 | 18.00 | 65.00 | 45.47 | 33.83 |
| S-P-P | 75.21 | 25.17 | 71.50 | 44.78 | 50.00 |
| S-P-S | 72.57 | 21.00 | 66.50 | 45.23 | 34.50 |
| S-S-S | 74.61 | 23.67 | 69.17 | 45.41 | 50.00 |
| Thoth (S-S-P) | 75.34 | 25.50 | 73.33 | 46.60 | 53.00 |

than the precise numerical range, and also suggests that it can be applied robustly to more complex open-ended tasks where reward scales may differ.

Table 26: Further analysis results on Reward Magnitudes.

(a)

| Reward_Scale | BERTScore-F1 | BERTScore-P | BERTScore-R | BLEU-1 | BLEU-2 |
|---|---|---|---|---|---|
| **Constant [0, 2.5]** | 97.80 | 97.98 | 97.62 | 67.79 | 47.50 |
| **Scaling_up [0, 5]** | 97.78 | 97.92 | 97.64 | 67.74 | 47.27 |
| **Shift [-1.25, 1.25]** | 97.79 | 97.98 | 97.60 | 67.76 | 47.45 |
| **Thoth [0, 1]** | 97.82 | 97.99 | 97.65 | 68.56 | 48.28 |

(b)

| Reward_Scale | BLEU-3 | BLEU-4 | BLEU-AVG | ROUGE-1 | ROUGE-2 |
|---|---|---|---|---|---|
| **Constant [0, 2.5]** | 33.60 | 22.00 | 42.72 | 60.06 | 36.38 |
| **Scaling_up [0, 5]** | 33.63 | 22.31 | 42.74 | 59.97 | 36.24 |
| **Shift [-1.25, 1.25]** | 33.57 | 22.82 | 42.90 | 60.11 | 36.69 |
| **Thoth [0, 1]** | 34.58 | 23.09 | 43.62 | 60.30 | 37.14 |

(c)

| Reward_Scale | ROUGE-L | Meteor | KW-F1 | KW-P | KW-R |
|---|---|---|---|---|---|
| **Constant [0, 2.5]** | 49.06 | 46.12 | 53.92 | 58.80 | 51.30 |
| **Scaling_up [0, 5]** | 49.07 | 46.01 | 53.02 | 56.72 | 51.28 |
| **Shift [-1.25, 1.25]** | 49.15 | 46.75 | 53.70 | 58.67 | 51.11 |
| **Thoth [0, 1]** | 50.02 | 47.39 | 54.13 | 58.82 | 51.70 |

(d)

| Reward_Scale | Order-LCS | Order-S | Order-Tau | Semantic-A | Step-M |
|---|---|---|---|---|---|
| **Constant [0, 2.5]** | 74.71 | 25.17 | 71.50 | 45.22 | 51.50 |
| **Scaling_up [0, 5]** | 73.92 | 26.17 | 72.33 | 45.14 | 51.67 |
| **Shift [-1.25, 1.25]** | 73.70 | 26.50 | 72.49 | 45.36 | 52.12 |
| **Thoth [0, 1]** | 75.34 | 25.50 | 73.33 | 46.60 | 53.00 |

## J.2 PRE-TRAINING ANALYSIS

Table 27: Performance differences on the MMLU dataset (Hendrycks et al., 2021) before and after pretraining. To examine potential catastrophic forgetting of general scientific knowledge, we evaluate MMLU on the OpenCompass platform (Contributors, 2023). "_pt" denotes the Thoth model after the pretraining stage only, and $\Delta$ indicates the performance gap between pretrained models and their base counterparts. The results clearly show that pretraining on a large collection of high-quality protocols not only preserves general capability but even yields improvements in certain disciplines.

| Dataset | Version | Qwen3-4B | Thoth-mini_pt | $\Delta$ | Qwen3-8B | Thoth_pt | $\Delta$ |
|---|---|---|---|---|---|---|---|
| lukaemon_mmlu_abstract_algebra | 2db373 | 91.00 | 87.00 | -4.00 | 88.00 | 91.00 | 3.00 |
| lukaemon_mmlu_anatomy | 72183b | 70.37 | 71.85 | 1.48 | 75.56 | 75.56 | 0.00 |
| lukaemon_mmlu_astronomy | d3ee01 | 84.87 | 89.47 | 4.60 | 90.79 | 88.82 | -1.97 |
| lukaemon_mmlu_business_ethics | 1dec08 | 77.00 | 72.00 | -5.00 | 81.00 | 77.00 | -4.00 |
| lukaemon_mmlu_clinical_knowledge | cb3218 | 80.38 | 81.89 | 1.51 | 83.77 | 86.04 | 2.27 |
| lukaemon_mmlu_college_biology | caec7d | 90.28 | 88.89 | -1.39 | 95.14 | 94.44 | -0.70 |
| lukaemon_mmlu_college_chemistry | 520aa6 | 66.00 | 63.00 | -3.00 | 71.00 | 71.00 | 0.00 |
| lukaemon_mmlu_college_computer_science | 99c216 | 80.00 | 83.00 | 3.00 | 87.00 | 83.00 | -4.00 |
| lukaemon_mmlu_college_mathematics | 678751 | 90.00 | 86.00 | -4.00 | 77.00 | 79.00 | 2.00 |
| lukaemon_mmlu_college_medicine | 38709e | 83.24 | 84.97 | 1.73 | 84.97 | 84.97 | 0.00 |
| lukaemon_mmlu_college_physics | 4f382c | 94.12 | 94.12 | 0.00 | 95.10 | 95.10 | 0.00 |
| lukaemon_mmlu_computer_security | ce7550 | 85.00 | 82.00 | -3.00 | 84.00 | 86.00 | 2.00 |
| lukaemon_mmlu_conceptual_physics | 63588e | 88.94 | 88.09 | -0.85 | 94.04 | 93.62 | -0.42 |
| lukaemon_mmlu_econometrics | d1134d | 71.93 | 71.93 | 0.00 | 80.70 | 84.21 | 3.51 |
| lukaemon_mmlu_electrical_engineering | 770ce3 | 83.45 | 80.00 | -3.45 | 80.69 | 84.14 | 3.45 |
| lukaemon_mmlu_elementary_mathematics | 269926 | 97.35 | 97.35 | 0.00 | 97.62 | 97.88 | 0.26 |
| lukaemon_mmlu_formal_logic | cfcb0c | 92.86 | 88.89 | -3.97 | 89.68 | 90.48 | 0.80 |
| lukaemon_mmlu_global_facts | ab07b6 | 51.00 | 52.00 | 1.00 | 44.00 | 57.00 | 13.00 |
| lukaemon_mmlu_high_school_biology | 37b125 | 85.81 | 87.74 | 1.93 | 90.00 | 93.87 | 3.87 |
| lukaemon_mmlu_high_school_chemistry | ae8820 | 89.16 | 89.66 | 0.50 | 91.13 | 89.16 | -1.97 |
| lukaemon_mmlu_high_school_computer_science | 9965a5 | 91.00 | 92.00 | 1.00 | 98.00 | 96.00 | -2.00 |
| lukaemon_mmlu_high_school_european_history | eefc90 | 82.42 | 81.82 | -0.60 | 87.27 | 83.03 | -4.24 |
| lukaemon_mmlu_high_school_geography | 0780e6 | 86.87 | 91.92 | 5.05 | 93.94 | 91.92 | -2.02 |
| lukaemon_mmlu_high_school_government_and_politics | 3c52f9 | 91.19 | 90.67 | -0.52 | 95.34 | 95.34 | 0.00 |
| lukaemon_mmlu_high_school_macroeconomics | a01685 | 87.95 | 86.92 | -1.03 | 92.82 | 91.54 | -1.28 |
| lukaemon_mmlu_high_school_mathematics | ed4dc0 | 95.19 | 95.19 | 0.00 | 67.41 | 64.44 | -2.97 |
| lukaemon_mmlu_high_school_microeconomics | 04d21a | 93.70 | 93.70 | 0.00 | 96.22 | 95.80 | -0.42 |
| lukaemon_mmlu_high_school_physics | 93278f | 86.75 | 86.75 | 0.00 | 90.07 | 86.75 | -3.32 |
| lukaemon_mmlu_high_school_psychology | 7db114 | 91.93 | 92.84 | 0.91 | 95.05 | 94.31 | -0.74 |
| lukaemon_mmlu_high_school_statistics | 8f3f3a | 89.81 | 89.35 | -0.46 | 91.67 | 90.74 | -0.93 |
| lukaemon_mmlu_high_school_us_history | 8932df | 85.78 | 87.75 | 1.97 | 90.20 | 88.24 | -1.96 |
| lukaemon_mmlu_high_school_world_history | 048e7e | 85.65 | 82.70 | -2.95 | 88.61 | 89.03 | 0.42 |
| lukaemon_mmlu_human_aging | 82a410 | 73.99 | 71.75 | -2.24 | 76.68 | 76.68 | 0.00 |
| lukaemon_mmlu_human_sexuality | 42407c | 83.97 | 81.68 | -2.29 | 89.31 | 85.50 | -3.81 |
| lukaemon_mmlu_international_law | cf3179 | 76.03 | 78.51 | 2.48 | 77.69 | 80.99 | 3.30 |
| lukaemon_mmlu_jurisprudence | 001f24 | 75.00 | 79.63 | 4.63 | 87.96 | 80.56 | -7.40 |
| lukaemon_mmlu_logical_fallacies | 9cebb0 | 85.28 | 84.66 | -0.62 | 86.50 | 85.89 | -0.61 |
| lukaemon_mmlu_machine_learning | 0283bb | 76.79 | 76.79 | 0.00 | 78.57 | 82.14 | 3.57 |
| lukaemon_mmlu_management | 80876d | 84.47 | 85.44 | 0.97 | 88.35 | 86.41 | -1.94 |
| lukaemon_mmlu_marketing | 7394e3 | 91.88 | 91.45 | -0.43 | 94.87 | 91.45 | -3.42 |
| lukaemon_mmlu_medical_genetics | 881ef5 | 88.00 | 88.00 | 0.00 | 91.00 | 89.00 | -2.00 |
| lukaemon_mmlu_miscellaneous | 935647 | 88.12 | 87.48 | -0.64 | 91.70 | 91.44 | -0.26 |
| lukaemon_mmlu_moral_disputes | a2173e | 74.28 | 72.83 | -1.45 | 72.25 | 73.99 | 1.74 |
| lukaemon_mmlu_moral_scenarios | f6dbe2 | 62.46 | 60.67 | -1.79 | 60.22 | 62.46 | 2.24 |
| lukaemon_mmlu_nutrition | 4543bd | 76.14 | 76.80 | 0.66 | 85.29 | 84.31 | -0.98 |
| lukaemon_mmlu_philosophy | 08042b | 72.03 | 68.81 | -3.22 | 76.53 | 79.10 | 2.57 |
| lukaemon_mmlu_prehistory | bbb197 | 81.48 | 80.25 | -1.23 | 85.80 | 84.88 | -0.92 |
| lukaemon_mmlu_professional_accounting | 444b7f | 73.40 | 73.40 | 0.00 | 81.21 | 80.85 | -0.36 |
| lukaemon_mmlu_professional_law | 5f7e6c | 49.80 | 50.07 | 0.27 | 55.61 | 56.39 | 0.78 |
| lukaemon_mmlu_professional_medicine | 857144 | 82.72 | 84.19 | 1.47 | 89.34 | 87.50 | -1.84 |
| lukaemon_mmlu_professional_psychology | 221a16 | 77.61 | 75.65 | -1.96 | 79.90 | 79.41 | -0.49 |
| lukaemon_mmlu_public_relations | e7d39b | 68.18 | 70.00 | 1.82 | 73.64 | 76.36 | 2.72 |
| lukaemon_mmlu_security_studies | 9b1743 | 71.84 | 75.51 | 3.67 | 76.33 | 77.96 | 1.63 |
| lukaemon_mmlu_sociology | c266a2 | 80.10 | 80.60 | 0.50 | 87.06 | 83.08 | -3.98 |
| lukaemon_mmlu_us_foreign_policy | 528cfe | 86.00 | 86.00 | 0.00 | 87.00 | 85.00 | -2.00 |
| lukaemon_mmlu_virology | 12e270 | 55.42 | 51.81 | -3.61 | 54.82 | 57.23 | 2.41 |
| lukaemon_mmlu_world_religions | 232c09 | 84.21 | 85.38 | 1.17 | 85.96 | 88.30 | 2.34 |
| AVG | | 81.41 | 81.21 | -0.20 | 83.88 | 83.79 | -0.09 |

# K CASE STUDIES

---

**Retrieval (Level2)**

**Query:**

During the nuclei washing step of the snPATHO-seq workflow, a researcher accidentally used wash solution 1 for the final resuspension instead of wash solution 2 before filtering through the 40 μm strainer. To correct this mistake and ensure the nuclei are properly prepared for counting, what sequence of corrective actions should be taken to replace wash solution 1 with wash solution 2 prior to proceeding?

**Action Library:**

"centrifuge", "titer", "remove", "pour", "resuspend", "filter", "dry".

---

**Model Output:**

<think>
The problem here is that the final resuspension was done with wash solution 1 instead of wash solution 2. Wash solution 2 is designed to remove residual contaminants and provide the correct buffer composition for downstream processing. Using wash solution 1 could compromise nuclei integrity or interfere with downstream assays. The logical correction is to immediately replace the incorrect solution with wash solution 2. Since the nuclei were just resuspended, the best approach is to centrifuge the sample to pellet the nuclei, remove the incorrect wash solution 1, and resuspend in the correct wash solution 2. After resuspension, filtering through a 40 μm strainer is necessary to ensure a single-cell suspension suitable for counting. This plan assumes that wash solution 2 is available and that the centrifuge and filter are functioning properly. The sequence is centrifuge → remove incorrect solution → resuspend in correct solution → filter, which directly addresses the mistake and restores the workflow.
</think>

<key>
Step 1: {"action": "centrifuge", "objects": ["nuclei suspension"], "parameters": ["500xg", "5 min", "4 °c"]}
Step 2: {"action": "remove", "objects": ["supernatant"], "parameters": ["carefully"]}
Step 3: {"action": "resuspend", "objects": ["nuclei pellet"], "parameters": ["1 ml", "wash solution 2"]}
Step 4: {"action": "filter", "objects": ["nuclei suspension"], "parameters": ["40 μm strainer"]}
</key>

<orc>
Step 1: Centrifuge the nuclei suspension at 500xg for 5 min at 4 °C.
Step 2: Remove the supernatant carefully.
Step 3: Resuspend the nuclei pellet in 1 ml wash solution 2.
Step 4: Filter the nuclei suspension through a 40 μm strainer.
</orc>

<note>
Handle all wash solutions on ice to maintain nuclei integrity. Centrifugation at 4 °C prevents enzymatic activity that could damage nuclei. When filtering, avoid applying excessive pressure to prevent rupturing nuclei. Dispose of waste solutions according to biosafety protocols for handling biological samples.
</note>

Figure 11: Case result of Thoth on the Retrieval task.

**Planning (Level2)**

**Query:**

How should I prepare a freshly cleaved mica surface and functionalize it with APS working solution in order to stably immobilize DNA for AFM liquid imaging of Twinkle helicase dynamics?

**Action Library:**

"dry", "recover", "deposit", "incubate", "wash", "peel".

---

**Model Output:**

<think>
The goal here is to generate a stable DNA-immobilized surface for AFM liquid imaging of Twinkle helicase. Mica is used as a support substrate because of its atomically flat surface, but it must be freshly cleaved to expose a clean lattice suitable for binding. The protocol specifies that mica should be peeled with Scotch tape to obtain a fresh, uniform surface. Next, the functionalization step is required—APS (amino-polyethylene glycol silane) is used to introduce amine groups, allowing subsequent DNA immobilization via covalent attachment. To ensure effective functionalization, the APS solution must be deposited evenly on the mica, incubated for a defined time (5 min), and then washed with Milli-Q water to remove unbound APS. Finally, the mica is dried briefly to prepare a stable substrate. This sequence ensures that the mica has the correct surface chemistry, the DNA will bind stably, and the substrate will be suitable for AFM imaging in liquid.
</think>

<key>
Step 1: {"action": "peel", "objects": ["mica sheet"], "parameters": ["with scotch tape", "to expose fresh layer"]}
Step 2: {"action": "deposit", "objects": ["aps working solution"], "parameters": ["200 µl", "on mica"]}
Step 3: {"action": "incubate", "objects": ["mica with aps"], "parameters": ["5 min"]}
Step 4: {"action": "wash", "objects": ["mica"], "parameters": ["milli-q water", "twice"]}
Step 5: {"action": "dry", "objects": ["mica sheet"], "parameters": ["2 min"]}
</key>

<orc>
Step 1: Peel the mica sheet with Scotch tape to expose a fresh layer.
Step 2: Deposit 200 µl of APS working solution on the mica.
Step 3: Incubate the mica with APS for 5 min.
Step 4: Wash the mica with Milli-Q water twice.
Step 5: Dry the mica sheet for 2 min.
</orc>

<note>
APS is a silane-based chemical and should be handled in a fume hood with gloves and eye protection to avoid inhalation or skin contact. Ensure that mica is freshly cleaved before use to prevent uneven surface binding. Dispose of APS-containing waste according to institutional hazardous chemical guidelines.
</note>

Figure 12: Case result of Thoth on the Planning task.

**Troubleshooting (Level1)**

**Query:**

During fecal microbiota transplantation, I observed that the supernatant collected after homogenization and centrifugation at 805 x g for 10 min was very cloudy, suggesting incomplete removal of solid material. This made oral gavage difficult because the gavage needle clogged repeatedly. How can I troubleshoot this issue to obtain a clearer bacterial suspension without losing viable microbes?

**Action Library:**

"filter", "collect", "centrifuge".

**Model Output:**

<think>
The problem is that the supernatant remains cloudy after the initial centrifugation, indicating that fine particulates are not being removed efficiently. This suggests either insufficient centrifugation force or inadequate filtering. The goal is to obtain a clear suspension suitable for oral gavage without damaging the microbes.

First, the most logical step is to repeat centrifugation at a higher speed to pellet finer solids. If the pellet is still unclear, the next best option is filtration through a sterile filter. The protocol mentions filtering with 0.45 μm and 0.2 μm filters, so using a 0.2 μm filter would ensure removal of remaining bacteria and debris while preserving viability.

This plan assumes that the centrifuge can be safely operated at higher speeds (e.g., 1600 x g) without damaging the samples and that sterile filters are available. By first clarifying the suspension with an additional centrifugation and then filtering, we can maximize clarity and ensure the gavage needle does not clog.
</think>

<key>
Step 1: {"action": "centrifuge", "objects": ["supernatant"], "parameters": ["1600xg", "10 min"]}
Step 2: {"action": "collect", "objects": ["supernatant"], "parameters": ["after centrifugation"]}
Step 3: {"action": "filter", "objects": ["supernatant"], "parameters": ["0.2 μm filter"]}
</key>

<orc>
Step 1: Centrifuge the supernatant at 1600xg for 10 min.
Step 2: Collect the supernatant after centrifugation.
Step 3: Filter the supernatant through a 0.2 μm filter.
</orc>

<note>
Handle centrifuged supernatant carefully to avoid disturbing the pellet. Filtration under sterile conditions is essential to prevent contamination. Use filters rated for microbial removal, and avoid applying excessive pressure during filtration to prevent filter rupture or aerosol formation.
</note>

Figure 13: Case result of Thoth on the Troubleshooting task.

**Constraint (Level1)**

**Query:**

During the generation of spheroids, we usually add 10 ml PBS to the bottom of the 10 cm dish to maintain humidity. However, PBS stocks are unavailable. How can we modify the hanging drop setup to maintain spheroid viability without PBS while still preventing evaporation and contamination?

**Action Library:**

"add", "sterilize", "place".

---

**Model Output:**

<think>
The key challenge here is that PBS is essential in the hanging drop setup to prevent evaporation and maintain spheroid viability. Without PBS, spheroids are at risk of drying out, leading to experimental failure. Since PBS is unavailable, we must find an alternative. A reasonable substitution is sterile distilled water, which can provide hydration and prevent excessive evaporation. To compensate for the lack of buffering capacity, we can sterilize the water and keep it at room temperature. Placing the dishes in a humid chamber will also help minimize evaporation. This plan assumes that sterile distilled water is available and that the spheroids are not highly sensitive to changes in buffer composition. The trade-off is that distilled water lacks ionic balance, which may slightly affect spheroid growth, but it is acceptable for short-term maintenance.
</think>

<key>
Step 1: {"action": "sterilize", "objects": ["distilled water"], "parameters": ["0.45 µm filter"]}
Step 2: {"action": "add", "objects": ["sterilized distilled water"], "parameters": ["10 ml", "bottom of 10 cm dish"]}
Step 3: {"action": "place", "objects": ["dish"], "parameters": ["humid chamber"]}
</key>

<orc>
Step 1: Sterilize distilled water through a 0.45 µm filter.
Step 2: Add 10 ml of sterilized distilled water to the bottom of the 10 cm dish.
Step 3: Place the dish in a humid chamber.
</orc>

<note>
When substituting PBS with sterilized distilled water, ensure the water is sterilized to prevent contamination. Avoid handling spheroids directly to prevent damage. Keep the setup in a humid chamber to minimize evaporation, but monitor humidity levels carefully to prevent excess moisture that could compromise spheroid viability.
</note>

Figure 14: Case result of Thoth on the Constraint task.

## Scaling (Level2)

**Query:**

You need to prepare only 100 ml of 4% formalin solution for perfusion instead of the standard 500 ml. The original recipe requires 300 ml isotonic saline and 200 ml 1:10 formalin solution to make 500 ml total. How would you scale the recipe to prepare exactly 100 ml while maintaining the correct proportions and final concentration?

**Action Library:**

"dilute", "measure", "setup", "mix", "blot", "combine", "chill".

---

**Model Output:**

<think>
The goal is to scale down the formalin solution recipe from 500 ml to 100 ml while maintaining the same ratio of isotonic saline to formalin solution. The original recipe has 300 ml saline and 200 ml formalin solution, totaling 500 ml. The ratio is therefore 300:200, or 3:2. To scale this down to 100 ml, we need to divide both components by 5. This results in 60 ml saline and 40 ml formalin solution. The formalin solution must then be diluted with saline to reach 100 ml total. This plan assumes that the 1:10 formalin solution is already prepared as per the recipe and that isotonic saline is available. By maintaining the ratio, the final concentration will remain 4% formalin.
</think>

<key>
Step 1: {"action": "measure", "objects": ["isotonic saline"], "parameters": ["60 ml"]}
Step 2: {"action": "measure", "objects": ["1:10 formalin solution"], "parameters": ["40 ml"]}
Step 3: {"action": "combine", "objects": ["isotonic saline", "1:10 formalin solution"], "parameters": ["final volume 100 ml"]}
Step 4: {"action": "mix", "objects": ["solution"], "parameters": ["thoroughly"]}
Step 5: {"action": "chill", "objects": ["4% formalin solution"], "parameters": ["4 °c"]}
</key>

<orc>
Step 1: Measure 60 ml isotonic saline.
Step 2: Measure 40 ml 1:10 formalin solution.
Step 3: Combine the isotonic saline and 1:10 formalin solution to a final volume of 100 ml.
Step 4: Mix the solution thoroughly.
Step 5: Chill the 4% formalin solution to 4 °c.
</orc>

<note>
Formalin is toxic and a strong irritant; handle it in a fume hood and wear gloves, lab coat, and eye protection. Isotonic saline is safe, but care should be taken to avoid spills. Ensure the solution is chilled properly to prevent degradation and maintain concentration accuracy.
</note>

Figure 15: Case result of Thoth on the Scaling task.

---

**Safety (Level2)**

**Query:**

During the SeqFISH tissue fixation step, how should the lab safely handle and dispose of 4% paraformaldehyde (PFA) solution when preparing and washing tissues, given that PFA is highly toxic and generates harmful fumes?

**Action Library:**

"load", "collect", "wash", "decontaminate", "prepare", "wear".

---

**Model Output:**

<think>
The goal here is to ensure safe handling of PFA, a toxic fixative that releases formaldehyde vapors. The major risks are inhalation of fumes and accidental skin or eye contact. To mitigate these, scientists must work in a certified chemical fume hood, wear appropriate PPE, and collect all liquid waste in designated formaldehyde-compatible containers for hazardous waste disposal. The plan should include preparing the fixative under a hood, washing tissues in fresh buffer after fixation, and decontaminating the workspace to prevent residual exposure. By structuring these steps, we ensure both operator safety and compliance with institutional chemical safety regulations. This plan assumes that a fume hood, PPE, and proper waste containers are available.
</think>

<key>
Step 1: {"action": "wear", "objects": ["lab coat", "nitrile gloves", "safety goggles"], "parameters": ["before handling pfa"]}
Step 2: {"action": "prepare", "objects": ["4% pfa solution"], "parameters": ["in fume hood"]}
Step 3: {"action": "wash", "objects": ["tissues"], "parameters": ["1x pbs", "in fume hood"]}
Step 4: {"action": "collect", "objects": ["pfa waste"], "parameters": ["in labeled hazardous waste container"]}
Step 5: {"action": "decontaminate", "objects": ["work area"], "parameters": ["with appropriate cleaning solution"]}
</key>

<orc>
Step 1: Wear a lab coat, nitrile gloves, and safety goggles before handling PFA.
Step 2: Prepare 4% PFA solution in a fume hood.
Step 3: Wash the tissues in 1x PBS in the fume hood.
Step 4: Collect PFA waste in a labeled hazardous waste container.
Step 5: Decontaminate the work area with an appropriate cleaning solution.
</orc>

<note>
PFA is toxic and releases formaldehyde vapors that are harmful when inhaled. Always handle it in a certified chemical fume hood while wearing gloves, goggles, and a lab coat. Collect all PFA waste in properly labeled hazardous waste containers for institutional disposal. Decontaminate surfaces immediately after use to prevent residual exposure.
</note>

Figure 16: Case result of Thoth on the Safety task.

## L   PROMPTS

You are a highly precise scientific information extraction assistant. Your sole purpose is to parse unstructured laboratory protocols from the provided text and reformat them into a structured JSON array.

Core Directives:
1. Extraction, Not Creation: You must ONLY extract information explicitly present in the provided text. Do NOT infer, guess, add external knowledge, or hallucinate any details, with the limited exception of the category field as defined in its schema. Your primary role is to organize the given information.
2. Comprehensive Extraction: For every field, you must extract all relevant information comprehensively. Do not shorten, paraphrase, or simplify the original text. If details such as concentrations, quantities, catalog numbers, or manufacturer names are present, they MUST be included.
3. JSON Sanitization and Escaping (Crucial for Parsing): The source text may contain formulas and non-printable control characters. Before outputting, you MUST perform two sanitization steps:
   - Remove Invalid Characters: Remove all non-printable ASCII control characters, especially the NULL character (\u0000), from the extracted text.
   - Escape Backslashes: Ensure all backslashes (\) within the JSON strings are properly escaped as double backslashes (\\). This is critical for file paths and LaTeX-style formulas (e.g., a literal \alpha must become \\alpha in the final JSON string).
4. Handle Missing Information: If a specific category of information (e.g., "notes" or "equipments") is not mentioned in a protocol, you MUST return "None".
5. Strict JSON Output: Your final output MUST be a valid JSON array [ { ... }, { ... } ]. Do not include any explanatory text, markdown formatting, or code fences like ```json and ``` in your response. The output must start with [ and end with ].

JSON Schema for each protocol object:
Each object in the array must contain the following six keys:
- "exp_name": A concise name for the experiment, extracted directly from headings or titles.
- "abstract": A paragraph or section describing the background, purpose, or application scenario of the experiment.
- "materials": A list or text block of all reagents, chemicals, solutions, and consumables mentioned.
- "equipments": A list or text block of all laboratory instruments, tools, and hardware required.
- "procedures": The detailed, step-by-step experimental instructions. Preserve the order and as much detail as possible from the original text.
- "notes": Any safety precautions, warnings, troubleshooting tips, or important side-notes mentioned in the protocol.
- "category": The scientific discipline of the experiment. To determine this, follow a strict two-step process:
   1. Step 1 (Direct Extraction): First, search the text for explicit keywords, subject headings, or category labels (e.g., "Field: Molecular Biology", "Keywords: Biochemistry"). If found, use them directly to determine the category.
   2. Step 2 (Constrained Classification): If and ONLY IF no explicit category is found in Step 1, infer the most appropriate category based on the exp_name and abstract. You MUST choose from the following list: Molecular Biology, Cell Biology, Biochemistry, Genetics, Immunology, Microbiology, Chemistry, Bioinformatics, Plant Science, General Laboratory Procedure. If the protocol is too generic or doesn't fit, default to General Laboratory Procedure.

Figure 17: System prompt 1 for structured integration with model_based.

You are a highly precise and intelligent scientific information extraction AI. Your purpose is to parse long, complex laboratory protocols containing multiple sub-protocols, tables, and image references, and consolidate them into a single, structured JSON object representing the main experimental workflow.

Core Directives:
1. Consolidation as the Primary Goal: Your main task is to synthesize the multiple related sub-protocols within the text into a single, cohesive JSON object. Identify the overarching experimental goal and use that as the basis for your summary. Your role is to intelligently aggregate, not simply concatenate, the information.
2. Extraction, Not Creation: While consolidating, you must still base your output ONLY on information explicitly present in the provided text. Do not add external knowledge or hallucinate details. The synthesis process should organize existing information, not create new information. The exception is the category field, which follows its own defined logic.
3. Comprehensive Aggregation: The final consolidated output must be comprehensive. When creating the single JSON object, ensure that all critical materials, equipment, and procedural steps from the various sub-protocols are included. Do not lose key experimental details during the consolidation process.
4. Handling Rich Content (Tables and Image Captions):
   - Tables: You must parse information from markdown tables. For example, a table listing reagents and their concentrations must be fully extracted and integrated into the materials field.
   - Image Captions: You cannot see images, but you MUST read and interpret the text in image captions (e.g., "Figure 1: Results of the kinase assay..."). Use this contextual information to enrich the abstract or procedures fields where it helps clarify the purpose or outcome of a step.
5. Fallback for Poor Consolidation: If the sub-protocols are too distinct or unrelated to be logically combined into a single coherent workflow (e.g., a protocol for DNA extraction followed by a completely separate protocol for protein crystallization), DO NOT force a summary. In this case, identify and extract ONLY the most central or the first comprehensive protocol from the document as a single JSON object.
6. Handle Missing Information: If a specific category of information (e.g., "notes" or "equipments") is not mentioned in a protocol, you MUST return "None".
7. Strict JSON Output: Your final output MUST be a valid JSON array [ { ... }, { ... } ]. Do not include any explanatory text, markdown formatting, or code fences like ```json and ``` in your response.
8. Strict Data Type: String ONLY: This is a critical rule. The value for every key in the output JSON object MUST be a single string. Even if the source content is a list of items (like materials or procedural steps), you must format it as a single, multi-line string. Do NOT use JSON arrays (e.g., ["item1", "item2"]) for any field's value.
9. JSON Sanitization and Escaping (Crucial for Parsing): The source text may contain formulas and non-printable control characters. Before outputting, you MUST perform two sanitization steps:
   - Remove Invalid Characters: Remove all non-printable ASCII control characters, especially the NULL character ($\backslash$u0000), from the extracted text.
   - Escape Backslashes: Ensure all backslashes ($\backslash$) within the JSON strings are properly escaped as double backslashes ($\backslash\backslash$). This is critical for file paths and LaTeX-style formulas (e.g., a literal $\backslash$alpha must become $\backslash\backslash$alpha in the final JSON string).

JSON Schema for the protocol object:
Each object in the array must contain the following seven keys:
- "exp_name": The name of the main, overarching experiment, synthesized from the titles of the main protocol and its sub-protocols.
- "abstract": A consolidated paragraph describing the overall background, purpose, and application of the entire experimental workflow.
- "materials": An aggregated and de-duplicated list of all reagents, chemicals, solutions, kits, and consumables mentioned across all sub-protocols and tables.
- "equipments": An aggregated and de-duplicated list of all laboratory instruments, tools, and hardware required for the entire workflow.
- "procedures": The synthesized, step-by-step experimental instructions representing the complete, logical workflow from start to finish. You should structure this clearly, for example, by using headings like "Part 1: Sample Preparation", "Part 2: Main Assay", etc., to indicate which part of the procedure comes from which sub-protocol.
- "notes": A consolidated list of all important safety precautions, warnings, and troubleshooting tips from across the entire document.
- "category": The scientific discipline of the overall experiment. To determine this, follow a strict two-step process:
  1. Step 1 (Direct Extraction): First, search the text for explicit keywords, subject headings, or category labels (e.g., "Field: Molecular Biology", "Keywords: Biochemistry"). If found, use them directly to determine the category.
  2. Step 2 (Constrained Classification): If and ONLY IF no explicit category is found in Step 1, infer the most appropriate category based on the exp_name and abstract. You MUST choose from the following list: Molecular Biology, Cell Biology, Biochemistry, Genetics, Immunology, Microbiology, Chemistry, Bioinformatics, Plant Science, General Laboratory Procedure. If the protocol is too generic or doesn't fit, default to General Laboratory Procedure.

Figure 18: System prompt 2 for structured integration with model_based.

You are a scientific protocol refinement assistant. Your sole purpose is to receive pre-extracted, but potentially messy and disordered, fields of a laboratory protocol and to clean, reorder, and format them into a clean, logical, and structured JSON object.

Core Directives:
1. Preserve Content Integrity: This is your most important rule. You MUST NOT add new scientific information, delete existing steps, or change critical details like chemical names, concentrations, quantities, or durations. Your role is to reformat and reorder, NOT to rewrite or summarize the core content.
2. Logical Reordering of Procedures: The procedures field may contain steps that are out of chronological order due to parsing errors. You must analyze the text of the steps to determine their logical sequence. Use explicit numbering (e.g., 1., 2., a., b.) or contextual clues (e.g., 'First...', 'Next...', 'After incubation...') to reconstruct the correct workflow. If the original numbering is correct but formatting is broken, fix the formatting.
3. Format Markdown and Artifacts:
  - Tables: If you encounter Markdown tables within a field, convert the information into a readable, clean text format (e.g., a bulleted list).
  - Image References: You may encounter Markdown image tags like . You MUST completely remove these tags. If a caption exists (e.g., "Figure 1: Western blot analysis."), you must retain this caption text and integrate it smoothly into the surrounding sentences to preserve the logical flow. The goal is to keep the descriptive information from the caption while discarding the non-textual image link.
  - Parsing Artifacts: Remove any artifacts from the PDF-to-MD conversion process, such as stray page numbers, repeated headers/footers, or awkward line breaks that interrupt sentences, while ensuring the scientific meaning is preserved.
4. Strict JSON Output: Your final output MUST be a single, valid JSON object. Do not wrap it in a JSON array [] or use markdown code fences like ```json. The output must start with { and end with }.
5. Strict Data Type: String ONLY: This is a critical rule. The value for every key in the output JSON object MUST be a single string. Even if the source content is a list of items (like materials or procedural steps), you must format it as a single, multi-line string. Do NOT use JSON arrays (e.g., ["item1", "item2"]) for any field's value.
6. JSON Sanitization and Escaping (Crucial for Parsing): The source text may contain formulas and non-printable control characters. Before outputting, you MUST perform two sanitization steps:
  - Remove Invalid Characters: Remove all non-printable ASCII control characters, especially the NULL character (\u0000), from the extracted text.
  - Escape Backslashes: Ensure all backslashes (\) within the JSON strings are properly escaped as double backslashes (\\). This is critical for file paths and LaTeX-style formulas (e.g., a literal \alpha must become \\alpha in the final JSON string).

JSON Schema for the output object:
The output JSON object must contain these exact seven keys, with their content cleaned and reordered according to the directives above:
- "exp_name"
- "abstract"
- "materials"
- "equipments"
- "procedures"
- "notes"

Figure 19: System prompt 3 for structured integration with model_based.

You are an expert scientific reasoner and problem-solver, training a world-class scientific research assistant AI. Your primary task is to devise and articulate logical solutions to experimental challenges. You will express these solutions in a high-quality, structured format based on the context of a provided scientific protocol.

You will be given a Scientific Experiment Protocol in a structured JSON format. Your generated response must strictly adhere to the following XML structure:

<question>
[A clear, relevant question you formulate that poses a scientific challenge or query.]
</question>
<think>
[This is the core of your response. Articulate the scientific strategy to solve the problem in the question. Start by analyzing the goal, then formulate a hypothesis or a clear plan of attack. Justify why the proposed sequence of actions is the most logical and effective approach. This must reflect the critical thinking of a scientist.]
</think>
<key>
[The structured, machine-readable plan derived from the reasoning in the <think> tag. Each step represents a single, atomic action.]
Step 1: {"action": "detach", "objects": ["bmdcs"], "parameters": ["1x pbs-5 mm edta", "10 min", "37 °c"]}
Step 2: {"action": "wash", "objects": ["cells"], "parameters": ["1x pbs", "twice"]}
Step 3: {"action": "centrifuge", "objects": ["cells"], "parameters": ["367xg", "10 min"]}
...
</key>
<orc>
[A concise, natural-language summary of the plan in <key>. Each step must be an imperative sentence.]
Step 1: Detach bmdcs with 1x pbs-5 mm edta for 10 min at 37 °c.
Step 2: Wash the cells with 1x pbs twice.
Step 3: Centrifuge the cells at 367xg for 10 min.
</orc>
<note>
[A concise paragraph summarizing the most critical safety information relevant to the steps in the plan.]
</note>

1. Role Descriptions
<think> (CRITICAL): The important part of your output. It is the explicit, step-by-step scientific reasoning that justifies the plan presented in <key>. It must bridge the gap between the problem in the <question> and the solution in the <key>.
<orc> (Natural Language Summary): This section translates each step from the <key> tag into a simple, human-readable instruction. Each step must be a concise, imperative (command) sentence that strictly reflects the action, objects, and parameters of the corresponding step in <key>, adding no new information. The number of steps must match <key> exactly.

2. <key> Formatting Rules (CRITICAL)
1) One Action Per Step: Each Step must correspond to a single, atomic action and contain exactly one JSON dictionary. If a logical operation requires multiple actions (e.g., 'wash and then centrifuge'), you must break it down into sequential, separate steps.
   - Source: "Stop the reaction by adding cold PBS and placing the tubes on ice."
   - Correct Breakdown:
      Step 1: {"action": "add", "objects": ["cold pbs"], "parameters": ["to stop reaction"]}
      Step 2: {"action": "place", "objects": ["tubes"], "parameters": ["on ice"]}
2) Lowercase Content: All string values within the JSON object (action, elements in objects, and elements in parameters) must be in lowercase. Numbers and standard units (e.g., °C, xg, mM) are exempt.
3) Structure of Each Action:
   - "action": A string containing a single, concise verb in lowercase (e.g., "add", "incubate", "centrifuge", "wash").
   - "objects": A list of strings. These are the direct objects of the action, in lowercase. Use specific, meaningful nouns (e.g., "bmdcs", "cell pellet", "1x pbs").
   - "parameters": A list of strings that modify the action, in lowercase. This includes quantities, durations, temperatures, speeds, and key conditions.
4) Parameter-Object Relationship: Parameters in the "parameters" list must apply jointly to the action performed on all objects in the "objects" list.
5) Conciseness and Semantic Distillation: Your primary goal is to distill the core scientific action, not to transcribe the original text verbatim. Simplify verbose language into concise keywords or short phrases.
   - Bad (Verbose): "parameters": ["in a water bath set at 37 °c"]
   - Good (Concise & Distilled): "parameters": ["37 °c", "in water bath"]

Figure 20: System prompt used for constructing SciRecipe.

**"Retrieval":** "Generate a data sample that demands precise extraction of factual, lab-ready parameters from authoritative sources, emphasizing exact values, units, tolerances, environmental conditions, and contextual qualifiers (e.g., temperature, pH, grade, catalog numbers) that can be directly transcribed into protocol fields; require cross-checking for consistency across related parameters (stock vs. working concentrations, volumes vs. instrument ranges), explicit unit normalization, correct significant figures, and clear source attribution or provenance notes; the scenario should penalize guesswork and reward verifiability, internal coherence, and readiness for immediate use without human reinterpretation."

**"Planning":** "Generate questions that demand the model translate a high-level research objective into a granular, step-by-step, executable experimental workflow. These prompts should challenge the model to logically sequence actions, identify necessary inputs and outputs for each stage, specify required equipment and reagents, and integrate quality control checkpoints. The aim is to assess the model's ability to understand dependencies, allocate resources, and construct a coherent experimental plan that is both feasible and reproducible. Focus on scenarios where multiple steps are interconnected, requiring a holistic understanding of the scientific process, from sample preparation to final analysis. The generated data should reflect the thought process of designing a robust, end-to-end experiment."

**"Troubleshooting":** "Your task is to create questions that simulate real-world experimental failures or unexpected outcomes, requiring the model to systematically diagnose problems and propose actionable, testable solutions. These prompts should present a specific symptom or deviation from expected results. The model should then identify potential causes, prioritize them, suggest specific, single-variable corrective actions, and outline how to verify the fix. The generated data should encourage iterative problem-solving, mimicking the 'observe-hypothesize-test-verify' cycle crucial for resilience and efficiency in research. Think about common issues in various lab techniques (e.g., PCR, Western blot, cell culture contamination) where a methodical approach to error resolution is paramount."

**"Constraint":** "Generate questions that require the model to adapt experimental plans under non-ideal, constrained conditions. These prompts should present a scenario where a critical resource (e.g., specific reagent, equipment, budget, time, or computational power) is unavailable or limited. The model needs to identify acceptable alternative methods or materials that can still achieve the core experimental objective, while clearly articulating the trade-offs involved (e.g., cost, efficiency, sensitivity, comparability of results). The goal is to train the model to be flexible and pragmatic, capable of navigating 'Constraint Satisfaction Problems' in a lab setting and preventing dead ends when ideal conditions are not met. Consider real-world limitations that often arise in resource-limited environments."

**"Scaling":** "Your task is to create questions that test the model's essential practical skills in unit conversion, stoichiometric calculations, and scaling experimental protocols. These prompts should involve adjusting reaction volumes, reagent concentrations, or sample sizes while maintaining critical ratios and ensuring technical feasibility (e.g., considering minimum pipetting volumes, solubility limits, or required excess for practical handling). The aim is to train the model to accurately translate theoretical recipes into precise, actionable laboratory instructions, preventing errors arising from incorrect calculations or impractical volumes. Focus on scenarios where a protocol needs to be scaled up or down, or where units need to be consistently converted for different components."

**"Safety":** "Generate questions that prompt the model to proactively identify and address potential hazards associated with specific chemicals, biological materials, or procedures. The model should automatically append necessary safety precautions, personal protective equipment (PPE) requirements, waste disposal guidelines, and compliance reminders, even if not explicitly asked. The goal is to train the model to inherently integrate critical safety and regulatory information into any advice it provides, minimizing risks in a laboratory setting. Think about scenarios involving hazardous chemicals, biohazardous materials, specialized equipment, or waste generation, where adherence to safety protocols and regulatory standards (e.g., OSHA, institutional biosafety guidelines) is non-negotiable."

**"Overview":** "Generate a question asking for a high-level summary of an experimental workflow. The goal is to provide a concise 'bird's-eye view' of the major stages while preserving the logical sequence. The generated <key> should be concise, containing approximately 2 to 6 steps. To achieve this, intelligently select the appropriate level of the hierarchical_protocol to summarize: if the top-level sections are few enough to fit this length, summarize their titles. If there are too many top-level sections, choose a single, major first-level section and summarize the titles of its second-level sub-sections instead."

**"Specific":** "Generate a question that asks for a granular, step-by-step breakdown of a specific, continuous segment of the experiment. The question must target exactly one major first-level section from the hierarchical_protocol. Crucially, you must select a section of appropriate length such that its detailed procedural breakdown results in a concise <key> of approximately 2 to 6 atomic steps. Avoid selecting sections that are either too simple (1 step) or overly complex (7+ steps). The <key> answer must be derived from the lowest-level procedural text within that single section, not its titles, adhering to all distillation and formatting rules."

Figure 21: Prompt used for defining each task.

[Protocol]
- Experiment Name: {exp_name}
- Background: {abstract}
- Materials and Reagents: {materials}
- Equipments: {equipment}
- Procedures: {procedure}
- Notes/Precautions: {notes}

You are now tasked with generating {num_qa} high-quality, single-turn QA pair(s) for the {type_name} category.
Task Details for Category: {type_name}
Your task is guided by the following definition: {type_instruction}

Core Instructions and Examples
1. Primary Goal: Create Rich, Focused, and Self-Contained Scenarios
Your most important goal is to ensure every QA pair is a closed logical loop. The <question> is not just a query; it is a complete problem statement. It must be constructed with enough detail that the <key> can be logically deduced from its contents alone.
- The Principle of the Complete Problem Statement: The <question> must unambiguously provide all the necessary "given" information for the problem. Before finalizing, you must confirm that the <key> is a logical consequence of the information presented within the question itself, without requiring external knowledge beyond general scientific principles. To achieve this, your question must include:
    The Specific Goal or Symptom: Clearly state the objective (e.g., "scale down the reaction") or the problem (e.g., "observed very low cell viability").
    Relevant Quantitative Data: Embed all necessary numbers, concentrations, and volumes (e.g., "scale from 3x10^6 to 0.5x10^6 cells," "the wash buffer contains 5 mM EDTA").
    Key Reagents and Equipment: Name the specific materials involved in the problem (e.g., "the GM-CSF from Peprotech is unavailable," "using a FACSCalibur flow cytometer").
    Experimental Context: Pinpoint where in the overall process the problem occurs (e.g., "immediately after the first centrifugation step," "during the 'chase' phase").
- The Principle of Focused Scope and Conciseness: The scenario you create must be solvable with a concise plan. The resulting <key> must contain between 2 and 6 steps, inclusive. This is a non-negotiable requirement. Actively design a problem that is targeted and manageable to fit this length.
- The Principle of Varied Phrasing: Use diverse phrasing (procedural inquiry, goal-oriented request, direct command, scenario-based) to avoid generating repetitive questions.

2. Critical Generation Rules
You must strictly adhere to the following rules for every QA pair you generate:
Rule 1: Context-Grounded Generation (Most Important Rule). The provided protocol is your context and foundation, not a script to be copied. Your generated <key> plan must be a novel, logical solution to the problem posed in your <question>. This new plan must be scientifically plausible and consistent with the techniques, reagents, and equipment mentioned in the protocol context. For example, a troubleshooting plan will not be in the original text; you must invent it.
Rule 2: Strict Output Structure. Each QA pair must strictly follow the required structure: <question>, <think>, <key>, <orc>, and <note> as five separate, top-level tags. All text values inside the <key> JSON must be lowercase. Each Step must contain only a single action. The <orc> tag must be a natural-language summary of the <key> tag, with each step written as a concise imperative sentence.
Rule 3: Justify Your Novel Plan in <think>. The <think> tag is where you must explain the scientific reasoning behind the new plan you created in <key>. Start by analyzing the scenario in your <question>, propose a hypothesis or strategy, and justify why the sequence of actions in your <key> is the most logical way to solve the problem.
    - Do This (Scientific Reasoning): Start by analyzing the scenario in your <question>. Then, propose a hypothesis (e.g., "The likely cause is X"). Finally, justify why the sequence of actions in your <key> is the most logical and efficient way to test that hypothesis or solve the problem.
    - Do NOT Do This (Meta-Commentary): Do not describe your generation process or refer to the source protocol directly (e.g., "I will invent a troubleshooting step..."). Your response must be a standalone scientific explanation.
Rule 4: Radical Distillation for ALL Generated Steps (Most Important Rule). The goal of the <key> tag is to achieve maximum semantic distillation. This rule applies to all steps you generate, even if they are novel. You must aggressively simplify and decompose your planned actions into atomic keywords. A parameter must be a concise modifier (e.g., '10 min', 'prewarmed', 'on ice'), not a descriptive phrase or a sentence fragment (e.g., 'until the solution turns clear').
Example of Radical Distillation:
    - Original Text: "resuspend the cells in 100ul total volume of prewarmed conditioned complete medium"
    - Bad (Literal Transcription): {"action": "resuspend", "objects": ["the cells"], "parameters": ["in 100ul total volume of prewarmed conditioned complete medium"]}
    - Okay (Simple Simplification): {"action": "resuspend", "objects": ["cells"], "parameters": ["100 µl", "prewarmed conditioned complete medium"]}
    - Excellent (Radical Distillation): {"action": "resuspend", "objects": ["cells"], "parameters": ["100 µl", "conditioned complete medium", "prewarmed"]}
Rule 5: Question-Answer Alignment. The novel plan you create in <key> must directly and completely address the specific scenario you created in <question>.
Rule 6: Acknowledge Assumptions. When creating a new plan, you will inevitably make assumptions. You must state these assumptions within the <think> tag, framing them as part of your expert planning (e.g., "This plan assumes the availability of a standard spectrophotometer to check reagent concentration...").
Rule 7: Keep the <note> Brief and Focused. The <note> tag must be a single, concise paragraph. It should highlight only the most critical and specific safety hazards directly related to the plan in <key>. Avoid generic or obvious advice.

Figure 22: Input prompt used for constructing SciRecipe.

Your previous output contained formatting errors and did not meet the required schema. Please correct the output to conform strictly to the structure and rules outlined below.
<question>
[Your generated question text here.]
</question>
<think>
[Your thought process for solving the experimental problem.]
</think>
<key>
Step 1: {"action": "verb1", "objects": ["object1"], "parameters": ["param1"]}
Step 2: {"action": "verb2", "objects": ["object2"], "parameters": ["param2"]}
...
</key>
<orc>
Step 1: Verb1 the object1 with param1.
Step 2: Verb2 the object2 with param2.
...
</orc>
<note>
[Relevant safety and handling information.]
</note>

Mandatory Rules to Follow:
1. Correct Nesting: The root must contain exactly five tags in this specific order: <question>, <think>, <key>, <orc>, and <note>. These are all top-level tags.
2. <key> Step Format: The content inside the <key> tag must be a sequence of steps. Each step must begin with the exact string Step X: (where X is a number starting from 1) followed by a newline.
3. JSON String Value (for <key>): The value for each Step X: in the <key> tag must be a single, valid JSON string.
    This JSON string must parse into a single Dictionary.
    This dictionary must contain exactly three keys: "action", "objects", "parameters".
    All string values (for action, and inside the objects/parameters lists) must be lowercase.
    - Correct Example: Step 1: {"action": "add", "objects": ["pbs"], "parameters": ["10 ml", "cold"]}
    - Incorrect Example (not a valid JSON string): Step 1: {'action': 'add', 'objects': ['pbs']} (Uses single quotes)
    - Incorrect Example (is a list, not a dictionary): Step 1: [{"action": "add", "objects": ["pbs"], "parameters": ["10 ml", "cold"]}]
4. <orc> Summary Format: The <orc> tag must be a natural language summary. The number of Step X: entries must exactly match the number of steps in <key>. Each step must be a single, concise imperative sentence that accurately reflects the corresponding step in <key>.
5. No Extra Commentary: Return only the corrected XML block starting with <question> and ending with </note>. Do not include any apologies, explanations, or any text outside of these tags.

Figure 23: System prompt for the repair module.

You are an expert scientific protocol validator and safety compliance officer AI. Your task is to rigorously review a given Question-Answer pair, which was generated from a scientific experiment protocol, against a set of strict criteria. You will focus solely on the scientific content, safety implications, and practical utility.

Your primary responsibilities are:
1) Scientific Accuracy: Verify that all scientific facts, parameters, procedures, and conditions stated in the answer are consistent with typical laboratory practices and scientifically sound. Crucially, ensure they directly derive from or are logically inferable from the original protocol context provided in the question. Identify any hallucinations, inaccuracies, or inconsistencies (e.g., incorrect calculations, mismatched concentrations, inappropriate reagent use).
2) Safety and Compliance: Critically assess the <note> section for completeness, accuracy, and relevance regarding safety hazards (chemical, biological, physical), Personal Protective Equipment, engineering controls (e.g., fume hoods), waste disposal, and general lab safety best practices. Ensure that the safety advice is specific, actionable, and aligns with standard regulatory guidelines (e.g., OSHA, institutional biosafety). Flag any missing or incorrect safety information.
3) Logical Coherence and Actionability: Evaluate the <think>, <plan>, <tool>, and <answer> sections for logical flow, feasibility, and clarity. Ensure the plan logically addresses the question, the tools are appropriate and correctly parameterized for the plan steps, and the final answer is a concise, accurate summary.
4) Clarity and Ambiguity Check: Identify any vague terms, ambiguous instructions, or unclear descriptions within the question or answer that could lead to misinterpretation or error during experimental execution. Ensure all critical parameters (e.g., concentrations, temperatures, durations, volumes) are precisely stated with correct units and no missing information.
5) Generality and Specificity Check: Assess whether the recommended steps, reagents, or equipment are appropriately general or specific for the context. Flag if a recommendation is unnecessarily restrictive, overly vague when precision is needed, or if an alternative or optimization option should have been mentioned for broader applicability (e.g., primer Tm optimization range).
6) Efficiency and Resource Optimization Check: Evaluate if the proposed experimental plan or tool usage is efficient and optimizes resource utilization (time, reagents, equipment). Flag any unnecessarily complex, redundant, or resource-intensive steps or recommendations where simpler or more cost-effective alternatives could achieve the same scientific objective without compromising quality.

You will be given a question that includes the necessary protocol context, and an answer previously generated by another AI.

Your output must be structured as follows:
<validation_report>
<accuracy_check>
[Detail any scientific inaccuracies, inconsistencies, or hallucinations found. If none, state "No significant scientific inaccuracies found."]
</accuracy_check>
<safety_compliance_check>
[Detail any missing, incorrect, or insufficient safety information, PPE, waste disposal, or compliance issues. If none, state "Safety and compliance information is adequate and accurate."]
</safety_compliance_check>
<logical_coherence_check>
[Detail any issues with the logical flow, feasibility of the plan, appropriateness of tools, or clarity of the overall answer. If none, state "Logical coherence and actionability are sound."]
</logical_coherence_check>
<clarity_ambiguity_check>
[Detail any issues with vagueness, ambiguity, or missing precise parameters/units. If none, state "All information is clear and unambiguous."]
</clarity_ambiguity_check>
<generality_specificity_check>
[Detail any issues where recommendations are inappropriately general or specific, or where useful alternatives/optimizations were overlooked. If none, state "Appropriate balance of generality and specificity."]
</generality_specificity_check>
<efficiency_resource_optimization_check>
[Detail any inefficiencies, redundancies, or sub-optimal resource usage. If none, state "Plan demonstrates good efficiency and resource optimization."]
</efficiency_resource_optimization_check>
</validation_report>

If a section has no issues, explicitly state the positive affirmation (e.g., "No issues found", "Adequate and accurate") as specified for each section. Do not leave sections empty or just say "None".

Figure 24: System prompt for the scientific review module.

Please validate the following Question-Answer pair based on the protocol context provided in the question:
<question>
{question_text}
</question>
<answer_to_validate>
{answer_text}
</answer_to_validate>

VALIDATION GUIDELINE: Apply a Principle of Generous Interpretation
Your primary goal is to validate for scientific plausibility and safety, not for perfect, literal adherence to the source protocol. Your default stance should be to approve the answer unless you find a severe and unambiguous error.
A severe error is one that, if uncorrected, would almost certainly lead to:
a) Guaranteed or highly probable experimental failure (e.g., a concentration that is off by an order of magnitude, a fundamentally wrong sequence of critical steps). If a step represents a common or scientifically valid alternative practice, it should not be flagged as an error, even if it differs from the source text.
b) A clear safety hazard (e.g., omitting a crucial warning for a highly toxic or reactive chemical).
c) Fundamentally flawed logic that would invalidate the entire experimental approach.
If you find issues that are minor, debatable, or represent a slightly different but still valid scientific approach, you should treat the section as passed and use the corresponding mandatory affirmation text. Only report issues that meet the 'severe error' threshold defined above.

CRITICAL OUTPUT RULES:
1. Your output must be a complete <validation_report> containing all six check sections.
2. If you find issues in a section, provide a concise explanation of the problems.
3. If a section has no issues, your response for that section must consist only of the exact mandatory affirmation text listed below. There should be no additional words, explanations, or leading/trailing text.
    - Correct Example:
      <accuracy_check>No significant scientific inaccuracies found.</accuracy_check>
    - Incorrect Examples:
      <accuracy_check>After review, no significant scientific inaccuracies found.</accuracy_check>
      <accuracy_check>No significant scientific inaccuracies found. The data looks solid.</accuracy_check>
Mandatory Affirmations for "No Material Errors Found":
- accuracy_check: "No significant scientific inaccuracies found."
- safety_compliance_check: "Safety and compliance information is adequate and accurate."
- logical_coherence_check: "Logical coherence and actionability are sound."
- clarity_ambiguity_check: "All information is clear and unambiguous."
- generality_specificity_check: "Appropriate balance of generality and specificity."
- efficiency_resource_optimization_check: "Plan demonstrates good efficiency and resource optimization."
Failure to adhere strictly to these rules for all six sections will result in an invalid output. Begin your validation now.

Figure 25: Input prompt for the scientific review module.

