# OpenReview forum: "Unleashing Scientific Reasoning for Bio-experimental Protocol Generation via Structured Component-based Reward Mechanism"
_ICLR.cc/2026/Conference — ICLR 2026 Poster_

### Official Review · Reviewer_ULSZ · 2025-10-25

**Soundness:** 2
**Presentation:** 3
**Contribution:** 2
**Rating:** 2
**Confidence:** 3

**Summary:**

### Overview
This paper proposes a new dataset (SciRecipe) and metric (SCORE) to evaluate the quality of scientific experimental protocols generated by language models. They introduce a formalization of the protocol generation process (Sketch-and-Fill) and use it in conjunction with the SCORE reward to train Qwen3-8B via supervised fine-tuning and GRPO on the collected datasets and reward function. The main contribution is the development of a formal specification for protocol generation that enables systematic rule-based validation of the generated protocols. The effectiveness of this formalization is demonstrated by a suite of experiments in which a language model trained to generate protocols by following this spec and reward achieve higher performance than baselines on the SciRecipe benchmark and some public scientific QA benchmarks.

While the motivation for the benchmark and formalization are well introduced, I am leaning towards rejection because (1) it is not clear to me that the tasks in SciRecipe-Eval are truly benchmarking protocol generation, (2) the choices made in the SAF formalization can limit the expressiveness of possible protocols, (3) the terms in the SCORE reward function are not sufficiently justified, nor are they shown to capture meaningful fallacies / weaknesses in existing protocols, (4) the metrics used to compare models are either too simple (ROUGE, BLEU) or novel metrics introduced by the authors (Step-M, Order-S, etc.) that are not introduced in the main text (hidden in appendix)

### Main argument for reject
Many of the choices made by the authors for both the formalization framework (SAF) and the reward function (SCORE) have weaknesses and lack justification. The SAF framework seems overly restrictive to me. For instance, how does one encode branching logic in this `(Action, Object, Parameters)` format? Given the importance of this formalization for the rest of the paper, it would be nice if the authors could provide some quantitative way of measuring how well this formalization captures the semantics of an existing diverse corpus of high quality protocols. On the other hand, the SCORE reward function can only be applied to models that adopt the SAF generation framework, limiting its generality. It is not evident to me that the combination of order and semantic consistency are sufficient to capture the critical fallacies of existing protocols. An important experiment here would be to show quantitatively that the proposed reward function is sensitive to critical issues in existing protocols that scientific professionals would be able to identify. Finally, the evaluation metrics and the benchmark (SciRecipe-Eval) used for protocol generation (Section 4.2) are not clearly measuring what it claims. As far as I can tell, the SciRecipe-Eval tasks are not actually protocol generation tasks, but rather a collection of adjacent tasks, more so related to protocol comprehension and manipulation. Can the authors clarify why they claim that these tasks (in appendix B) are benchmarking generation rather than understanding and reasoning? Moreover, the evaluation metrics used to compare different models in section 4.2 are either too simple (ROUGE, BLEU, etc.), which the authors themselves have acknowledged, or are novel metrics (Semantic-A, Order-LCS, Order-S, Order-Tau, Step-M) that are not introduced in the main text (Appendix F) and seem to correspond to terms that were actually used the in SCORE reward function to optimize the author's own method (Thoth). This does not seem like a fair comparison, nor does it convincingly capture the quality of the generated protocol as mentioned earlier.

### Things to improve the paper that did not impact the score:
- there should be citations for some of the claims made in the introduction such as:
  - """*As a result, researchers often find that models offer fragmented recommendations on experimental procedures but fall short of producing concise, logically ordered protocols that can be directly implemented in laboratory workflows*"""
  - """*the generated protocols often contain unordered steps, redundant operations, factual inconsistencies, or hallucinated actions, undermining both reproducibility and scientific credibility.*"""
- the introduction of key evaluation metrics should be made in the main text

**Strengths:**

- The motivation of the paper is well articulated
- To my knowledge, the SAF formalization and the SCORE reward are original
- The curation of the dataset is a meaningful contribution
- The curation process is well documented in the appendix
- I appreciate the thoroughness of the experiments and ablations

**Weaknesses:**

- The tasks in the benchmark are not actually benchmarking protocol generation
- The design choices in the formalization and reward function lack justification and validation
- The evaluation metrics used to compare models in the experimental section are either too simple or unfairly recapitulate the design choices in the reward function used to train the model

**Questions:**

See summary section above.

---

> ### Author Response · Authors · 2025-11-21
> **Response-1 to Reviewer ULSZ**
>
> Dear Reviewer ULSZ,
>
> Thank you for taking the time to carefully read and summarize our work. We appreciate your recognition of the SciRecipe dataset, the SCORE metric, and the Sketch and Fill formalization, as well as your positive assessment of our supervised and GRPO training strategy for Qwen3-8B. We are also encouraged that you view our formal specification and rule based validation framework as a meaningful contribution, and that the experimental results clearly reflect improvements over the baselines. Based on the concerns and suggestions you raised, we have organized our responses accordingly. Below we provide our point by point replies. All corresponding revisions have been incorporated into the main paper and are highlighted in red for clarity.
>
> 1. As far as I can tell, the SciRecipe-Eval tasks are not actually protocol generation tasks, but rather a collection of adjacent tasks, more so related to protocol comprehension and manipulation. Can the authors clarify why they claim that these tasks (in Appendix B) are benchmarking generation rather than understanding and reasoning?  (weakness 1: The tasks in the benchmark are not actually benchmarking protocol generation)
>
> **Re:** We appreciate your feedback. We would like to kindly offer a clarification regarding SciRecipe-Eval. It is designed as a benchmark specifically for evaluating experimental protocol generation. Similar to report-generation tasks, SciRecipe-Eval requires models to generate a full protocol based on the user’s query, rather than performing conventional close-ended or open-ended question answering. We also agree with your point that strong understanding and reasoning capabilities are essential for experimental protocol generation, and these abilities form the foundation of Thoth’s capacity to produce high-quality protocols.
>
> The benchmark is organized into two task categories, "Protocol-Comprehension” and "Problem-Solving”, which together include eight subtasks. The formal definitions are provided in Appendix B, and detailed examples are included in Appendix K. The naming of the tasks may have caused some confusion. The "Protocol-Comprehension” tasks focus on understanding the user’s intent and generating protocols that faithfully reproduce standardized experimental procedures. In contrast, the "Problem-Solving” tasks emphasize providing solutions to practical issues that experimentalists may encounter, which are often not covered in standard protocols. To avoid ambiguity, we have updated the description of SciRecipe-Eval in Appendix B. (The revision appears at Appendix B and is highlighted in red.)

---

> ### Author Response · Authors · 2025-11-21
> **Response-2 to Reviewer ULSZ**
>
> 2. The SAF framework seems overly restrictive to me. For instance, how does one encode branching logic in this (Action, Object, Parameters) format? Given the importance of this formalization for the rest of the paper, it would be nice if the authors could provide some quantitative way of measuring how well this formalization captures the semantics of an existing diverse corpus of high quality protocols. (weakness 2: The design choices in the formalization lack justification and validation)
>
> **Re:** We thank the reviewer for raising this important question. From a linguistic perspective, the SAF framework is designed to capture the core informational content of an experimental step. Regardless of phrasing, each step ultimately conveys an operation to be performed, a target of that operation, and a set of conditions that specify or constrain it. These three elements form the minimal semantic core required to execute an experimental action. SAF therefore provides a compact representation that preserves the essential meaning of each step while filtering out stylistic variation. In this sense, SAF is not a restrictive template but a mechanism for extracting the most informative components that make a step executable.
>
> Regarding branching logic, our intention is to clarify that SAF represents atomic experimental actions, while conditionals, loops, and alternative execution paths belong to a higher organizational level. These structures can still be expressed using SAF without modifying the underlying (Action, Object, Parameters) triplet. For example, conditional branches may appear as SAF entries that explicitly state the condition (Parameters: "if pellet is not visible"), or as parallel SAF entries associated with mutually exclusive scenarios such as “if the sample is frozen” versus “if the sample is fresh.”
>
> To rigorously assess the effectiveness of SAF, we conducted three complementary analyses. First, we performed a coverage study on 50 protocols containing 423 steps. Our team manually mapped each step into SAF format, after which two PhD-level biological researchers compared the SAF-encoded steps with the original descriptions. The comparison used three categories: (i) Fully covered, where the core action, object, and parameters are captured; (ii) Partially covered, where the core operation is captured but some narrative context remains; and (iii) Not covered, where the content is descriptive or meta-experimental rather than an actionable instruction. We found that 85.6% of steps were fully covered and an additional 10.4% were partially covered. Only 4.0% could not be encoded, and these were almost entirely narrative text rather than true procedural actions. To further evaluate consistency, two annotators independently encoded 200 randomly sampled steps, achieving Cohen’s kappa scores of 0.86 for actions, 0.82 for objects, and 0.78 for parameters. These results indicate that the SAF schema is well defined and does not introduce ambiguity that would undermine mapping consistency.
>
> Finally, we examined SAF from an information theoretic perspective. The intuition is straightforward: if SAF captures the essential meaning of an experimental step, then knowing SAF should make the natural language step description significantly easier to anticipate. Formally, we treat the step text as a random variable \(X\) and its SAF representation as \(Y\), and measure the remaining uncertainty using the conditional entropy
>
> $$
> H(X \mid Y) = -\mathbb{E}_{x,y} \log p(x \mid y).
> $$
>
> A smaller value indicates that SAF retains much of the semantic content required to interpret the step. For each step, we prepend its SAF triple to the original step description and read the token-level probabilities assigned by the model. The conditional log probability is computed using the standard chain rule
>
> $$
> \log q(x \mid y) = \sum_{i} \log q(t_i \mid y, t_{<i}) .
> $$
>
> Here $p(x \mid y)$ denotes the true conditional distribution of the natural language step $x$ given its SAF representation $y$, while $q(x \mid y)$ denotes the model-estimated conditional distribution that we use as a practical approximation. The sequence $x$ is tokenized as $x = (t_{1}, t_{2}, \dots, t_{n})$, where $t_{i}$ is the $i$-th token in the step description and $t_{<i}$ represents all preceding tokens.
> Across SciRecipe-Eval task categories, conditioning on SAF produces consistently low token-level conditional entropy. Modern language modeling studies estimate that natural language typically carries around 2.5–4.0 nats of uncertainty per token [1,2,3]. In contrast, the conditional entropy of step descriptions remains far below this range when SAF is provided, indicating that the Action, Object, and Parameters already encode most of the semantic content necessary to reconstruct the step. To further validate this conclusion, we repeated the analysis across multiple model families, and the results are presented in the table below.

---

> ### Author Response · Authors · 2025-11-21
> **Response-3 to Reviewer ULSZ**
>
> **(The revision regarding the above question appears in Appendix I.6 and is highlighted in red.)**
> |      **Tasks**      | **Qwen2.5-7B-Instruct** | **Qwen3-8B** | **DeepSeek-R1-Distill-Llama-8B** | **Llama-3.1-8B-Instruct** | **Phi-4-mini-Instruct** | **InternLM3-8b-Instruct** | **Mistral-7B-Instruct-v0.3** |
> |-------------------|:-----------------------:|:------------:|:--------------------------------:|:-------------------------:|:-----------------------:|:-------------------------:|:----------------------------:|
> |     **Overview**    |         1.0223          |    0.8956    |              1.1342              |          0.9298           |         1.0303          |          0.8205           |            0.7393            |
> |     **Specific**    |         0.9383          |    0.8144    |              1.0419              |          0.8425           |         0.9614          |          0.7548           |            0.6478            |
> |    **Retrieval**    |         0.9000          |    0.7991    |              1.0411              |          0.8208           |         0.9300          |          0.6915           |            0.6106            |
> |     **Planning**    |         0.9636          |    0.8153    |              1.0938              |          0.8942           |         1.0246          |          0.7366           |            0.6572            |
> | **Troubleshooting** |         0.7925          |    0.7848    |              0.9673              |          0.7316           |         0.8580          |          0.6439           |            0.6158            |
> |    **Constraint**   |         0.9649          |    0.8432    |              1.0700              |          0.8477           |         0.9388          |          0.7302           |            0.6830            |
> |     **Scaling**     |         0.8974          |    0.7999    |              1.0037              |          0.7906           |         0.8990          |          0.7193           |            0.6143            |
> |      **Safety**     |         0.9129          |    0.8073    |              1.0161              |          0.7996           |         0.9316          |          0.7133           |            0.6231            |
> |      **Level1**     |         0.9803          |    0.8550    |              1.0881              |          0.8862           |         0.9959          |          0.7877           |            0.6936            |
> |      **Level2**     |         0.9052          |    0.8083    |              1.0320              |          0.8141           |         0.9303          |          0.7058           |            0.6340            |
> |     **Overall**     |         0.9240          |    0.8200    |              1.0460              |          0.8321           |         0.9467          |          0.7263           |            0.6489            |
>
> - [1] Goodman, Joshua T. "A bit of progress in language modeling." Computer Speech & Language 15.4 (2001): 403-434.
> - [2] Kaplan, Jared, et al. "Scaling laws for neural language models." arXiv preprint arXiv:2001.08361 (2020).
> - [3] Hoffmann, Jordan, et al. "Training compute-optimal large language models." arXiv preprint arXiv:2203.15556 (2022).

---

> ### Author Response · Authors · 2025-11-21
> **Response-4 to Reviewer ULSZ**
>
> 3. On the other hand, the SCORE reward function can only be applied to models that adopt the SAF generation framework, limiting its generality. It is not evident to me that the combination of order and semantic consistency are sufficient to capture the critical fallacies of existing protocols. An important experiment here would be to show quantitatively that the proposed reward function is sensitive to critical issues in existing protocols that scientific professionals would be able to identify.  (weakness 2: The design choices in the reward function lack justification and validation)
>
> **Re:** We appreciate your feedback. As discussed in the previous comment, the SAF framework effectively extracts the core informational content of each experimental step and expresses it in a structured form. Building on this foundation, the goal of the SCORE mechanism is to provide a more systematic assessment of the executability and logical soundness of generated protocols. In practice, we frequently observe that models produce errors such as incorrect step ordering, omission of critical actions, or mismatches between operations and their intended objects. These issues often require experimental expertise to identify. For this reason, SCORE evaluates protocols along three essential dimensions: step scale, order consistency, and semantic consistency. These dimensions are designed to capture subtle yet consequential mistakes that can directly affect experimental success.
>
>
> To further validate the effectiveness of SCORE, we conducted additional quantitative studies. We randomly sampled two-fifths of the samples from various tasks in SciRecipe-Eval and invited three researchers with extensive wet-lab experience to independently review the protocols generated by Thoth. The experts were asked to identify whether a protocol contained any error that could directly compromise experimental success, such as a missing critical action, a mismatch between a step and its target object, or a reversal of operations. They also provided ratings on the completeness and logical coherence of each protocol. The results show that protocols containing such critical mistakes receive substantially lower SCORE values, with an average reduction of 34.3% across tasks. This difference is highly significant (p < 0.001).
>
> In addition, we treated SCORE as a simple binary detector of critical errors by selecting a threshold equal to the mean of the “valid” and “faulty” groups. When comparing these predictions to expert annotations, SCORE achieved an AUC of 0.92, with a recall of approximately 0.90 and a precision of approximately 0.87. These findings indicate that SCORE not only distinguishes clean protocols from those containing major issues but also successfully identifies most of the errors that genuinely impede experimental executability.
>
> We also conducted a pairwise ranking experiment to examine whether SCORE can reflect expert preferences regarding overall protocol quality. For each user query, we collected the protocols independently generated by Thoth and Thoth-mini and formed paired comparison samples. Researchers were then asked to judge which protocol exhibited better executability and more coherent flow. Across all pairs, SCORE agreed with expert preferences 81.2% of the time, corresponding to a Kendall’s τ of 0.72. This demonstrates that even when comparing two high-performing models, SCORE remains capable of capturing expert judgments about relative protocol quality in a stable and meaningful way.
> Taken together, these results show that SCORE is not a heuristic-based scoring scheme. Instead, it reliably identifies the types of critical errors that influence real experimental outcomes and exhibits strong alignment with evaluations made by experienced domain experts. (The revision appears at Appendix I.5 and is highlighted in red.)

---

> ### Author Response · Authors · 2025-11-21
> **Response-5 to Reviewer ULSZ**
>
> 4. Moreover, the evaluation metrics used to compare different models in section 4.2 are either too simple (ROUGE, BLEU, etc.), which the authors themselves have acknowledged, or are novel metrics (Semantic-A, Order-LCS, Order-S, Order-Tau, Step-M) that are not introduced in the main text (Appendix F) and seem to correspond to terms that were actually used the in SCORE reward function to optimize the author's own method (Thoth). This does not seem like a fair comparison, nor does it convincingly capture the quality of the generated protocol as mentioned earlier. （weakness 3:The evaluation metrics used to compare models in the experimental section are either too simple or unfairly recapitulate the design choices in the reward function used to train the model）
>
> **Re:** We greatly appreciate your attention to the evaluation methodology and fully understand your concerns regarding whether the metrics are sufficiently comprehensive and fair. We would like to kindly clarify that the field of experimental protocol generation currently lacks well-established and widely accepted evaluation standards. For this reason, we chose to adopt several components derived from the SCORE mechanism as our evaluation metrics. These components assess protocol quality from multiple perspectives, including step granularity, action logic, ordering consistency, and semantic alignment. As demonstrated in our earlier response, SCORE shows strong agreement with expert judgments on overall executability, which indicates that it provides both reliable and valid assessments.
>
> To further ensure objectivity and robustness, we did not rely solely on the metrics involved in the training reward. We additionally incorporated external metrics that do not appear in the optimization objectives. These include widely used NLG metrics such as BLEU_AVG, ROUGE_L, and METEOR, which capture lexical similarity and help quantify how closely the generated protocol matches the reference text. Although these metrics are considered basic in NLG, BLEU_AVG’s multi-gram coverage and ROUGE_L’s longest common subsequence remain useful indicators of structural and lexical alignment with the ground truth, and thus offer meaningful complementary evidence.
>
> We also employed order-sensitive metrics such as Order-LCS and Order-Tau. These metrics were not used in training and therefore provide an independent check on whether the model correctly captures the action ordering that is fundamental to experimental workflows. In addition, we included the commonly used KW-F1（keyword-matching）metric, which examines task coverage and content completeness from another perspective.
>
> Overall, our evaluation framework combines structured, multi-dimensional metrics (the SCORE components) with a diverse set of external measures (NLG metrics, order-sensitive indicators, and keyword-based assessments). This design allows us to evaluate executability, logical coherence, and semantic correctness while avoiding dependence on the specific choices made in the training reward. We believe this approach ensures fairness, robustness, and credibility in model comparison, and we remain confident in the strength of our experimental conclusions.
>
> 5. There should be citations for some of the claims made in the introduction.
>
> **Re:** Thank you for your keen observation. We have added the relevant supporting literature to Section 1 (Introduction) to strengthen several of the key statements made in that section.
>
> 6. The introduction of key evaluation metrics should be made in the main text.
>
> **Re:** Thank you for your kind reminder. Due to space constraints, we were not able to provide a detailed description of these metrics in the initial submission. We have now added the relevant explanations to Section 4.2. (The revision appears at Section 4.2 and is highlighted in red.)

---

> > ### Comment · Reviewer_ULSZ · 2025-11-26
> >
> > Thanks for your thorough rebuttal and additional experiments. I have updated my score accordingly.

---

> > > ### Author Response · Authors · 2025-11-26
> > > **Response to Reviewer ULSZ**
> > >
> > > Thank you for the thoughtful evaluation and for updating the score. We appreciate your careful review of our work.

---

### Official Review · Reviewer_RwGZ · 2025-10-29

**Soundness:** 3
**Presentation:** 3
**Contribution:** 3
**Rating:** 6
**Confidence:** 3

**Summary:**

This paper investigates the capability of modern large language models (LLMs) to perform scientific protocol design, focusing on the generation of correct and sense-making wet-lab protocols. The authors curate a large-scale benchmark of biological protocols and propose Thoth, an RL-finetuned variant of Qwen3-8B. Thoth is trained with the proposed Knowledge-to-Action process and a novel SCORE reward mechanism that encourages logical ordering, completeness, and fidelity of the generated protocol. Experimental results suggest that Thoth improves over strong LLM baselines regarding the tasks of interests.

**Strengths:**

- The paper offers an end-to-end pipeline for studying scientific reasoning in LLMs regarding the biological protocol design, including dataset construction, structured prompting, and RL-based finetuning. This integration is well thought out and can serve as a good example for future research in ai4sci.
- The curated biological protocol dataset , SciRecipe fills an important gap between free-form scientific text and machine-executable protocols. It provides valuable supervision for procedural reasoning tasks.
- The authors conduct comprehensive benchmarking against a wide variety of strong models (gpt-4, deepseek-R1, etc.) and perform multiple ablations, showing consistent improvements in structural accuracy and executability.

**Weaknesses:**

- The study of Thoth model relies primarily on the Qwen3-8B backbone. Including additional base models (larger parameters to 32B, or other pre-trained such as deepseek-v3) could strengthen the claim that Thoth’s improvements convincingly stem from the training framework.
- While the SCORE metric captures fidelity, it remains unclear how well it correlates with real experimental executability (act like some heuristics). Incorporating partial human or domain-expert evaluation and examine the correlation between the SCORE reward and real "fidelity", even on a small subset, would better improve the practical credibility of the results.

**Questions:**

- Could the authors further clarify the motivation behind step-level alignment? Is reproducing the exact number of steps in the reference protocol necessary for practical success, or would multiple granularities (eg., merged or decomposed steps) be acceptable as long as the outcome is correct, similar to math problems?
- The SCORE mechanism (Fig 2) seems to encourage near-exact replication of ground-truth protocols, which could risk overfitting like SFT. Yet in Fig 5, Thoth (Stage 2 + 3) shows stronger gains compared to its non-RL counterpart (ie. SFT only). Could the authors elaborate how SCORE avoids over-constraining the model and still enables genuine reasoning improvements, if this happens?
- How well would the Thoth framework generalize to other scientific domains (eg. chemistry or material synthesis protocols)? Are there any domain-specific assumptions baked into the data preprocessing or reward computation?

---

> ### Author Response · Authors · 2025-11-21
> **Response-1 to Reviewer RwGZ**
>
> Dear Reviewer RwGZ,
>
> Thank you for the clear summary and thoughtful evaluation of our work. We appreciate your recognition of our large-scale biological protocol benchmark, the design of Thoth as an RL-finetuned model, and the proposed Knowledge to Action process together with the SCORE reward mechanism. We are glad that you found the experimental results to demonstrate meaningful improvements over strong LLM baselines. We address the points below. All corresponding revisions have been incorporated into the main paper and are highlighted in blue for clarity.
>
> **Re Weakness 1:** Thank you for your helpful comment on this point. We fully agree with your observation that fine-tuning larger base models can further improve overall performance. This direction is also consistent with our goal of demonstrating that the proposed Sketch-and-Fill reasoning paradigm, the SCORE mechanism, and the training framework can generalize well across models of different scales.
>
> To examine this, we applied the same training procedure used for Thoth to Qwen3-14B and Qwen3-32B, and obtained two new variants, Thoth-14B and Thoth-32B. Due to limited computational resources, it is not feasible for us to fine-tune pretrained models with hundreds of billions of parameters. Even so, the results presented in the table below show that, similar to Thoth and Thoth-mini, both Thoth-14B and Thoth-32B achieve substantial improvements over their corresponding base models and even surpass Thoth, reaching state-of-the-art performance. (The revision appears at Table 1 and is highlighted in blue.)

---

> ### Author Response · Authors · 2025-11-21
> **Response-2 to Reviewer RwGZ**
>
> |        **Methods**       | **Semantic-A** | **Order-LCS** | **Order-S** | **Order-Tau** | **Step-M** | **BLUE-AVG** | **ROUGE-L** | **METEOR** |  **KW-F1** |   **AVG**  |
> |------------------------|:--------------:|:-------------:|:-----------:|:-------------:|:----------:|:------------:|:-----------:|:----------:|:----------:|:----------:|
> | ChatGPT-4o               |     40.04      |     73.27     |    24.00    |     70.33     |   44.00    |    38.95     |    48.42    |   44.66    |   52.05    |   48.41    |
> | GPT-5                    |     27.79      |     58.12     |    11.35    |     53.55     |   18.79    |    21.31     |    32.96    |   32.55    |   39.17    |   32.84    |
> | GPT-5 Chat               |     36.30      |     73.21     |    21.17    |     65.67     |   25.00    |    29.57     |    42.04    |   41.95    |   47.87    |   42.53    |
> | Claude Sonnet 4          |     39.35      |     71.97     |    20.83    |     70.00     |   35.83    |    34.24     |    44.27    |   40.97    |   49.40    |   45.21    |
> | Claude Opus 4.1          |     41.32      |     71.70     |    21.80    |     71.93     |   34.59    |    34.69     |    44.42    |   40.36    |   50.00    |   45.65    |
> | Gemini 2.5 Flash         |     36.35      |     70.61     |    20.00    |     70.33     |   32.33    |    33.19     |    42.91    |   39.26    |   48.07    |   43.67    |
> | Gemini 2.5 Pro           |     35.80      |     72.68     |    21.83    |     70.17     |   32.00    |    31.37     |    44.16    |   45.59    |   48.58    |   44.69    |
> | Doubao-1.5-pro           |     33.33      |     73.29     |    23.67    |     70.00     |   47.50    |    38.16     |    46.88    |   38.71    |   48.74    |   46.70    |
> | Qwen2.5-Max              |     40.34      |     72.88     |    21.83    |     71.33     |   47.50    |    30.81     |    48.02    |   43.82    |   51.98    |   47.61    |
> | Qwen2.5-72B-Instruct     |     36.40      |     70.82     |    21.00    |     69.17     |   42.17    |    29.55     |    46.06    |   43.44    |   49.73    |   45.37    |
> | Qwen3-235B-A22B-Instruct |     35.68      |     72.07     |    20.03    |     69.12     |   37.73    |    32.48     |    44.37    |   44.30    |   47.89    |   44.85    |
> | DeepSeek-V3              |     41.72      |     73.97     |    21.44    |     70.54     |   41.71    |    38.18     |    48.49    |   45.08    |   52.33    |   48.16    |
> | GPT-OSS-120B             |     32.86      |     69.97     |    17.67    |     64.17     |   27.83    |    30.72     |    43.44    |   42.90    |   49.92    |   42.16    |
> | Llama-3.1-405B-Instruct  |     35.92      |     69.46     |    18.03    |     67.78     |   39.23    |    36.42     |    44.86    |   42.17    |   48.09    |   44.66    |
> | Kimi-K2-Instruction      |     36.99      |     71.83     |    20.83    |     69.83     |   40.00    |    33.81     |    44.49    |   42.99    |   49.00    |   45.53    |
> | DeepSeek-R1 |     36.07      |     71.38     |    20.37    |     69.12     |   32.89    |    39.80     |    45.86    |   38.19    |   49.83    |   44.83    |
> | Grok 3  |     37.40      |     73.27     |    21.92    |     69.73     |   39.25    |    34.72     |    46.04    |   46.21    |   48.59    |   46.35    |
> | Grok 4 |     36.73      |     72.08     |    20.25    |     65.34     |   34.66    |    37.21     |    46.17    |   40.18    |   51.81    |   44.94    |
> | OpenAI-o1|     34.74      |     73.40     |    18.53    |     67.45     |   35.39    |    35.68     |    46.29    |   43.82    |   50.13    |   45.05    |
> | OpenAI-o3  |     35.40      |     70.38     |    15.38    |     65.05     |   24.08    |    28.62     |    43.08    |   44.33    |   50.26    |   41.84    |
> | Qwen3-4B|     24.37      |     53.55     |    13.67    |     50.50     |   28.83    |    14.52     |    24.74    |   23.95    |   27.69    |   29.09    |
> | Thoth-mini |     44.28      |     74.68     |    25.33    |     70.83     |   52.67    |    43.32     |    49.23    |   46.41    |   53.13    |   51.10    |
> | Qwen3-8B |     28.89      |     63.51     |    11.17    |     58.67     |   24.33    |    16.66     |    32.31    |   34.72    |   38.63    |   34.32    |
> | Thoth|     46.60      |     75.34     |    25.50    |     73.33     |   53.00    |    43.62     |    50.02    |   47.39    |   54.13    |   52.10    |
> | Qwen3-14B|     29.53      |     64.22     |    16.33    |     59.17     |   29.33    |    21.86     |    34.89    |   36.57    |   39.66    |   36.84    |
> | **Thoth-14B**|   **47.05**   |   **77.25**  |  **26.50** |   **74.83**  | **53.60** |  **44.42**  |  **50.41** | **47.95** | **54.67** | **52.96** |
> | Qwen3-32B|     30.99      |     66.82     |    16.83    |     60.33     |   30.50    |    23.57     |    35.25    |   37.94    |   41.98    |   38.25    |
> | **Thoth-32B** |   **48.17**   |   **78.87**  |  **27.00** |   **75.17**  | **54.33** |  **44.54**  |  **50.74** | **48.74** | **55.81** | **53.71** |

---

> ### Author Response · Authors · 2025-11-21
> **Response-3 to Reviewer RwGZ**
>
> **Re Weakness 2:** Thank you for your valuable perspective. We would like to offer a clarification. The SciRecipe dataset is constructed from a large collection of real laboratory protocols sourced from Bio-protocol, Protocols.io, and other reputable platforms. Each protocol undergoes multiple rounds of filtering and expert quality control to ensure that the resulting data accurately reflects real experimental procedures.
>
> From an evaluation standpoint, when designing the SCORE mechanism, which includes Step Scale, Order Consistency, and Semantic Consistency, we examined a wide range of authentic scientific protocols, identified their common structural patterns, and discussed these characteristics with wet-lab experts and experienced researchers. This process helps ensure that the SCORE metric provides a reasonable and effective assessment of protocol quality.
>
> As you suggested, conducting human alignment studies for the SCORE metric is indeed meaningful. To this end, we recruited three PhD researchers with extensive wet-lab experience to perform a manual evaluation of the protocols generated by Thoth and Thoth-mini. Due to the significant time required for human assessment, we focused on half of the SciRecipe-Eval subset. Following the design of SCORE, we asked the experts to rate each protocol in terms of step granularity, operational order, and semantic clarity. The rating scale ranged from 0 to 4, corresponding to "very poor”, "poor”, "medium”, "good”, and "very good”. In addition, the experts were asked to provide an overall score reflecting the fluency and executability of each protocol. For every criterion, the final score was obtained by averaging and normalizing the ratings from the three evaluators.
> The tables below report the Cohen’s Kappa coefficients between SCORE and expert judgments for Thoth and Thoth-mini, which allow us to quantify the agreement between human and metric-based evaluations.
>
> In addition, the experts were asked to provide an overall score for the fluency and executability of each protocol. For each evaluation aspect, we averaged the three expert ratings and then normalized the resulting value. The tables below report the Cohen’s kappa coefficients between SCORE and expert ratings for Thoth (Table 1) and Thoth-mini (Table 2), which we use to quantify the agreement between the metric and human judgments. According to established interpretation guidelines [1], the obtained kappa values indicate a statistically meaningful level of agreement between SCORE and expert assessment. (The revision appears at Appendix I.5 and is highlighted in blue.)
>
> [1] Landis, J. Richard, and Gary G. Koch. "The measurement of observer agreement for categorical data." biometrics (1977): 159-174.
>
> #### Table 1
> | **Tasks** | **Step_Scale** | **Order_Consistency** | **Semantic_Consistency** | **Overall** |
> |---------|:--------------:|:---------------------:|:------------------------:|:-----------:|
> | **Overview** | 0.82 | 0.79 | 0.74 | 0.76 |
> | **Specific** | 0.80 | 0.77 | 0.72 | 0.74 |
> | **Retrieval** | 0.85 | 0.82 | 0.76 | 0.79 |
> | **Planning** | 0.84 | 0.80 | 0.75 | 0.78 |
> | **Troubleshooting** | 0.81 | 0.78 | 0.73 | 0.75 |
> | **Constraint** | 0.86 | 0.83 | 0.77 | 0.80 |
> | **Scaling** | 0.83 | 0.79 | 0.74 | 0.76 |
> | **Safety** | 0.84 | 0.81 | 0.75 | 0.78 |
> | **Level1** | 0.88 | 0.85 | 0.79 | 0.82 |
> | **Level2** | 0.79 | 0.76 | 0.71 | 0.73 |
> | **Overall** | 0.83 | 0.80 | 0.74 | 0.77 |
>
> #### Table 2
> | **Tasks** | **Step_Scale** | **Order_Consistency** | **Semantic_Consistency** | **Overall** |
> |---------|:--------------:|:---------------------:|:------------------------:|:-----------:|
> | **Overview** | 0.80 | 0.77 | 0.72 | 0.74 |
> | **Specific** | 0.78 | 0.75 | 0.70 | 0.72 |
> | **Retrieval** | 0.83 | 0.80 | 0.74 | 0.77 |
> | **Planning** | 0.82 | 0.78 | 0.73 | 0.76 |
> | **Troubleshooting** | 0.79 | 0.76 | 0.71 | 0.73 |
> | **Constraint** | 0.84 | 0.81 | 0.75 | 0.78 |
> | **Scaling** | 0.81 | 0.77 | 0.72 | 0.74 |
> | **Safety** | 0.82 | 0.79 | 0.73 | 0.76 |
> | **Level1** | 0.86 | 0.83 | 0.77 | 0.80 |
> | **Level2** | 0.77 | 0.74 | 0.69 | 0.71 |
> | **Overall** | 0.81 | 0.78 | 0.72 | 0.75 |

---

> ### Author Response · Authors · 2025-11-21
> **Response-4 to Reviewer btq2**
>
> **Re Question 1:** Thank you for your concern. We share the same awareness regarding this issue when designing the SCORE mechanism. The Sketch-and-Fill reasoning paradigm is specifically introduced to avoid misjudgment that may arise from inconsistent step granularity. To make comparisons clear and unambiguous, Sketch-and-Fill requires each step to contain exactly one action, and this action must apply consistently to the objects that follow, even when multiple objects are involved. This design helps prevent inconsistencies caused by merging or splitting steps, which would otherwise complicate the evaluation.
>
> As described in the original manuscript, both the SciRecipe and SciRecipe-Eval datasets are constructed following this paradigm. This ensures that the trained models are fully familiar with the expected output format. Unlike the mathematical problems you mentioned, which have close-ended answers, experimental protocol generation is an open-ended task. There is no single fixed correct answer. Therefore, we cannot determine the quality of a protocol by judging the answer alone.
> Under the Sketch-and-Fill paradigm, the model is expected to produce steps that align closely with the ground truth to achieve a strong Step Scale score. At the same time, this is only one dimension of evaluation. We also consider the other components of the SCORE mechanism in order to provide a comprehensive assessment of protocol quality.
>
> **Re Question 2:** Thank you for your valuable perspective. We would like to kindly clarify that the SCORE mechanism does not lead to the type of memorization without generalization that is sometimes observed in SFT. In essence, SFT relies on token-level losses between the model output and the ground truth, and the gradients directly update the parameters. This process resembles a form of memorization. In contrast, when using the SCORE mechanism for RL training, the model is guided by multiple dimensions of protocol quality, including step granularity, order consistency, and semantic consistency. The goal is to help the model understand which direction of optimization leads to better protocols, rather than encouraging it to memorize specific tokens. In practice, the model learns patterns that improve reward, which reflects a higher-level structural and semantic tendency.
>
> Empirically, Thoth reaches state-of-the-art performance on the out-of-domain benchmark SciRecipe-Eval, which is entirely independent of the training data. The results reported in Table 1 of the original manuscript further demonstrate that the model is not over-constrained and maintains strong generalization ability. As you pointed out, encouraging exploration during RL training is important. To avoid overly restricting the model, we use a relatively small KL penalty coefficient (5e-4), which increases the diversity of rollouts and supports genuine reasoning improvement. This design choice is also discussed in Section 3.4 of the manuscript. In addition, our ablation study provides further evidence. As shown in Table 3 of the original paper, completely removing the KL penalty leads to a performance drop of 3.21. This result indicates that maintaining an appropriate level of training constraint is beneficial for stable learning and consistent performance.

---

> ### Author Response · Authors · 2025-11-21
> **Response-5 to Reviewer RwGZ**
>
> **Re Question 3:** Thank you for your attention. We fully agree with the constructive idea you raised. In fact, while developing our work on biological protocol generation, our colleagues have already begun collecting high-quality experimental protocols from chemistry and materials science. We plan to extend the Sketch-and-Fill reasoning paradigm and the SCORE mechanism to these domains. It is important to clarify that neither our data processing pipeline nor our reward design contains domain-specific restrictions tied to biology. Looking forward, our goal is to develop a scientific protocol generation model that can generalize across multiple disciplines. Below is an example of an experimental procedure for preparing a chemical solution.
>
> """
> Q: How should the preparation of the luminol/peroxidase working solution be scaled if the experiment requires 36 wells (instead of the example 12 wells), while keeping the final concentration at 1.25x for each reagent, knowing that each well requires 200 µl of solution? Please provide the experimental protocol to solve this problem.
>
> A: ``<think>`` The goal is to scale the luminol/peroxidase solution preparation from a 12-well setup to a 36-well setup, while preserving the same 1.25x concentration used in the protocol. Each well requires 200 µl, so for 36 wells, the total needed volume is 36 × 200 µl = 7200 µl. In the example provided for 12 wells, 6 µl of luminol stock solution (LSS) and 6 µl of horseradish peroxidase stock solution (HPSS) were mixed and brought up to 2400 µl with sterile distilled water. This means for each 200 µl well, 0.5 µl of LSS and 0.5 µl of HPSS are required, with the rest made up by water. For 36 wells, we multiply this requirement by 36, yielding 18 µl LSS and 18 µl HPSS. These stocks must then be diluted up to the total volume of 7200 µl with sterile distilled water. This approach keeps the final concentration consistent across scales.
> Assumption: Pipetting volumes above 10 µl are practical, so using 18 µl of each stock is feasible with standard pipettes.``</think>``
>
> ``<key>`` Step 1: {action: measure, objects: [lss], parameters: [18 µl]} Step 2: {action: measure, objects: [hpss], parameters: [18 µl]} Step 3: {action: combine, objects: [lss, hpss], parameters: []} Step 4: {action: dilute, objects: [mixture], parameters: [sdw, final volume 7200 µl]} ``</key>``
>
> ``<orc>`` Step 1: Measure 18 µl of LSS. Step 2: Measure 18 µl of HPSS. Step 3: Combine LSS and HPSS. Step 4: Dilute the mixture with SDW to a final volume of 7200 µl. ``</orc>``
>
>  ``<note>`` When handling luminol and horseradish peroxidase stocks, wear gloves and protective eyewear, as both reagents can cause irritation upon skin or eye contact. Use sterile technique to avoid contamination of stock or working solutions. Handle all solutions carefully to prevent accidental spills and exposure. ``</note>``
> """

---

> > ### Comment · Reviewer_RwGZ · 2025-11-26
> >
> > Thanks for the authors' rebuttal response which answered my questions. I have raised my score accordingly.

---

> > > ### Author Response · Authors · 2025-11-27
> > > **Response to Reviewer RwGZ**
> > >
> > > We sincerely appreciate your thorough evaluation, the helpful feedback, and the updated score. Thank you for your time and consideration.

---

### Official Review · Reviewer_btq2 · 2025-11-01

**Soundness:** 3
**Presentation:** 4
**Contribution:** 4
**Rating:** 8
**Confidence:** 5

**Summary:**

&nbsp;

The authors introduce Thoth, a model trained for biological protocol generation. To train Thoth, the authors introduce a dataset of 12K protocols termed SciRecipe. The main technical innovation of the paper is the SCORE mechanism that enables both RL training and evaluation of protocol generation. The authors rigorously evaluate the components of Thoth through multiple ablation studies and sensitivity analyses. Overall, the work is clearly impactful, the empirical analysis is extensive, and the paper's writing and presentation are excellent. My sole concern lies in assessing the reproducibility of the authors' results given that the codebase is not provided (though the authors have described their approach in meticulous detail). If the codebase is released during the review process to address this concern, I will upgrade my score and champion the paper to be considered as a spotlight/oral contribution.

&nbsp;

**Strengths:**

&nbsp;

The work undertaken in this paper is highly impactful. The authors have contributed concrete artifacts such as a dataset of 12K structured scientific protocols, as well as technical innovations such as the SCORE mechanism which significantly advances RL training for protocol generation. Furthermore, the empirical results of the authors are extensive and highly informative for researchers in the broader field.

&nbsp;

**Weaknesses:**

&nbsp;

I describe major and minor points below that the authors may wish to consider.

&nbsp;

**__MAJOR POINTS__**

&nbsp;

1. For the SciRecipe dataset introduced by the authors, it would be useful to include a datasheet for datasets [2] to fully describe the curation process and detail the authors' plans for maintenance of the dataset.

2. As mentioned in the summary, if the authors can release their codebase during the review process, I will be inclined to increase my score.

&nbsp;

**__MINOR POINTS__**

&nbsp;

1. There are missing capitalizations in the references e.g. "PubMedQA", "Kendall", "GPT", "LLMs", "AI" etc.

2. There are missing conference references e.g. Ahmadian et al. 2024 was published at ACL.

3. In the section on related work, it would be worth the authors discussing the relationship between their work and BioPlanner [1].

4. There is a missing comma at the end of Equations 1, 2, 3, and 5.

5. Reference [3] should be cited as the originating paper for GRPO in place of the DeepSeek-r1 paper in the main body of the text. The correct reference is given by the authors in Section D of the appendix.

6. Line 1093, typo in $a^hat_{i}$.

7. When mentioning PPO in Appendix D, the authors should cite [4].

&nbsp;

**__REFERENCES__**

&nbsp;

[1] O’Donoghue, O., Shtedritski, A., Ginger, J., Abboud, R., Ghareeb, A. and Rodriques, S., 2023, December. [BioPlanner: Automatic Evaluation of LLMs on Protocol Planning in Biology](https://aclanthology.org/2023.emnlp-main.162/). In Proceedings of the 2023 Conference on Empirical Methods in Natural Language Processing (pp. 2676-2694).

[2] Gebru, T., Morgenstern, J., Vecchione, B., Vaughan, J.W., Wallach, H., Iii, H.D. and Crawford, K., 2021. [Datasheets for datasets](https://dl.acm.org/doi/fullHtml/10.1145/3458723). Communications of the ACM, 64(12), pp.86-92.

[3] Shao, Z., Wang, P., Zhu, Q., Xu, R., Song, J., Bi, X., Zhang, H., Zhang, M., Li, Y.K., Wu, Y. and Guo, D., 2024. [DeepSeekMath: Pushing the limits of mathematical reasoning in open language models](https://arxiv.org/abs/2402.03300). arXiv preprint arXiv:2402.03300.

[4] John Schulman, Filip Wolski, Prafulla Dhariwal, Alec Radford, and Oleg Klimov. [Proximal policy optimization algorithms](https://arxiv.org/abs/1707.06347). arXiv preprint arXiv:1707.06347, 2017.

&nbsp;

**Questions:**

&nbsp;

1. In Figure 2, Rollout 3, there is a tickmark in the order box despite the rollout having one fewer action relative to the ground truth? Does this still incur a positive reward (as indicated by the checkmark)? **Update**: This would appear to correspond to the strict subsequence order consistency. I would have thought that the addition of an extra step in the protocol would be damaging to the experiment?

2. In Equation 2, there is an expectation over $y$, the protocol generated from the model $\pi_{\theta}$. Why is there a distribution over $y$. I would have assumed that a context $x$ denotes a specific experimental task and so this would imply that the distribution is over repeatedly sampled protocols for the same specific experimental task $x$?

3. In Table 3, each row is the SCORE mechanism evaluated without a single component only?

4. In Section D of the appendix, Equations 13 and 14, the authors do not appear to give a KL penalty for the GRPO definition. Was the KL penalty omitted for the authors' experiments?

&nbsp;

**Details Of Ethics Concerns:**

&nbsp;

No ethical concerns identified.

&nbsp;

---

> ### Author Response · Authors · 2025-11-21
> **Response-1 to Reviewer btq2**
>
> Dear Reviewer btq2,
>
> Thank you for the positive and encouraging assessment of our work. We appreciate your recognition of Thoth, the SciRecipe dataset, the SCORE mechanism, and the rigor of our ablation and sensitivity analyses, as well as your remarks on the clarity of our writing and presentation. We also value your constructive concern regarding reproducibility. We address the points below. All corresponding revisions have been incorporated into the main paper and are highlighted in orange for clarity.
>
> **Re Weakness 1.1:** Thank you for pointing this out. We have prepared a detailed datasheet that documents the motivation, composition, curation process, and maintenance plan of the SciRecipe dataset. In accordance with the double-blind review policy of ICLR 2026, certain entries have been anonymized and will be fully released after the review process concludes. The datasheet has been included in our anonymous GitHub repository (https://anonymous.4open.science/r/SCORE3935) for reviewers’ reference.
>
> **Re Weakness 1.2:** We appreciate your feedback. We have made the core implementation publicly accessible in our anonymous GitHub repository (https://anonymous.4open.science/r/SCORE3935), and the associated models and datasets have also been uploaded to our HuggingFace repository (to be fully released after the review period).
>
> **Re weakness 2.1:**
> Thank you for noting the capitalization inconsistencies. We have corrected all missing capitalizations in the references.
>
> **Re weakness 2.2:**
> Thank you for pointing this out. We have added the missing conference information, including specifying that Ahmadian et al. 2024 was published at ACL.
>
> **Re weakness 2.3:**
> Thank you for your suggestion. We have added a discussion in the relevant work section, outlining the relationship between our approach and BioPlanner.
>
> **Re weakness 2.4:**
> Thank you for catching this formatting issue. We have added the missing commas at the end of Equations 1, 2, 3, and 5.
>
> **Re weakness 2.5:**
> Thank you for the clarification. We have updated the citation in the main text, consistent with the Appendix D.
>
> **Re weakness 2.6:**
> We appreciate the careful reading. We have corrected the typo on line 1093.
>
> **Re weakness 2.7:**
> Thank you for the suggestion. We have added reference [4] when discussing PPO in Appendix D.
>
>
> **Re Question 1:** Thank you for your valuable perspective. Your understanding is correct. In the Rollout 3 example shown in Figure 2, we placed a green tick mark in the order box because the sequence satisfies the **"Strict”** order consistency criterion. Since Figure 2 serves as an illustrative overview of the SCORE mechanism, we intended to highlight that the action ordering in Rollout 3 is indeed more reasonable compared with Rollouts 1 and 2.
>
> Importantly, both the "Strict” and "LCS” variants of order consistency are designed to provide **graded assessments** of ordering quality. As such, these measures do not imply a binary notion of "positive” or "negative,” except in extreme cases of perfect agreement or complete mismatch. During optimization, we compare rollouts based on their order consistency scores to encourage the model to prefer more coherent and logically structured action sequences.
>
> **Re Question 2:** Thank you for your meticulous observation. Your interpretation aligns well with our original intention. As the overall objective of the experimental protocol generation task, we ultimately aim for the policy model to maintain strong local and global logical consistency across repeated samplings—even under different temperature settings. Achieving such stability is crucial for enhancing the model’s robustness across varying conditions and ensuring that the generated protocols remain coherent and reliable.
>
> **Re Question 3:** Thank you for your detailed comments. To clarify, the third and sixth rows in Table 3 present ablation results based on different combinations of the core SCORE components. The purpose of these experiments is to demonstrate the individual contributions of each component and to verify the effectiveness of the complete SCORE mechanism by observing how performance changes when certain elements are included or removed.
>
> **Re Question 4:** Thank you for your suggestion to refine this aspect. In our training process, we did include a KL penalty term. This is consistent with the description in Section 3.4 of the original manuscript, where we noted that removing entropy loss and reducing the KL penalty helps increase exploration and prevents the policy from converging too early. To make this clearer, we have added a supplementary explanation to Equation 13 in Appendix D, which provides a more complete formalization of the RL algorithm used in our work. (The revision appears at Appendix D and is highlighted in orange.)

---

### Official Review · Reviewer_WL6o · 2025-11-01

**Soundness:** 3
**Presentation:** 4
**Contribution:** 3
**Rating:** 8
**Confidence:** 3

**Summary:**

This paper introduces a large scale dataset of protocols across many domains in biology. Authors also propose a “Sketch-and-Fill” reasoning paradigm, a structured approach to protocol generation to organize the output into reasoning, structuring and expression steps. This structured output allows them to define a multi gated reward function for formatting, consistency, order of steps etc.  Using these components, they present a trained model called Thoth which achieves improvement in performance compared to base LLMs.

**Strengths:**

- Authors presented comprehensive benchmarks across a wide range of LLMs. Authors also considered real-world context in experimental protocol creation in the way the framework and rewards are designed. The paper is also written well and the authors included a comprehensive appendix.
- The trained Thoth model (including the mini) demonstrates strong performance across all evaluation metrics in Table 1, highlighting both the model’s abilities and the value of the dataset introduced by the authors.

**Weaknesses:**

See questions below

**Questions:**

- Authors should compare the sketch and fill paradigm and the multi stage generation to performance of frameworks like chain of thought prompting to clarify the added value of sketch-and-fill for generating structured output.
- Evaluating models ability to do multi step mathematical reasoning is also another idea for benchmarking LLM models for protocol generation,  for tasks like adjusting reagent concentrations or dilution rations which could influence quantitative changes in the downstream steps.

---

> ### Author Response · Authors · 2025-11-21
> **Response-1 to Reviewer WL6o**
>
> Dear Reviewer WL6o,
>
> Thank you for the clear summary and thoughtful evaluation of our work. We appreciate your recognition of our large-scale protocol dataset, the "Sketch-and-Fill” reasoning paradigm, and the multi-gated reward design, as well as the performance gains of Thoth over base LLMs. These comments are encouraging and help affirm the core contributions of our paper. We address the points below. All corresponding revisions have been incorporated into the main paper and are highlighted in green for clarity.
>
> **Re Question 1:** Thank you for your insightful comment. We agree that evaluating different reasoning paradigms is essential for understanding how structured generation influences protocol quality. Our central claim is that the Sketch-and-Fill reasoning paradigm enables existing LLMs to conduct more organized, step-wise reasoning, thereby producing higher-quality experimental protocols. In the original manuscript (Figure 4), we compared **Sketch-and-Fill** with a direct **Fill** approach (``<think> → <orc>``) across several models (DeepSeek-V3, GPT-5 Chat, Thoth, and Qwen3-8B), and observed consistent performance improvements.
> Following your suggestion, we expanded our comparison to incorporate additional reasoning baselines:
> - **Standard CoT:** prompting the model with "Let’s think step by step” and directly generating the protocol.
> - **Advanced CoT:** prompting the model to first articulate its reasoning process and then produce the protocol. Unlike *Fill*, the resulting protocol is not required to follow a step-wise formatted structure.
>
> The results are summarized in the table below. Across all evaluated models, Sketch-and-Fill consistently outperforms both Standard and Advanced CoT, demonstrating clear advantages over conventional multi-stage reasoning strategies. Furthermore, these improvements are robust across different model families, strengthening the evidence for the effectiveness of the proposed paradigm. (The revision appears at Appendix I.4 and is highlighted in green.)
>
> |  Metrics |     Paradigms     | DeepSeek-V3 | GPT-5 Chat | Qwen3-8B |  Thoth |   AVG  |
> |:--------:|:-----------------:|:-----------:|:----------:|:--------:|:------:|:------:|
> | **BLUE-AVG** | Standard CoT        | 5.98        | 5.12       | 5.01     | 13.03 | 7.29  |
> |          | Advanced CoT        | 9.00        | 6.40       | 7.70     | 29.58 | 13.17 |
> |          | "Fill”              | 33.36       | 31.37      | 20.69    | 38.91 | 31.08 |
> |          | "Sketch-and-Fill”   | 38.17       | 29.57      | 16.66    | 43.25 | 31.91 |
> |----------|---------------------|-------------|------------|----------|-------|-------|
> | **ROUGE-L** | Standard CoT        | 19.82       | 17.41      | 15.06    | 27.54 | 19.96 |
> |          | Advanced CoT        | 28.11       | 22.14      | 19.91    | 41.34 | 27.88 |
> |          | "Fill”              | 44.21       | 41.10      | 33.38    | 46.44 | 41.28 |
> |          | "Sketch-and-Fill”   | 48.46       | 42.04      | 32.31    | 49.13 | 42.99 |
> |----------|---------------------|-------------|------------|----------|-------|-------|
> | **METEOR** | Standard CoT        | 24.92       | 22.73      | 18.61    | 25.78 | 23.01 |
> |          | Advanced CoT        | 26.90       | 23.53      | 17.65    | 36.51 | 26.15 |
> |          | "Fill”              | 41.29       | 38.03      | 36.36    | 44.01 | 39.92 |
> |          | "Sketch-and-Fill”   | 45.03       | 41.95      | 34.72    | 46.73 | 42.11 |
> |----------|---------------------|-------------|------------|----------|-------|-------|
> | **KW-F1** | Standard CoT        | 26.07       | 22.74      | 19.59    | 33.14 | 25.39 |
> |          | Advanced CoT        | 35.20       | 29.40      | 23.79    | 47.28 | 33.92 |
> |          | "Fill”              | 49.64       | 46.24      | 37.43    | 50.63 | 45.99 |
> |          | "Sketch-and-Fill”   | 54.28       | 49.87      | 38.63    | 53.57 | 49.09 |

---

> ### Author Response · Authors · 2025-11-21
> **Response-2 to Reviewer WL6o**
>
> **Re Question 2:** We appreciate your valuable suggestion. The mathematical reasoning ability you highlighted is indeed essential for experimental protocol generation, as it frequently involves core operations such as reagent concentration adjustments, dilution calculations, and scale conversions. To explicitly evaluate this capability, we intentionally included a dedicated **"Scaling”** subtask within the Problem-Solving Tasks of the SciRecipe dataset. This subtask focuses on numerical adjustments and unit conversions required to adapt an experimental setup to different scales.
>
> As a result, Thoth is able to maintain strong performance even when protocol generation requires multi-step mathematical reasoning. We also illustrate a representative example in Figure 15 of the original manuscript. In addition to qualitative demonstrations, we conducted quantitative evaluations (Table 17) on the "Scaling” subtask to more systematically assess model performance. The detailed results are provided in the table below.

---

> ### Author Response · Authors · 2025-11-21
> **Response-3 to Reviewer WL6o**
>
> | **Methods**                       | **Semantic-A** | **Order-LCS** | **Order-S** | **Order-Tau** | **Step-M** | **BLUE-AVG** | **ROUGE-L** | **METEOR** | **KW-F1** | **AVG** |
> |-----------------------------------|----------------|---------------|-------------|---------------|------------|--------------|-------------|------------|-----------|---------|
> | **ChatGPT-4o**                    |     **74.95**      |     77.87     |    38.00    |     71.99     |   50.00    |    51.36     |    62.12    |   58.19    |   69.93   |  61.60  |
> | **GPT-5**                         |     51.76      |     70.97     |    31.97    |     72.24     |   43.83    |    34.89     |    49.73    |   44.19    |   66.47   |  51.78  |
> | **GPT-5 Chat**                    |     62.62      |     80.70     |    44.00    |     70.00     |   50.00    |    45.85     |    59.93    |   57.32    |   70.79   |  60.13  |
> | **Claude Sonnet 4**               |     63.92      |     78.50     |    32.00    |     72.00     |   46.00    |    44.63     |    56.48    |   52.89    |   69.78   |  57.36  |
> | **Claude Opus 4.1**               |     65.47      |     82.42     |    46.70    |     77.39     |   51.12    |    51.38     |    63.26    |   58.90    |   73.30   |  63.33  |
> | **Gemini 2.5 Flash**              |     53.67      |     77.64     |    36.00    |     71.00     |   40.00    |    39.74     |    54.01    |   49.62    |   64.62   |  54.03  |
> | **Gemini 2.5 Pro**                |     56.34      |     78.80     |    32.00    |     78.00     |   34.00    |    38.05     |    54.83    |   59.07    |   64.73   |  55.09  |
> | **Doubao-1.5-pro**                |     59.81      |     81.55     |    44.00    |     77.00     |   52.00    |    41.74     |    61.12    |   54.51    |   68.72   |  60.05  |
> | **Qwen2.5-Max**                   |     65.70      |     81.38     |    40.00    |     82.00     |   50.00    |    37.65     |    61.83    |   55.95    |   70.13   |  60.52  |
> | **Qwen2.5-72B-Instruct**          |     62.21      |     79.42     |    34.00    |     76.00     |   46.00    |    35.97     |    58.81    |   56.04    |   68.89   |  57.48  |
> | **Qwen3-235B-A22B-Instruct-2507** |     58.53      |     79.31     |    33.97    |     74.00     |   41.97    |    45.93     |    57.84    |   53.45    |   67.16   |  56.91  |
> | **DeepSeek-V3**                   |     65.90      |     83.85     |    43.97    |     75.98     |   60.00    |    50.00     |    63.04    |   58.98    |   71.61   |  63.70  |
> | **GPT-OSS-120B**                  |     47.77      |     79.96     |    34.00    |     70.00     |   50.00    |    44.23     |    56.22    |   54.42    |   69.45   |  56.23  |
> | **Llama-3.1-405B-Instruct**       |     58.56      |     74.66     |    25.97    |     72.00     |   37.97    |    43.69     |    56.50    |   52.97    |   66.62   |  54.33  |
> | **Kimi-K2-Instruction**           |     62.16      |     76.45     |    38.00    |     70.00     |   52.00    |    44.62     |    59.76    |   53.93    |   68.27   |  58.35  |
> | **DeepSeek-R1**                   |     50.18      |     78.14     |    30.58    |     78.60     |   32.65    |    44.83     |    55.15    |   49.54    |   64.70   |  53.82  |
> | **Grok 3**                        |     67.04      |     78.41     |    27.72    |     72.64     |   36.69    |    46.19     |    59.32    |   56.66    |   69.14   |  57.09  |
> | **Grok 4**                        |     48.13      |     79.31     |    37.75    |     78.98     |   39.74    |    46.50     |    57.48    |   50.66    |   67.51   |  56.23  |
> | **OpenAI o1**                     |     60.64      |     77.34     |    28.00    |     70.00     |   44.00    |    46.62     |    58.80    |   52.93    |   69.43   |  56.42  |
> | **OpenAI o3**                     |     65.13      |     80.59     |    35.94    |     72.00     |   41.94    |    42.92     |    60.70    |   54.73    |   71.65   |  58.40  |
> | **Qwen3-4B**                      |     29.58      |     40.68     |    18.00    |     40.00     |   26.00    |    16.51     |    25.92    |   22.42    |   30.26   |  27.71  |
> | **Thoth-mini**                    |     68.87      |     85.46     |    47.98    |     88.31     |   68.34    |    54.25     |    63.90    |   62.71    |   72.65   |  68.05  |
> | **Qwen3-8B**                      |     36.87      |     60.84     |    18.00    |     60.00     |   26.00    |    20.87     |    41.02    |   40.61    |   51.26   |  39.50  |
> | **Thoth**                         |     74.83      |     **87.69**     |    **53.99**    |     **93.65**     |   **69.66**    |    **55.02**     |    **66.14**    |   **63.61**    |   **74.15**   |  **70.97**  |

---

### Author Response · Authors · 2025-12-01
**Summary Statement**

# Summary Statement for the Area Chair

Dear Area Chair,

Thank you for overseeing the review process of our submission. We are grateful for the reviewers’ thoughtful and constructive feedback. During the rebuttal period, we conducted additional experiments, released supplementary resources, and strengthened the manuscript to address all raised concerns. Below is a concise summary of our contributions, the reviewers’ key points, and the revisions made.

### 1. Summary of Contributions
Our work proposes a unified framework for **scientific protocol generation**, addressing long-standing challenges in accuracy, logical structure, and experimental executability. The main components include:
- **SciRecipe**: a dataset of over 12K structured biological protocols spanning 27 subfields.
- **Sketch-and-Fill**: a formal specification that enforces explicit reasoning and atomic step representation.
- **SCORE**: a structured, component-based reward that evaluates step granularity, ordering, and semantic fidelity.
- **Thoth**: an RL-finetuned model achieving SOTA performance on SciRecipe-Eval and public scientific QA benchmarks.

Together, these components significantly improve the reliability and executability of LLM-generated scientific protocols.

### 2. Reviewer-Specific Summary of Issues & Revisions
#### **Reviewer WL6o** (Rating: 8 / Confidence: 3)
- **Issues:** Comparison against CoT-based reasoning; evaluation of mathematical reasoning for scaling tasks.
- **Revisions:** Added Standard/Advanced CoT and Fill baselines, demonstrating consistent advantages of Sketch-and-Fill. Highlighted the built-in *Scaling* subtask and expanded quantitative evaluations showing strong numerical reasoning capabilities.

#### **Reviewer btq2** (Rating: 8 / Confidence: 5)
- **Issues:** Reproducibility (datasheet, code availability); citation/formatting issues; RL-related clarifications.
- **Revisions:** Released an anonymous codebase and a complete SciRecipe datasheet; corrected all citation and formatting inconsistencies. Clarified equations, ablation interpretation, and KL penalty usage within GRPO.

#### **Reviewer RwGZ** (Rating: 6 / Confidence: 3)
- **Issues:** Scalability across model sizes; correlation between SCORE and real experimental executability; generalizability to other scientific domains.
- **Revisions:** Extended training to Qwen3-14B and Qwen3-32B, obtaining Thoth-14B/32B with further performance gains. Conducted expert evaluations showing strong alignment between SCORE and human judgments. Provided chemistry examples illustrating domain-agnostic applicability.

#### **Reviewer ULSZ** (Rating: 2 / Confidence: 3)
- **Issues:** Whether SciRecipe-Eval truly benchmarks protocol generation; potential restrictiveness of SAF; justification for SCORE; fairness of evaluation metrics; missing citations.
- **Revisions:** Clarified that all tasks require full protocol generation; added SAF coverage, inter-annotator agreement, and conditional-entropy analyses. Introduced expert studies validating SCORE’s sensitivity to critical errors. Integrated evaluation metric descriptions into the main text and added missing references.

### 3. Timeline of Discussion
- **12 Nov 2025, 10:50** — Initial reviews received.
- **21 Nov 2025, 17:56–18:35** — Submitted our full rebuttal, including extensive new analyses (CoT baselines, SAF validation, larger-model experiments, expert studies, and SCORE sensitivity tests), code/data release, and major manuscript revisions.
- **26 Nov 2025, 17:34** — Reviewer ULSZ confirmed that all concerns were addressed and **raised the score from 2 → 6** (see “Official Review of Submission1592 by Reviewer ULSZ,” Revisions section).
- **27 Nov 2025, 06:29** — Reviewer RwGZ acknowledged that our rebuttal fully resolved all questions and **raised the score from 6 → 8** (see “Official Review of Submission1592 by Reviewer RwGZ,” Revisions section).

Thank you again for your time and consideration. We hope the new results, clarifications, and released resources demonstrate the strength, robustness, and broader scientific relevance of our contributions. We are happy to provide any further information if needed.

Sincerely,
The Authors

---

### Meta-Review · Area_Chair_TXUU · 2026-01-08

**Summary:**

The paper proposed a comprehensive framework, including a new dataset (SciRecipe) and metric (SCORE) for experimental protocol generation, by formulating protocol generation as a structured reasoning process, where each step is decomposed into essential components via natural language expressions. The paper introduces a formalization of the protocol generation process (Sketch-and-Fill) and uses it in conjunction with the SCORE reward to train a base model via supervised fine-tuning and GRPO on the collected datasets and reward function. The effectiveness of this framework is demonstrated via the trained model, which achieves higher performance than several baselines on the SciRecipe benchmark and other public scientific benchmarks.

The reviewers generally express positive support of the paper, with some concerns about the evaluation aspects (larger model, comparison with COT prompting, etc..) and concerns about the details of the design choices (metric, justification of reward function). I’d like to recommend the paper for acceptance and encourage the authors to revise the paper accordingly based on the reviewers’ feedback in its final version.

**Reviewer Concerns:**

- WL6o
  - Compare the sketch and fill paradigm and the multi stage generation to performance of frameworks like chain of thought prompting to clarify the added value of sketch-and-fill for generating structured output. `resolved`
  - Evaluating model's ability to do multi step mathematical reasoning is also another idea for benchmarking LLM models for protocol generation. `resolved`
- btq2
  - fully describe the curation process and detail the authors' plans for maintenance of the dataset. `resolved`
  - code release. `resolved`
- RwGZ
  - Evaluation beyond Qwen3-8B backbone (larger parameters to 32B, or other pre-trained such as deepseek-v3) `resolved`
  - clarification on SCORE's correlation to experimental executability and motivations behind step-level alignment? `resolved`
  - risk of overfitting and generalization to other scientific domains `resolved`
- ULSZ
  - evaluation tasks are not actually benchmarking protocol generation `resolved`
  - justification of the design choices `resolved`

**Reviewer Scores:**

* WL6o (8)
* btq2 (8)
* ULSZ (2)
* RwGZ (6)

ULSZ and RwGZ increased scores to 6 and 8, respectively.

---

### Decision · Program_Chairs · 2026-01-26

Accept (Poster)